# Optogenetic investigation of BMP target gene expression diversity

Katherine W Rogers[1], Mohammad ElGamacy[1,2,3†], Benjamin M Jordan[4†], Patrick Müller[1,2]*

[1]Systems Biology of Development Group, Friedrich Miescher Laboratory of the Max Planck Society, Tübingen, Germany; [2]Modeling Tumorigenesis Group, Translational Oncology Division, Eberhard Karls University Tübingen, Tübingen, Germany; [3]Heliopolis Biotechnology Ltd, London, United Kingdom; [4]Department of Organismic and Evolutionary Biology, Harvard University, Cambridge, United States

**Abstract** Signaling molecules activate distinct patterns of gene expression to coordinate embryogenesis, but how spatiotemporal expression diversity is generated is an open question. In zebrafish, a BMP signaling gradient patterns the dorsal-ventral axis. We systematically identified target genes responding to BMP and found that they have diverse spatiotemporal expression patterns. Transcriptional responses to optogenetically delivered high- and low-amplitude BMP signaling pulses indicate that spatiotemporal expression is not fully defined by different BMP signaling activation thresholds. Additionally, we observed negligible correlations between spatiotemporal expression and transcription kinetics for the majority of analyzed genes in response to BMP signaling pulses. In contrast, spatial differences between BMP target genes largely collapsed when FGF and Nodal signaling were inhibited. Our results suggest that, similar to other patterning systems, combinatorial signaling is likely to be a major driver of spatial diversity in BMP-dependent gene expression in zebrafish.

*For correspondence:
pmueller@tuebingen.mpg.de

†These authors contributed equally to this work

## Introduction

Embryogenesis is orchestrated by signaling pathways that activate spatiotemporally diverse patterns of gene expression. A prominent theory relating signaling to gene expression diversity is the gradient threshold model, in which a signaling gradient across a tissue defines unique spatial gene expression domains by activating target genes at different signaling thresholds (*Figure 1A*; *Sharpe, 2019*; *Briscoe and Small, 2015*; *Dubrulle et al., 2015*; *Rogers and Schier, 2011*; *Barkai and Shilo, 2009*; *Ashe and Briscoe, 2006*). Gene expression patterns can also be influenced by signaling dynamics and expression kinetics (*Sagner and Briscoe, 2017*) as well as interactions with other signaling pathways (*Briscoe and Small, 2015*). However, in many patterning systems the factors leading to diverse developmental gene expression profiles are incompletely characterized. Here, we investigate how signaling levels, target gene expression kinetics, and combinatorial signaling contribute to gene expression diversity during dorsal-ventral patterning in zebrafish.

We focused on patterning mediated by BMP, a TGF-β superfamily member with important developmental roles across the animal kingdom (reviewed in *Zinski et al., 2018*). BMP ligands bind and assemble complexes of type I and II receptor serine/threonine kinases, resulting in the phosphorylation of the signal transducers Smad1/5/9 and activation of BMP target genes (*Figure 1B*; *Derynck and Budi, 2019*). The regulation of BMP gradient formation during early development has been analyzed in a variety of organisms including *Drosophila*, *Nematostella*, and *Xenopus* (*Genikhovich et al., 2015*; *Iber and Gaglia, 2007*; *Mizutani et al., 2005*; *Plouhinec et al., 2013*) as well as zebrafish. During late blastula and early gastrulation stages in zebrafish embryos, graded transcription and subsequent diffusion of BMP ligands, together with

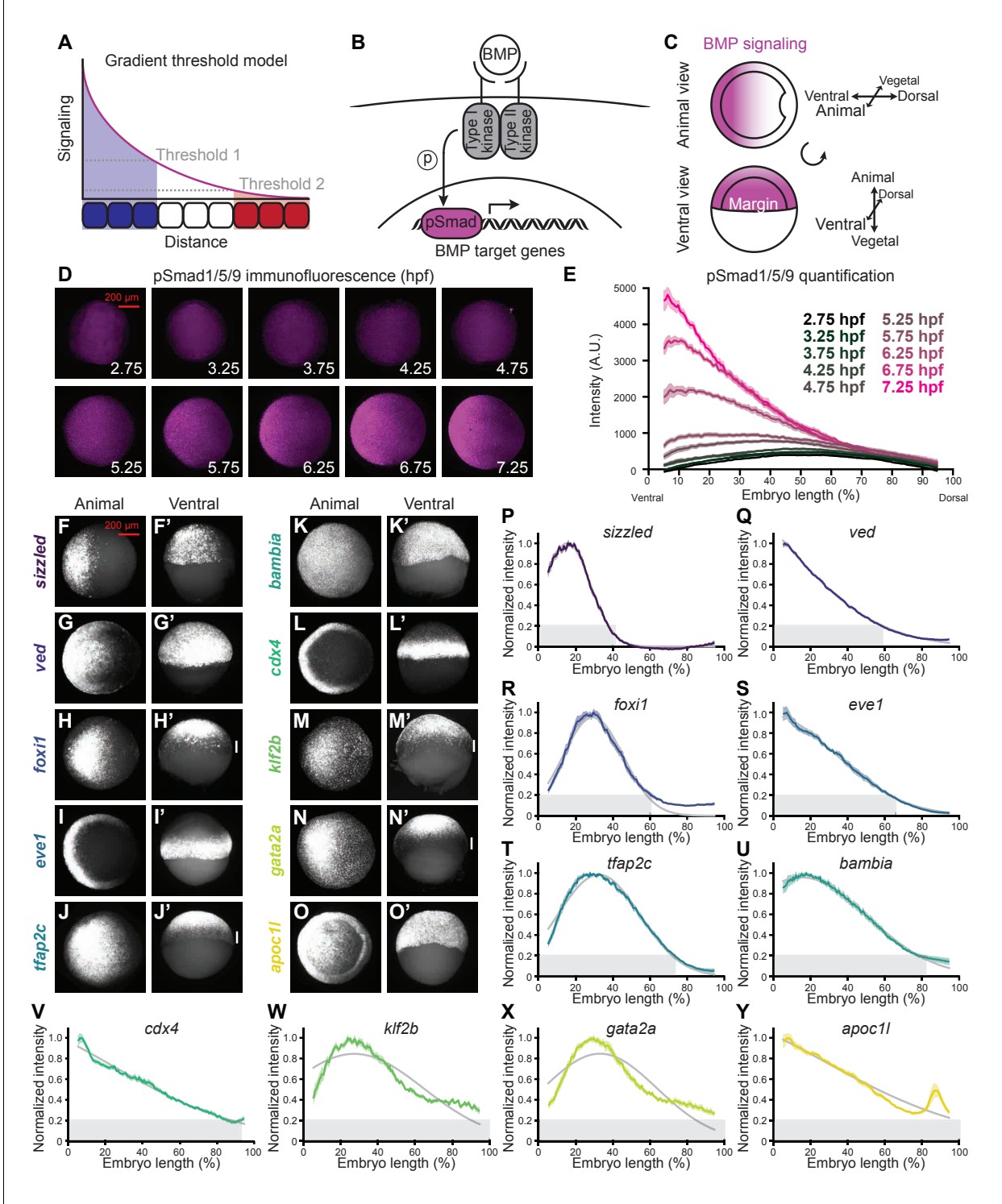

**Figure 1.** BMP target genes have diverse spatial expression patterns at shield stage. (A) The gradient threshold model states that a signaling gradient activates genes (blue, white, red) at different thresholds (dashed gray lines). (B) BMP binding induces receptor complex formation, phosphorylation of Smad1/5/9, and activation of target genes. (C) Schematic of shield-stage zebrafish embryos with BMP signaling gradients (magenta) along the dorsal-ventral axis. (D-E) Representative images (D) of pSmad1/5/9 immunofluorescence in embryos at the indicated time post-fertilization and quantification (E). (F-O′) Fluorescence *in situ* hybridization (FISH) showing spatial expression of the indicated high-confidence BMP target genes at shield stage (~6.75 h post-fertilization (hpf)). (F,G,H,I,J,K,L,M,N,O) are animal views, dorsal to the right. (F′,G′,H′,I′,J′,K′,L′,M′,N′,O′) are ventral views. Vertical white bars indicate regions where expression is excluded from the margin. (P-Y) Quantification of FISH signal along the dorsal-ventral axis for the indicated BMP

*Figure 1 continued on next page*

*Figure 1 continued*

target genes at shield stage (ventral on the left, dorsal on the right as in (E)). Normalized intensities are shown; error bars represent standard error. The Gaussian function $Ae^{-\frac{(x-\mu)^2}{\varsigma}}$ was fitted to each profile (gray lines), and gene expression range was defined as $r = \mu + 2\sqrt{\varsigma/2}$ (gray bars). Some BMP target genes could not be reliably quantified due to weak FISH signal (*bmp4*, *id2a*, *smad6a*, *smad7*, and *znfl2b*) or inability to reliably identify the ventral side in all assays (*crabp2b*). See the *Figure 1—source data 1* file for source data.

The online version of this article includes the following source data and figure supplement(s) for figure 1:

**Source data 1.** Source data for *Figure 1*.
**Figure supplement 1.** BMP target gene identification and spatial quantification.
**Figure supplement 1—source data 1.** Source data for *Figure 1—figure supplement 1*.

dorsally secreted BMP inhibitors such as Chordin, generate a ventrally-peaking gradient of BMP signaling that patterns the dorsal-ventral axis (*Figure 1C*; *Pomreinke et al., 2017*; *Zinski et al., 2017*). Loss of BMP signaling results in dorsalization, whereas excess BMP signaling produces ventralized embryos (*Zinski et al., 2018*). The degree of dorsalization or ventralization can be modulated by mutations in BMP pathway components with different strengths (*Mintzer et al., 2001*; *Barth et al., 1999*; *Nguyen et al., 1998*; *Mullins et al., 1996*) or by injecting different amounts of mRNA encoding pathway activators or inhibitors (*Schumacher et al., 2011*; *Dick et al., 2000*; *Kishimoto et al., 1997*; *Neave et al., 1997*).

These observations have led to the suggestion that BMP functions as a morphogen to pattern the dorsal-ventral axis by activating different target genes at different signaling level thresholds (*Figure 1A*; *Zinski et al., 2018*; *Tuazon and Mullins, 2015*; *Schumacher et al., 2011*; *Barth et al., 1999*; *Nguyen et al., 1998*; *Neave et al., 1997*; *Mullins et al., 1996*). However, overexpression and genetic manipulations can affect the duration of signal exposure, dysregulate other signaling pathways, and modify earlier aspects of development such as morphogenetic movements, complicating the interpretation of these experiments. Moreover, patterning of the dorsal-ventral axis by BMP and the germ layers by FGF and Nodal occurs simultaneously in zebrafish (*Zinski et al., 2018*), and although these pathways are known to interact, how FGF and Nodal influence the spatiotemporal expression of BMP target genes has not been systematically assessed.

To identify the factors that contribute to differences in BMP target gene expression and rule out factors that do not contribute, we first identified BMP targets in early zebrafish embryos and quantified their diverse spatial (*Figure 1*) and temporal (*Figure 2*) expression patterns. We then used an optogenetic approach to generate acute BMP signaling pulses (*Figure 3*) and found that while most target genes can respond to early BMP signaling (*Figure 4*), differential transcription kinetics do not fully account for the observed expression differences (*Figure 5*). Further, target gene responses to high- and low-amplitude signaling pulses suggest that not all spatiotemporal target gene expression differences are due to different signaling activation thresholds (*Figure 6*). In contrast, inhibition of FGF and Nodal signaling homogenized the spatial expression patterns of BMP targets, suggesting that combinatorial regulation by BMP, FGF, and Nodal is a major driver of BMP target gene spatial diversity (*Figure 7*).

## Results

### BMP target genes have diverse spatiotemporal expression patterns

We used RNA-sequencing to systematically identify genes activated by BMP during early zebrafish gastrulation, when BMP is engaged in dorsal-ventral patterning (shield stage,~6.75 h post-fertilization (hpf)) (*Zinski et al., 2018*). We identified 16 high-confidence target genes that are significantly upregulated in *bmp*-overexpressing embryos and downregulated in embryos overexpressing the BMP inhibitor *chordin* (*Figure 1—figure supplement 1A–D* and *Supplementary file 1*). 14 of these genes (*apoc1l*, *bambia*, *bmp4*, *cdx4*, *eve1*, *foxi1*, *gata2a*, *id2a*, *klf2b*, *smad6a*, *smad7*, *sizzled*, *tfap2c*, and *ved*) are known to be positively regulated by BMP in zebrafish (*Kashiwada et al., 2015*; *Wang et al., 2015*; *Kotkamp et al., 2014*; *Wang et al., 2013*; *Das and Crump, 2012*; *de Pater et al., 2012*; *Kwon et al., 2010*; *Li and Cornell, 2007*; *Poulain et al., 2006*; *Chong et al., 2005*;

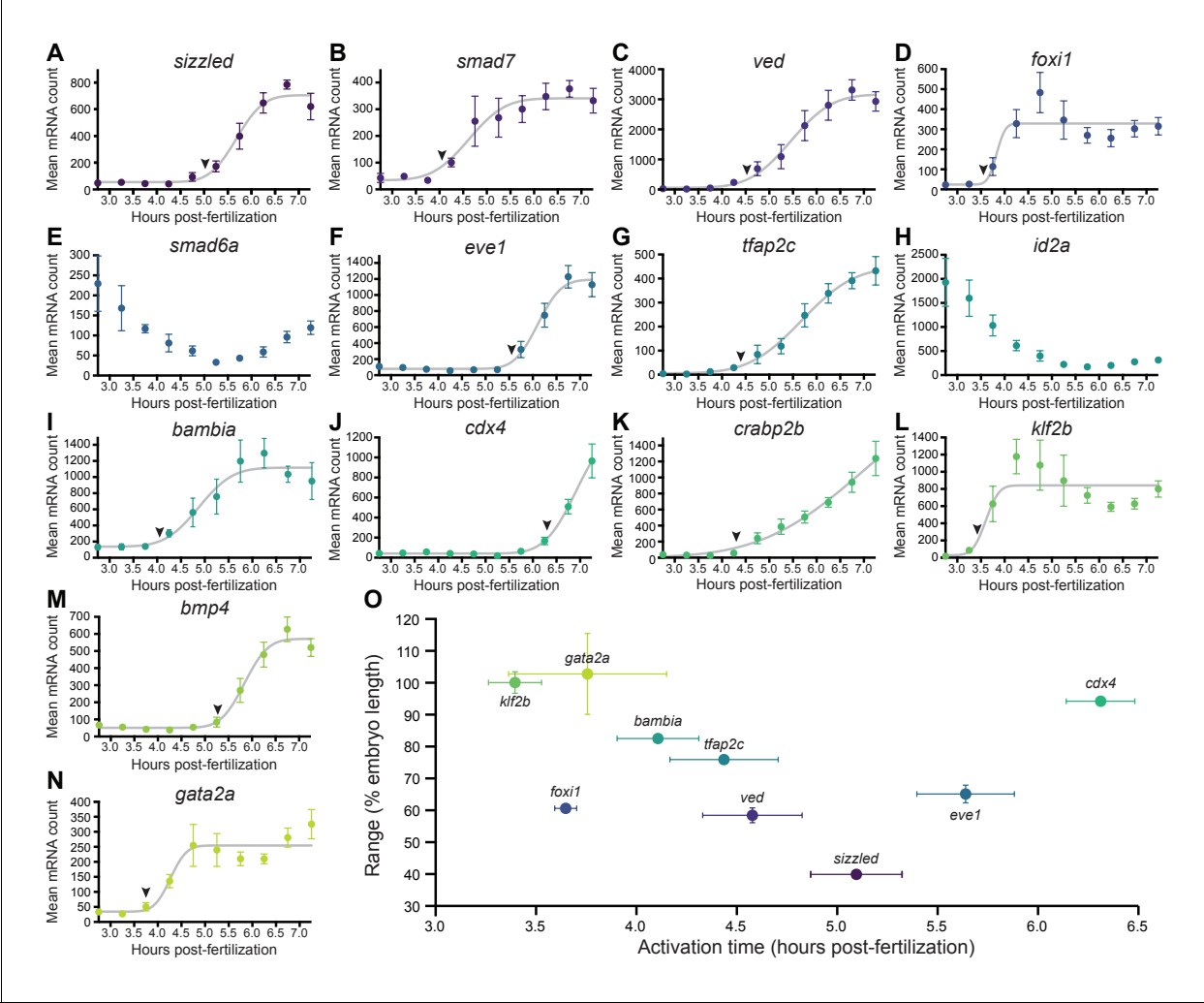

**Figure 2.** BMP target genes have diverse temporal expression profiles. (**A–N**) Embryos were collected every 30 min from 2.75 to 7.25 hpf, and transcript levels were quantified using NanoString technology. Error bars represent standard error. Temporal profiles were fit with the cumulative distribution function of the normal distribution (gray lines), and activation time (arrowheads) was defined as the average time point at which the curves reached about two mean average deviations (i.e., $1.5 \cdot \tau$) from the inflection point $\nu$ (excluding the maternally deposited genes *id2a* [*Chong et al., 2005*] and *smad6a* [*White et al., 2017*]). NanoString probes for two high-confidence activated BMP target genes (*apoc1l* and *znfl2b*) were not functional. (**O**) Average gene expression spatial range is plotted against average activation time. See the *Figure 2—source data 1* file for source data.

The online version of this article includes the following source data for figure 2:

**Source data 1.** Source data for *Figure 2*.

*Davidson et al., 2003*; *Martyn and Schulte-Merker, 2003*; *Nissen et al., 2003*; *Solomon et al., 2003*; *Yabe et al., 2003*; *Pogoda and Meyer, 2002*; *Shimizu et al., 2002*; *Oates et al., 2001*; *Tsang et al., 2000*; *Chin et al., 1997*; *Nikaido et al., 1997*; *Hammerschmidt et al., 1996a*; *Hammerschmidt et al., 1996b*; *Mullins et al., 1996*; *Detrich et al., 1995*; *Joly et al., 1993*; *Joly et al., 1992*), whereas *crabp2b* (*Sharma et al., 2005*) and *znfl2b* (*Hogan et al., 2006*) have not previously been implicated as BMP targets. Four of the 16 target genes encode repressors of BMP signaling (*bambia*, *sizzled*, *smad6a*, and *smad7*) and one encodes *bmp4*, consistent with roles for negative and positive feedbacks in TGF-β-mediated patterning (*Zinski et al., 2018*).

According to the gradient threshold model, target genes are activated by distinct signaling levels, leading to different spatial domains of target gene expression in the presence of a signaling gradient (*Figure 1A*). To determine whether the BMP patterning system fits this paradigm, we first sought to characterize both BMP signaling distribution and spatial target gene expression. We assessed spatial BMP signaling from 2.75 to 7.25 hpf (256-cell stage – 60% epiboly) using immunofluorescent

stainings to detect the BMP signal transducer pSmad1/5/9. We imaged embryos using selective plane illumination microscopy (SPIM) and quantified fluorescence along the dorsal-ventral axis (*Figure 1—figure supplement 1E*, Materials and methods). Similar to previous studies (*Pomreinke et al., 2017*; *Zinski et al., 2017*; *Ramel and Hill, 2013*; *Tucker et al., 2008*), we observed a ventrally-peaking BMP signaling gradient that increases in amplitude over time (*Figure 1D,E*).

We then used fluorescence *in situ* hybridization and SPIM to quantify the spatial expression profiles of BMP target genes along the dorsal-ventral axis at shield stage (~6.75 hpf) and found that target genes have different expression profiles along this axis (*Figure 1F–Y*, *Figure 1—figure supplement 1E*, Materials and methods; some genes could not be quantified due to weak signal or inability to reliably identify the ventral side). The shape of the expression profiles can be well described by bell curves. We therefore used regression analysis with the Gaussian function

$$Ae^{-\frac{(x-\mu)^2}{\varsigma}}$$

and defined the range of each target gene as

$$r = \mu + 2\sqrt{\varsigma/2}$$

Using this definition, spatial gene expression broadness ranges from 40–100% dorsal-ventral embryo length (*Figure 1F–Y*). Strikingly, pronounced differences along the orthogonal animal-vegetal axis were also evident: Genes were either uniformly expressed along this axis on the ventral side (*sizzled*, *ved*, *apoc1l*, and *bambia*), restricted to the margin (*cdx4* and *eve1*), or excluded from the margin (*foxi1*, *klf2b*, *gata2a*, and *tfap2c*) (*Figure 1C,F–O'*). Margin exclusion resulted in distinct dorsal-ventral profiles in which mRNA levels peak around 30% embryo length (*Figure 1R,T,W,X*), compared to non-excluded genes that peaked more ventrally (*Figure 1P,Q,S,U,V,Y*). Therefore, some of the spatial diversity in BMP target gene expression arises from differences along the animal-vegetal axis, orthogonal to the dorsal-ventral BMP signaling gradient.

The gradient threshold paradigm (*Figure 1A*) implies that genes with broad ranges should be activated by lower signaling levels. Since signaling levels increase over time (*Figure 1D,E*; *Pomreinke et al., 2017*; *Zinski et al., 2017*; *Ramel and Hill, 2013*; *Tucker et al., 2008*), we sought to determine whether broadly expressed targets were activated earlier. To assess temporal expression of BMP targets, we used NanoString molecular barcoding (*Kulkarni, 2011*) to measure transcript levels from 2.75 to 7.25 hpf (256-cell stage – 60% epiboly) (*Figure 2A–N*). The shape of the temporal expression profiles can be well approximated by the modified cumulative distribution function of the normal distribution

$$\frac{1}{2}A\left(1 + \mathrm{erf}\left(\frac{x-\nu}{\tau\sqrt{2}}\right)\right) + b$$

We used this function for regression analysis of the temporal expression profiles and defined activation times as the average time point at which the curves reached about two mean average deviations (i.e., $1.5 \cdot \tau$) from the inflection point $\nu$. BMP target gene activation times defined in this way ranged from 3.4 to 6.3 hpf (*Figure 2*).

The gradient threshold model predicts a monotonic decrease when comparing range and activation time. While this relationship is not observed for the entire dataset (*Figure 2O*), there is a decreasing monotonic trend when *foxi1, eve1,* and *cdx4* are excluded (note that in contrast to the other genes, the expression of *eve1* and *cdx4* was only quantified in the embryonic margin [*Figure 1—figure supplement 1E*, Materials and methods]). This suggests the possibility that subsets of BMP target genes may behave consistently with the gradient threshold model. We therefore sought to investigate the relationship between BMP signaling and target gene expression further using an optogenetic strategy.

## Reversible optogenetic activation of BMP signaling *in vivo* using Opto-BMP

To assess how BMP target genes respond to BMP signaling, we developed a method to optogenetically manipulate BMP signaling *in vivo*. We fused zebrafish BMP receptor kinase domains to an algal

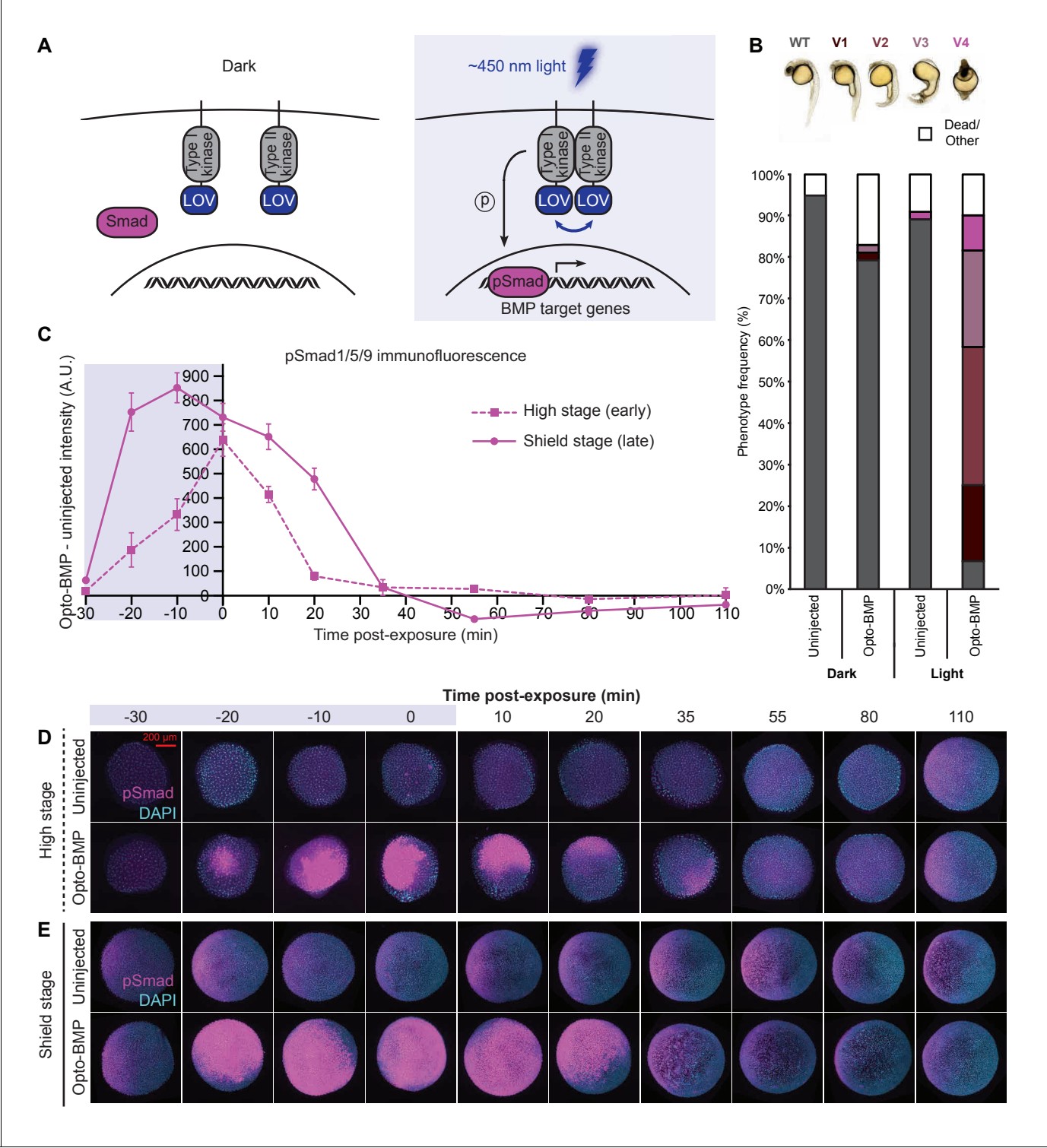

**Figure 3.** Reversible activation of BMP signaling using blue light-activated Opto-BMP. (**A**) Schematic of Opto-BMP strategy. Blue light-dimerizable VfLOV domains were fused to zebrafish BMP receptor kinase domains. Blue light exposure activates BMP signaling. (**B**) Embryos injected with mRNA encoding Opto-BMP at the one-cell stage and their uninjected siblings were reared in the dark or exposed to blue light for 10 h starting 70–80 min post-fertilization. Ventralization phenotypes V1-V4 (indicating excess BMP signaling) were scored at 1 day post-fertilization. Number of embryos: uninjected dark = 59, Opto-BMP dark = 53, uninjected light = 55, Opto-BMP light = 60. (**C-E**) Uninjected and Opto-BMP-injected embryos were exposed to blue light (2300 lux) for 30 min starting at high stage (3.5 hpf) or shield stage (6.75 hpf) and fixed during and after exposure. pSmad1/5/9 immunofluorescence was quantified and plotted in (**C**) as Opto-BMP minus uninjected signal with piecewise linear interpolation between timepoints;

*Figure 3 continued on next page*

*Figure 3 continued*

error bars represent standard error (see Materials and methods for statistical analysis). Blue background represents light exposure. Representative embryos from the high-stage (D) and shield-stage (E) experiments quantified in (C). pSmad1/5/9 signal is shown in magenta, DAPI in cyan. See the ***Figure 3—source data 1*** file for source data.

The online version of this article includes the following source data and figure supplement(s) for figure 3:

**Source data 1.** Source data for *Figure 3*.
**Figure supplement 1.** Opto-BMP characterization.
**Figure supplement 1—source data 1.** Source data for *Figure 3—figure supplement 1*.

blue light-homodimerizable LOV domain (*Rogers and Müller, 2020*; *Takahashi et al., 2007*) and targeted the fusions to the membrane using a myristoylation motif (*Figure 3A*), similar to previous approaches (*Ramachandran et al., 2018*; *Vopalensky et al., 2018*; *Sako et al., 2016*). Blue light (~450 nm) exposure should lead to dimerization of the LOV domains and interaction of the BMP kinase domains, activating BMP signaling (*Figure 3A*).

Injection of mRNA encoding Opto-BMP into zebrafish embryos at the one-cell stage resulted in strong ventralization in light-reared embryos, consistent with excess BMP signaling, whereas dark-reared siblings were mostly aphenotypic (*Figure 3B* and *Figure 3—figure supplement 1A,K,L*). Spatially localized activation of BMP signaling was also possible using SPIM, further demonstrating light-dependent signaling activation (*Figure 3—figure supplement 1E–H*).

To facilitate optogenetic experiments, we developed a light exposure device by embedding blue LEDs into the lid of a standard six-well plate and controlling light intensity and dynamics with a single-board computer (*Figure 3—figure supplement 1B–D*, Materials and methods). Using the LED array, we exposed uninjected and Opto-BMP-injected embryos to blue light for 30 min during high (3.5–4 hpf) or shield (6.75–7.25 hpf) stages, fixed embryos during and after exposure, and quantified BMP signaling using pSmad1/5/9 immunofluorescence (*Figure 3C–E* and *Figure 3—figure supplement 1I,J*). At both stages, Opto-BMP embryos showed a dramatic increase in BMP signaling within 10 min of light exposure, and signaling levels returned to normal after light removal. These experiments demonstrate that Opto-BMP reversibly activates BMP signaling in zebrafish embryos in response to light.

## Most BMP target genes are competent to respond to BMP at early stages

BMP target genes are activated over a range of developmental stages, from 3.4 to 6.3 hpf (*Figure 2*). Time-dependent differences in competence – a gene's ability to respond to signaling – may underlie the diversity in activation timing (*Figure 4A*). To test this, we quantified BMP target gene expression in uninjected and Opto-BMP-injected embryos exposed to 30 min blue light during either high (3.5–4 hpf) or shield stage (6.75–7.25 hpf) (*Figure 4* and *Figure 5—figure supplement 1A–F*).

In response to a strong BMP signaling pulse at high or shield stage (*Figure 3C–E* and *Figure 3—figure supplement 1I,J*), we observed corresponding significant pulses of BMP target gene expression for all genes except *crapb2b* and *cdx4* (*Figure 4K,L*). While *cdx4* is not competent to respond to an early BMP signaling pulse and *crabp2b* did not clearly respond to either an early or late signaling pulse, all other tested high-confidence BMP target genes responded at high stage. Therefore, differences in competence to respond to BMP signaling at early stages do not explain the majority of diversity in activation timing.

## Transcription kinetics in response to BMP do not fully explain spatiotemporal expression

Target gene transcription kinetics can play important roles in defining spatial expression domains. For example, it has been suggested that Nodal target genes with faster transcript accumulation rates have broader spatial expression domains (*Dubrulle et al., 2015*). To investigate how the transcription kinetics of BMP target genes may influence their spatiotemporal expression patterns, we assessed the dynamics of target gene responses (*Figure 4*) to

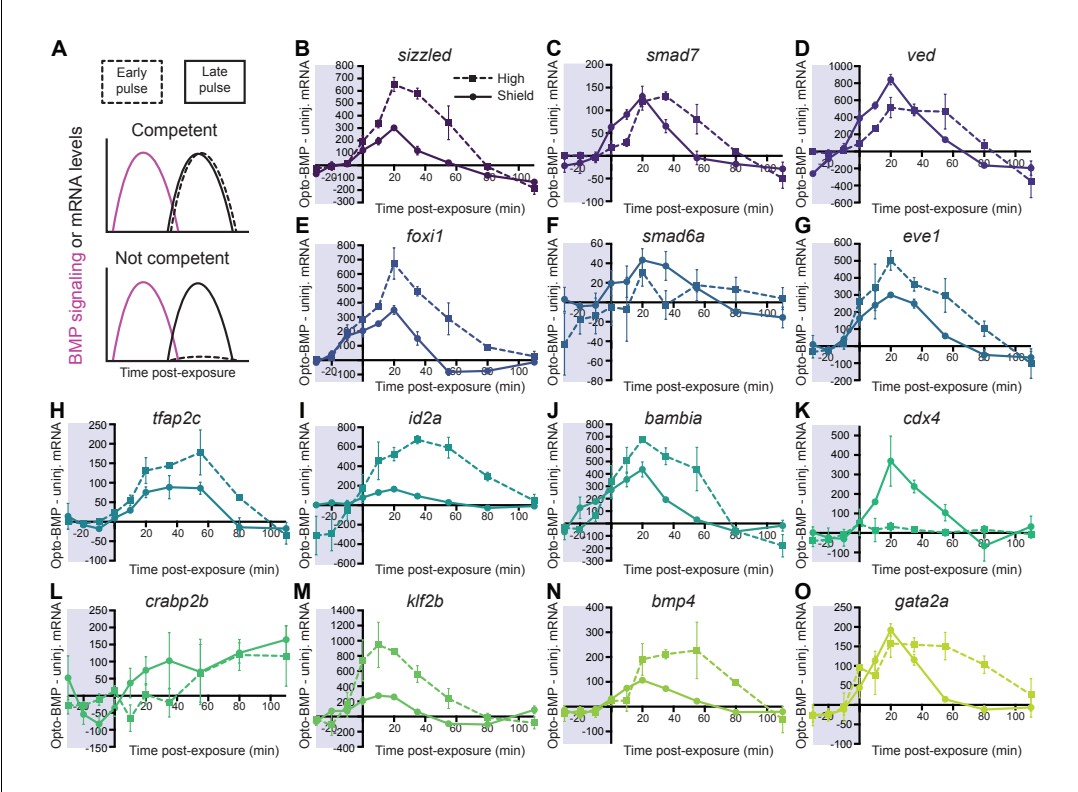

**Figure 4.** Most BMP target genes are competent to respond to an early BMP signaling pulse. (**A**) Schematic of competence model: Late-activated genes should respond to a late (shield stage, solid line), but not early (high stage, dashed line) BMP signaling pulse. (**B-O**) High-confidence BMP target gene responses after an early (high stage,~3.5 hpf, dashed line) or late (shield stage,~6.75 hpf, solid line) BMP signaling pulse delivered by exposing uninjected and Opto-BMP-injected embryos to 30 min blue light (*Figure 3C–E* and *Figure 3—figure supplement 1I,J*). To assess induced transcription, NanoString transcript counts from uninjected embryos were subtracted from Opto-BMP transcript counts and are plotted here with piecewise linear interpolation between timepoints; error bars represent standard error (see Materials and methods for statistical analysis). See the *Figure 4—source data 1* file for source data.

The online version of this article includes the following source data for figure 4:

**Source data 1.** Source data for *Figure 4*.

optogenetically generated BMP signaling pulses (*Figure 3C–E* and *Figure 3—figure supplement 1I,J*). We reasoned that the early activation timing and broad spatial range of some BMP targets might be explained by more rapid transcription in response to BMP. In this paradigm, early BMP signaling activates expression of all target genes at the same time, but transcripts of more slowly transcribed genes only accumulate to detectable levels at later stages, causing them to appear to be 'late-activated' (*Figure 5A*). Similarly, broader spatial ranges could be caused by faster accumulation of rapidly produced transcripts that would therefore be detectable farther from the ventral side than more slowly produced transcripts.

To determine whether higher transcript accumulation rates correlate with broader spatial ranges or earlier activation times, we first assessed maximum transcript counts in response to BMP signaling pulses at high or shield stage (*Figure 4*). Assuming similar transcript degradation kinetics, transcripts with faster production rates should accumulate to higher levels in response to a BMP signaling pulse (*Figure 5A*). However, we observed a weak negative correlation (*Figure 5B*) or no correlation (*Figure 5D*) between maximum transcript counts and activation time, and found similar results for range (*Figure 5C,E*). This suggests that differences in transcript accumulation rates in response to BMP do not fully account for differences in activation timing and spatial broadness.

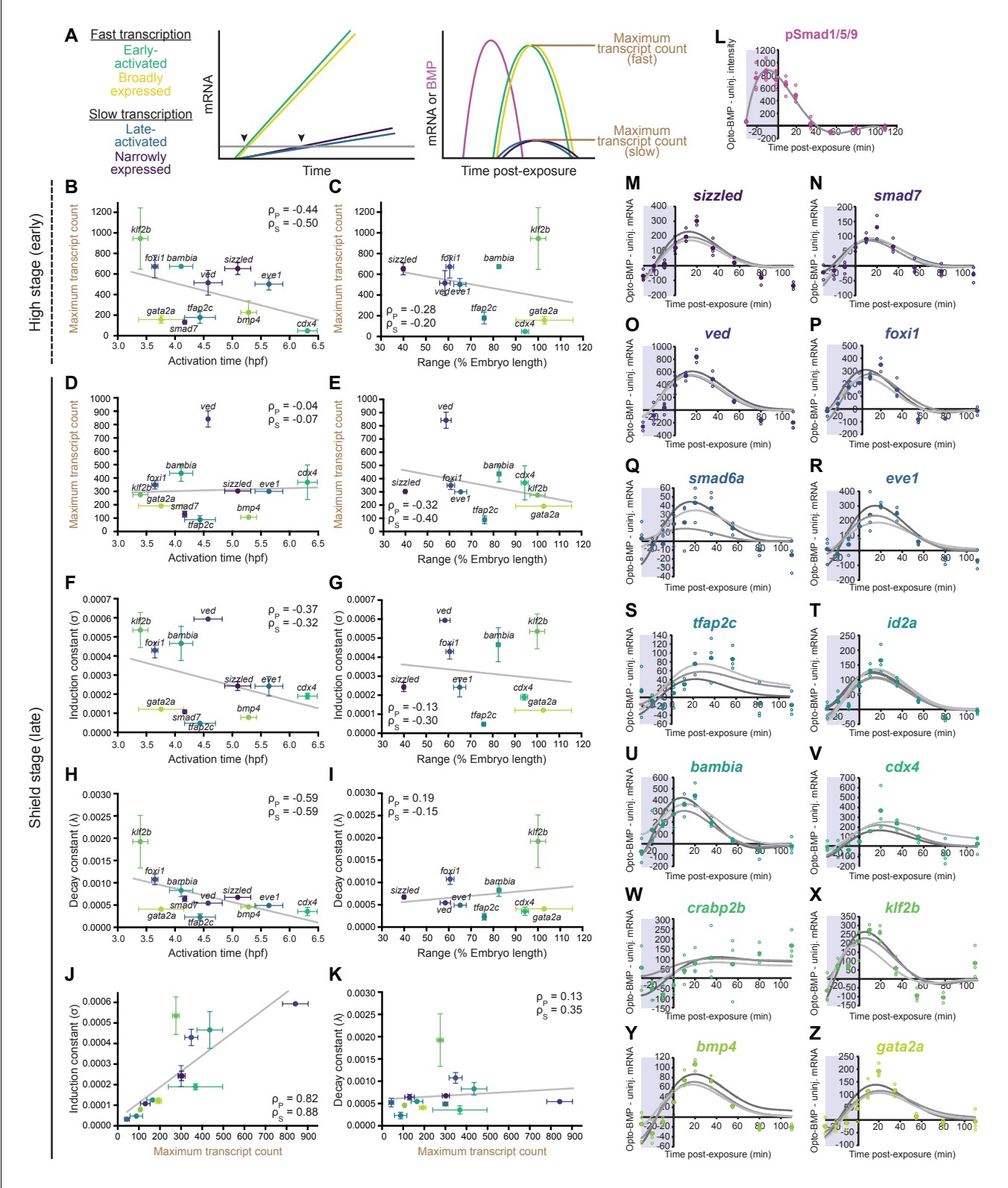

**Figure 5.** Differential expression kinetics do not fully explain BMP target gene spatiotemporal diversity. (**A**) Different transcription kinetics may lead to differences in apparent activation times (arrowheads) based on assay detection thresholds (gray line). Assuming similar degradation kinetics, transcripts with faster induction rates should accumulate to higher levels in response to BMP. (**B-E**) Uninjected and Opto-BMP-injected embryos were exposed to blue light for 30 min at high (~3.5 hpf, B,C) or shield stage (~6.75 hpf, D,E), and target gene expression in response to the resulting BMP signaling

*Figure 5 continued on next page*

*Figure 5 continued*

pulses (*Figure 3C–E* and *Figure 3—figure supplement 1I,J*) was quantified using NanoString technology (*Figure 4*). Maximum average transcript counts were determined, and are plotted against activation time (B,D) (*Figure 2*) or spatial range (C,E). Error bars represent standard error, gray lines represent linear fits, $\rho_s$ = Spearman correlation coefficient, $\rho_P$ = Pearson correlation coefficient. *crabp2b* is not included due to lack of significant induction. (F-L) All three target gene response repeats were fitted with a model of induction and decay (Materials and methods). The average induction constant ($\sigma$) is plotted against activation time (F), spatial range (G), or maximum transcript count (J). The average decay constant ($\lambda$) is plotted against activation time (H), range (I), or maximum transcript count (K). Error bars represent standard error, $\rho_s$ = Spearman correlation coefficient, $\rho_P$ = Pearson correlation coefficient. *crabp2b* is not included due to lack of significant induction. pSmad1/5/9 immunofluorescence (*Figure 3*) was fitted with a polynomial (gray line, L) and used as signaling input. (M-Z) Individual fits of transcriptional responses (*Figure 4*); closed circles represent averages of three data points, open circles represent individual data points, and gray lines represent individual fits of each repeat. See the *Figure 5—source data 1* source data file for source data.

The online version of this article includes the following source data and figure supplement(s) for figure 5:

Source data 1. Source data for *Figure 5*.
Figure supplement 1. Opto-BMP-induced BMP target gene responses and alternative fitting method.
Figure supplement 1—source data 1. Source data for *Figure 5—figure supplement 1*.

We then used a second approach to assess transcript accumulation kinetics that does not require the assumption of similar transcript degradation rates (*Figure 5F–Z*). We fitted the transcription data from the shield-stage BMP signaling pulse with a model involving the known pSmad1/5/9 input (*Figure 3C*, *Figure 5L*, and *Figure 3—figure supplement 1J*) and parameters reflecting transcript induction ($\sigma$) and decay ($\lambda$) (*Figure 5M–Z*, Materials and methods). Each of the three experimental repeats was fitted individually, and average $\sigma$ and $\lambda$ values were calculated for each gene. We found a weak negative correlation between $\sigma$ and activation time (*Figure 5F*), and no correlation between $\sigma$ and range (*Figure 5G*). We also observed a weak negative correlation between $\lambda$ and activation time (*Figure 5H*), and no obvious correlation between $\lambda$ and spatial broadness (*Figure 5I*). These results are consistent with the maximum transcript count analysis (*Figure 5B–E*) and with an alternative fitting approach (*Figure 5—figure supplement 1G–W*, Materials and methods). In addition, we observed a strong positive correlation between maximum transcript count and $\sigma$ (*Figure 5J*), and no correlation between maximum transcript count and $\lambda$ (*Figure 5K*), suggesting that production dominates transcription kinetics, and supporting the use of maximum transcript count as a proxy for induction rate.

Together, our analyses indicate that differential transcription kinetics in response to BMP signaling play a minor role in generating the distinct spatiotemporal expression patterns of BMP target genes.

## Differential activation thresholds do not fully explain spatiotemporal expression

In the gradient threshold paradigm, target genes are activated by distinct signaling thresholds that define gene expression ranges (*Figure 1A*). This model therefore predicts that broadly expressed genes, but not narrowly expressed genes, should be activated by low levels of signaling (*Figure 6A*).

To test this idea, we exposed uninjected and Opto-BMP-injected embryos to high- (3900 lux) or low-intensity (70 lux) blue light for 10 or 20 min at shield stage – resulting in high- or low-amplitude BMP signaling pulses, respectively (*Figure 6B*) – and then quantified BMP target gene responses using NanoString technology. As expected, target activation was generally stronger following higher amplitude, longer duration pulses (*Figure 6C–F* and *Figure 6—figure supplement 1*). However, after a 10 min low-amplitude exposure, the third most narrowly expressed gene, *foxi1*, was significantly activated, whereas the broader genes were not robustly induced (*Figure 6C*). A longer 20 min low-amplitude pulse significantly activated both narrowly and broadly expressed genes (*Figure 6E*). A 10 min low-amplitude pulse significantly activated two of the top 50% earliest expressed genes (*foxi1* and *smad7*), whereas a 20 min low-amplitude pulse significantly activated both early and late-expressed genes (*Figure 6D,F*). High-amplitude pulses activated genes of all ranges and activation times (*Figure 6C–F*).

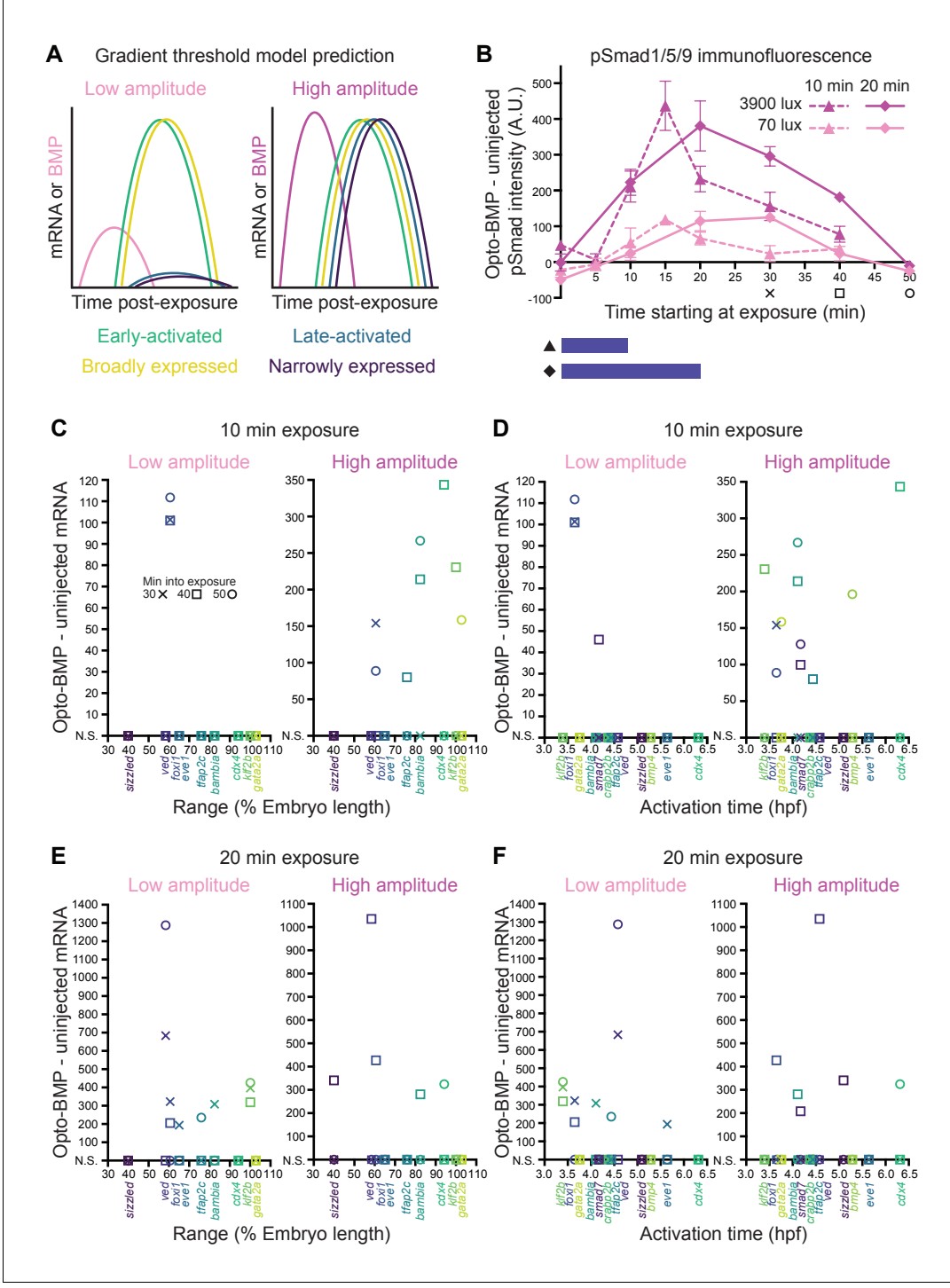

**Figure 6.** Differential sensitivity to BMP does not fully explain target gene expression diversity. (**A**) The activation threshold model predicts that broadly expressed genes will be activated by lower amplitude signaling. (**B**) pSmad1/5/9 immunofluorescence in uninjected and Opto-BMP-injected embryos exposed to 10 (triangle, dashed line) or 20 (diamond, solid line) min of 70 (light pink) or 3900 (magenta) lux blue light starting at shield stage. Immunofluorescence was quantified and plotted as Opto-BMP signal - uninjected with piecewise linear interpolation between timepoints; error bars represent standard error (see Materials and methods for statistical analysis). Embryos for the transcriptional response experiment were collected 30 (x), 40 (square), or 50 (circle) min after the start of light exposure. (**C-F**) Transcriptional responses in Opto-BMP embryos exposed to conditions shown in (B) were quantified using NanoString technology and are plotted against spatial range (C,E) or activation

*Figure 6 continued on next page*

*Figure 6 continued*

time (D,F). Embryos were collected 30 (x), 40 (square), or 50 (circle) min after the start of light exposure. Responses that are not statistically significant are anchored to the x-axis (N.S.; see Materials and methods for statistical analysis). See the *Figure 6—source data 1* file for source data.

The online version of this article includes the following source data and figure supplement(s) for figure 6:

**Source data 1.** Source data for *Figure 6*.

**Figure supplement 1.** Responses to different amplitudes and durations of BMP signaling.

Our experiments exposing embryos to different amplitude BMP signaling pulses therefore suggest that not all spatiotemporal target gene expression differences are due to different signaling activation thresholds, although a subset may be (see Discussion).

## FGF and Nodal modify BMP signaling and target gene expression

We noted that BMP target genes have unique expression patterns along the animal-vegetal axis that contribute to differences in their dorsal-ventral expression profiles (*Figure 1F–Y*). Specifically, 6 out of the 10 spatially quantified high-confidence BMP target genes are either restricted to (*cdx4*, *eve1*) or excluded from (*foxi1*, *klf2b*, *gata2a*, *tfap2c*) the margin. We wondered how regulation by additional signaling pathways active at the margin might contribute to these differences. We focused on the FGF and Nodal pathways, which regulate mesoderm and mesendoderm specification, respectively, and are known to influence BMP signaling (*Figure 7*; *Rogers and Müller, 2019*).

To assess the effects of FGF and Nodal signaling on BMP target gene expression, we inhibited these pathways using the small molecule inhibitors SU-5402 (*Mohammadi et al., 1997*) and SB-505124 (*DaCosta Byfield et al., 2004*), respectively (*Figure 7—figure supplement 1A–J'*). At shield stage, Nodal inhibition did not observably affect BMP signaling (*Figure 7C* and *Figure 7— figure supplement 1F–G'*), whereas FGF inhibition increased the amplitude of the BMP signaling gradient (*Figure 7B* and *Figure 7—figure supplement 1D–E'*). Simultaneous inhibition of both FGF and Nodal signaling increased both the amplitude and spatial broadness of the BMP signaling gradient (*Figure 7D*, *Figure 7—figure supplement 1H–J'*, and *Figure 7—figure supplement 2A–K*). Consistent with enhanced BMP signaling, in the absence of FGF/Nodal several BMP-activated genes were upregulated (*Figure 7—figure supplement 2L–Y*). Reduced levels of the secreted BMP inhibitor Chordin in embryos lacking FGF/Nodal signaling (*Figure 7—figure supplement 2ZA*) are likely to contribute to this BMP signaling expansion (*Varga et al., 2007*; *Londin et al., 2005*; *Koshida et al., 2002*). Additionally, FGF restricts the expression of *bmp* (*Londin et al., 2005*; *Fürthauer et al., 2004*; *Fürthauer et al., 1997*), and we detected increased *bmp2b* expression in FGF/Nodal-inhibited embryos (*Figure 7—figure supplement 2Z*).

Loss of FGF, Nodal, or both simultaneously affected BMP target gene dorsal-ventral spatial expression profiles differently (*Figure 7F–I*, *Figure 7—figure supplement 1*, and *Figure 7—figure supplement 3*). To determine whether FGF and Nodal are responsible for the margin restriction or exclusion of some BMP target genes (*Figure 1F–O'*), we assessed target expression along the animal-vegetal axis in inhibitor-treated embryos. In embryos lacking both FGF and Nodal signaling, margin-restricted genes were still expressed and restricted to the margin, whereas the expression of margin-excluded genes shifted into the margin (*Figure 7I,L,M*, and *Figure 7—figure supplement 1*).

We reasoned that the shift of margin-excluded genes into the margin could either be due to loss of FGF/Nodal activity, or due to enhanced BMP signaling at the margin (*Figure 7D*, *Figure 7—figure supplement 1A–J'*, and *Figure 7—figure supplement 2A–K*). We therefore assessed the animal-vegetal expression of margin-excluded genes in *bmp*-overexpressing embryos, which have dramatically elevated levels of BMP signaling at the ventral margin (*Figure 7E*) but intact Nodal and FGF signaling (*Figure 7—figure supplement 1A–C'*; *Fürthauer et al., 1997*). Margin-excluded genes were still clearly excluded from the margin in *bmp*-overexpressing embryos, suggesting that direct inhibition by FGF and Nodal normally prevents expression of these genes at the margin (*Figure 7J,L,M*).

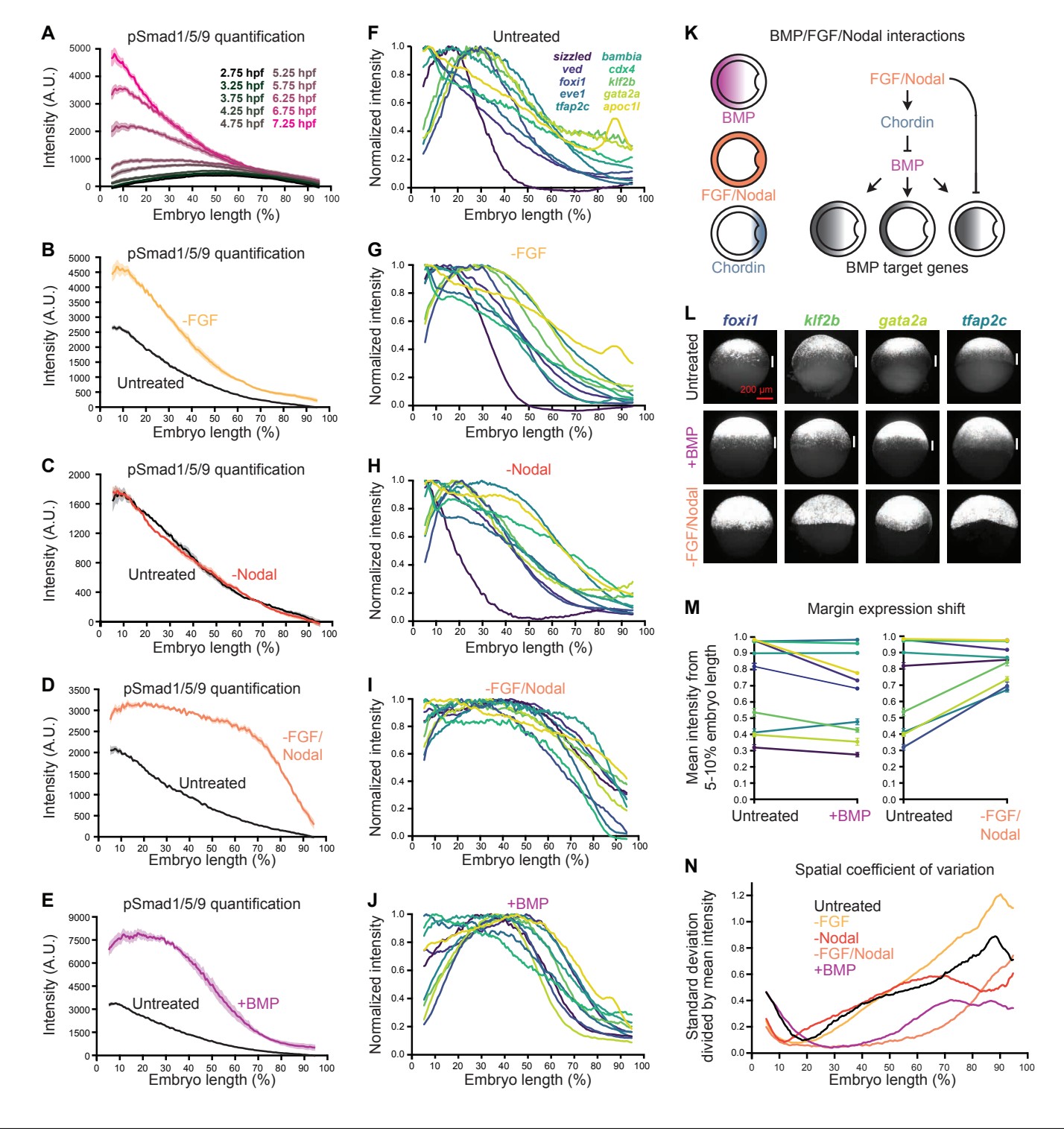

**Figure 7.** FGF and Nodal contribute to the spatial diversity of BMP target genes. (**A**) pSmad1/5/9 immunofluorescence in untreated embryos at the indicated times (data also shown in *Figure 1E*). (**B-E**) Embryos were treated with 10 μM FGF inhibitor SU-5402 (**B**), 50 μM Nodal inhibitor SB-505124 (**C**), or both (**D**) starting at 2 hpf, or injected with 0.5 pg *bmp2b* mRNA at the one-cell stage (**E**). The BMP signaling gradient was quantified along the dorsal-ventral axis at shield stage using pSmad1/5/9 immunofluorescence; error bars represent standard error. Note that the embryos in panels B-E came from different experiments and were processed and imaged on different days, but untreated controls were always siblings of treated embryos and processed and imaged simultaneously. (**F-J**) BMP target gene expression along the dorsal-ventral axis at shield stage in untreated (**F**), SU-5402-

*Figure 7 continued on next page*

*Figure 7 continued*

treated (**G**), SB-505124-treated (**H**), SB-505124 + SU-5402-treated (**I**), and *bmp*-overexpressing (**J**) embryos quantified using fluorescence *in situ* hybridization (untreated data from *Figure 1*). (**K**) FGF and Nodal block expression of a subset of BMP target genes at the margin, and restrict BMP signaling in part by activating the BMP inhibitor Chordin. (**L**) Ventral views of margin-excluded BMP target gene expression at shield stage assessed by FISH in untreated embryos (top row), *bmp*-overexpressing embryos (0.5 pg *bmp2b* mRNA, middle row), and embryos treated with SU-5402 + SB-505124 (bottom row). Vertical white bars indicate regions where expression is excluded from the margin. (**M**) Expression levels at the margin quantified by calculating the average normalized intensity from 5–10% embryo length in untreated versus *bmp*-overexpressing embryos (left) or untreated versus SU-5402+SB-505124-treated embryos (right). Lines connect treated and untreated conditions to visualize shifts, error bars represent standard error. Lower numbers indicate less expression at the margin. (**N**) Spatial coefficient of variation for the 10 BMP target genes assessed here in untreated (black), SU-5402-treated (yellow), SB-505124-treated (red), SU-5402+SB-505124-treated (salmon), and *bmp*-overexpressing (magenta) embryos. Lower numbers indicate less spatial diversity. See the *Figure 7—source data 1* file for source data.

The online version of this article includes the following source data and figure supplement(s) for figure 7:

**Source data 1.** Source data for *Figure 7*.
**Figure supplement 1.** Signaling and BMP target gene expression in *bmp*-overexpressing and FGF/Nodal-inhibited embryos.
**Figure supplement 1—source data 1.** Source data for *Figure 7—figure supplement 1*.
**Figure supplement 2.** BMP signaling and temporal target gene expression in the absence of FGF/Nodal signaling.
**Figure supplement 2—source data 1.** Source data for *Figure 7—figure supplement 2*.
**Figure supplement 3.** Individual fluorescence *in situ* hybridization profiles in *bmp*-overexpressing and FGF/Nodal-inhibited embryos.
**Figure supplement 3—source data 1.** Source data for *Figure 7—figure supplement 3*.

To determine whether FGF and Nodal contribute to diversity in BMP target gene activation timing, we quantified the temporal expression of BMP targets in embryos lacking FGF and Nodal signaling from 2.75 to 7.25 hpf (256-cell stage – 60% epiboly) (*Figure 7—figure supplement 2L–Y*). Although transcript levels of several BMP targets were higher in treated compared to untreated embryos at later stages, their activation times were still diverse, suggesting that inputs other than FGF and Nodal are responsible for differences in activation times.

Finally, we noticed that much of the spatial diversity in BMP target gene expression along the dorsal-ventral axis collapsed in embryos lacking both FGF and Nodal signaling (*Figure 7F,I*). To quantify the decrease in spatial diversity, we calculated the spatial coefficient of variation in untreated and treated embryos (see Materials and methods). Strikingly, embryos lacking both FGF and Nodal had lower coefficients of variation at almost all positions along the dorsal-ventral axis compared to untreated embryos (*Figure 7N*). Together, our results identify combinatorial FGF and Nodal signaling as a major driver of spatial diversity in BMP target gene expression.

## Discussion

### Minor roles for differential responses to BMP in generating spatiotemporal diversity

Signaling gradients are frequently observed in developing tissues, including the embryonic axes of gastrulating zebrafish, the neural tube in mice, and the wing precursor in *Drosophila* (*Briscoe and Small, 2015*; *Schier and Talbot, 2005*). However, how gradients are interpreted by cells is complex to ascertain. The gradient threshold model proposes that gene-specific activation thresholds are responsible for differences in the spatial expression of target genes (*Sharpe, 2019*; *Briscoe and Small, 2015*; *Dubrulle et al., 2015*; *Rogers and Schier, 2011*; *Barkai and Shilo, 2009*; *Ashe and Briscoe, 2006*). Can gradients be reliably generated and signaling thresholds accurately interpreted with high sensitivity, or do gradients simply provide a 'rough framework' for patterning that is refined over time by other mechanisms such as target gene cross-talk (*Briscoe and Small, 2015*; *Chen et al., 2012*) or cell sorting (*Akieda et al., 2019*; *Xiong et al., 2013*)? In the former case, is such precision actually required for patterning?

In the context of zebrafish dorsal-ventral patterning, our data suggest minor roles for gene-specific activation thresholds in generating BMP target gene expression diversity. We did not find a clear monotonically decreasing relationship between activation time and gene expression range

(*Figure 2O*), suggesting that more broadly expressed genes are not consistently more likely to be activated by the low levels of BMP present early (*Figure 1D–E*). We were also unable to detect an unambiguous correlation between range and the levels of signaling required for activation (*Figure 6C,E*). This suggests that not all BMP target expression boundaries are positioned by gene-specific BMP signaling thresholds (*Figure 1A*).

An alternative model proposes that diversity in spatiotemporal target gene expression is due to differences in expression kinetics. For example, it was shown that Nodal targets with higher transcript accumulation rates in response to Nodal signaling have broader spatial expression domains (*Dubrulle et al., 2015*). To determine whether the BMP patterning system might function similarly, we examined the transcriptional responses of BMP target genes (*Figures 4* and *5*, and *Figure 5—figure supplement 1*) to optogenetically generated pulses of BMP signaling (*Figure 3C–D* and *Figure 3—figure supplement 1I,J*). We did not detect a strong correlation between transcript induction rates and activation time or spatial range (*Figure 5* and *Figure 5—figure supplement 1G–W*). Therefore, differential transcription kinetics in response to BMP are unlikely to account for spatiotemporal expression diversity.

Our results do not rule out the possibility that a different subset of BMP target genes may behave more consistently with these models. We focused on a set of high-confidence BMP targets (*Figure 1—figure supplement 1*), but other known targets were excluded from our analyses (*Supplementary file 1*). For example, the BMP target gene *tp63* (*Bakkers et al., 2002*) is not expressed at shield stage, and was therefore excluded since it was not downregulated by *chordin* overexpression in our RNA-sequencing experiment (*Supplementary file 1*). We note that a subset of three genes (*sizzled*, *ved*, and *bambia*) that are neither restricted to nor excluded from the margin do show a monotonically decreasing relationship between range and activation time (*Figure 2O*) as well as activation dynamics that could be roughly commensurate with signaling input (*Figure 6C* and *Figure 6—figure supplement 1*), consistent with the gradient threshold model. However, it remains to be determined to what extent this subset of genes (or others) quantitatively follows the input-output relationships predicted by the gradient threshold model.

Our results also do not rule out other mechanisms of BMP signaling interpretation. For example, the graded distribution of many genes (*Figure 1P–Y*) could be consistent with a model in which gene expression is roughly proportional to the level of BMP signaling. In addition, BMP signaling duration may encode specific responses *in vivo*. Future work is needed to better define the relationship between BMP signaling levels and gene expression and to determine how BMP signaling dynamics are interpreted in embryos. Our study highlights the promise of optogenetic approaches in such investigations (*Rogers and Müller, 2020*). In contrast to pharmacological or genetic methods, optogenetic strategies can provide fast, tunable, and reversible spatiotemporal manipulation of signaling *in vivo* (*Figure 3*, *Figure 6B*, and *Figure 3—figure supplement 1E–H*), allowing more thorough characterization of input/output relationships.

In addition, our observations indicate that BMP signaling precision may not be required for proper patterning, or that the system is robustly buffered. For example, most embryos experiencing transient activation of BMP signaling lack gross morphological defects (*Figure 3C–E*, *Figure 4*, *Figure 3—figure supplement 1A,K,L*, and *Figure 5—figure supplement 1*). How patterning recovers from such insults will be an interesting avenue for future study. Together with previous work (reviewed in *Zinski et al., 2018*), several of our observations indicate that feedback is an important feature of the BMP patterning system: Five out of 16 high-confidence BMP target genes affect BMP signaling (*Figure 1—figure supplement 1*), and embryos can experience a dip in signaling levels after a signaling pulse (*Figure 3C* and *Figure 3—figure supplement 1J*). Cell sorting strategies that sharpen gene expression boundaries may also contribute to the observed recovery from BMP signaling manipulation (*Akieda et al., 2019*; *Xiong et al., 2013*).

## Margin restriction and competence of BMP target genes

One unresolved question from our study is the restriction of the BMP target genes *eve1* and *cdx4* to the margin (*Figure 1I,I',L,L'* and *Figure 7—figure supplement 1*). Consistent with previous work (*Swanhart et al., 2010*; *Ota et al., 2009*; *Bennett et al., 2007*; *Ho et al., 2006*; *Londin et al., 2005*; *Shimizu et al., 2005*; *Rentzsch et al., 2004*), in the absence of FGF or Nodal, *eve1* and *cdx4* were still expressed at the ventral margin (*Figure 7—figure supplement 1*; we note, however, conflicting reports with dominant-negative FGF receptors [*Ota et al., 2009*; *Kudoh et al., 2004*;

*Griffin et al., 1995*]). Inhibition by animal pole factors or a requirement for signaling pathways at the margin such as Wnt or retinoic acid might play a role in their margin restriction.

Both *eve1* and *cdx4* are also activated relatively late in development (*Figure 2F,J*), and *cdx4* is not competent to respond to an early BMP signaling pulse (*Figure 4K*). FGF and Nodal have no obvious roles in regulating their activation timing or competence since their temporal expression was not significantly affected by loss of FGF/Nodal signaling (*Figure 7—figure supplement 2Q,U*). Understanding how the activation timing of all BMP target genes including *eve1* and *cdx4* is regulated is an important future goal.

## FGF and Nodal are major contributors to BMP target gene spatial diversity

Inhibition of FGF, Nodal, or both together had distinct effects on BMP signaling (*Figure 7B–D*, *Figure 7—figure supplement 1A–J'*, and *Figure 7—figure supplement 2A–K*). The increase in BMP signaling in the absence of FGF is likely explained by several factors including the known role of FGF in activating *chordin* and inhibiting *bmp* transcription (*Figure 7—figure supplement 2Z*,ZA) (*Varga et al., 2007*; *Maegawa et al., 2006*; *Londin et al., 2005*; *Fürthauer et al., 2004*; *Kudoh et al., 2004*; *Koshida et al., 2002*; *Fürthauer et al., 1997*), as well as inactivating Smad1 (*Sapkota et al., 2007*; *Pera et al., 2003*; *Kretzschmar et al., 1997*). Loss of Nodal did not detectably alter BMP signaling at shield stage. This is surprising because early expression of *fgf* is thought to depend on Nodal (*van Boxtel et al., 2015*; *Maegawa et al., 2006*; *Mathieu et al., 2004*; *Gritsman et al., 1999*; *Rodaway et al., 1999*), although low levels of *fgf3* appear to be present at late blastula stages in Nodal signaling mutants (*Mathieu et al., 2004*), and weak FGF activity is detectable in Nodal inhibitor-treated embryos (*van Boxtel et al., 2015*). Nodal can activate *chordin* expression independently of FGF (*Varga et al., 2007*), and *chordin* is detectable albeit reduced in Nodal signaling mutants (*Gritsman et al., 1999*), suggesting that the reduction in *chordin* caused by Nodal loss is not sufficient to affect BMP signaling during early gastrulation. Future work is needed to explain why FGF, but not Nodal loss enhances BMP signaling at early gastrulation, and why simultaneous loss increases not only the amplitude but the broadness of the BMP signaling gradient.

Inhibition of FGF, Nodal, or both together also had distinct effects on BMP target gene expression (*Figure 7F–I*, *Figure 7—figure supplements 1*, *2* and *3*). Although Nodal loss did not detectably alter the BMP signaling gradient (*Figure 7C* and *Figure 7—figure supplement 1F–G'*), the spatial distributions of several BMP target genes were affected (*Figure 7H*, *Figure 7—figure supplement 1N,R*, and *Figure 7—figure supplement 3C*). Nodal is also responsible for the dorsal expression of the BMP target gene *apoc1l* (*Figure 1O,O',Y*), which is lost in the absence of Nodal (*Figure 7H*, *Figure 7—figure supplement 1N*, and *Figure 7—figure supplement 3C*). Although our study defines individual target gene responses at the phenomenological level, uncovering the DNA-level mechanisms (e.g., promoter regulation and chromatin status) that lead to the observed responses to BMP, FGF, and Nodal is an important future challenge.

The margin exclusion of the BMP target genes *foxi1*, *klf2b*, *gata2a*, and *tfap2c* can be explained by FGF/Nodal-mediated inhibition (*Figure 7K*). Loss of either FGF or Nodal signaling shifted the expression of margin-excluded genes toward the margin, although the shifts were most dramatic in the absence of both (*Figure 7F–I,L,M*, *Figure 7—figure supplement 1*, and *Figure 7—figure supplement 3*), with the exception of *tfap2c*, which was completely margin-shifted in FGF-inhibited embryos (*Figure 7G*, *Figure 7—figure supplement 1M,Q*, and *Figure 7—figure supplement 3B*). Excess BMP signaling at the margin in embryos lacking FGF and Nodal (*Figure 7D*, *Figure 7—figure supplement 1H–J'*, and *Figure 7—figure supplement 2A–K*) does not explain the observed gene expression shifts because no shifts were evident in *bmp*-overexpressing embryos (*Figure 7J,L,M*, *Figure 7—figure supplement 1L,P*, and *Figure 7—figure supplement 3A*). The FGF/Nodal-mediated margin exclusion of a subset of BMP targets contributes to the diversity in BMP target gene expression (*Figure 7F,I,K,N*), creating distinct dorsal-ventral profiles for margin-excluded genes (*Figure 1R,T,W,X*) compared to non-excluded genes (*Figure 1P,Q,S,U,V,Y*).

Our results suggest that much of the spatial diversity in BMP target gene expression arises from combinatorial signaling. A similar strategy is thought to regulate Bicoid target genes during *Drosophila* embryogenesis: Gene expression boundary shifts in response to Bicoid manipulation are often inconsistent with the gradient threshold model (*Chen et al., 2012*; *Ochoa-Espinosa et al., 2009*), and activation thresholds do not appear to explain target gene expression profiles at the

DNA level (*Ochoa-Espinosa et al., 2005*). Rather, Bicoid is thought to act within a system of repressive pathways that regulate Bicoid target gene expression (*Chen et al., 2012*). During zebrafish dorsal-ventral patterning, FGF and Nodal affect BMP target gene expression in two ways: by restricting BMP signaling (*Figure 7B–D*, *Figure 7—figure supplement 1D–E',H–J'*, and *Figure 7—figure supplement 2A–K*), and by inhibiting a subset of BMP target genes at the margin (*Figure 7F–I,L,M*, *Figure 7—figure supplement 1*, and *Figure 7—figure supplement 3*). These interactions sculpt the spatial expression profiles of BMP target genes and contribute to the patterning of the dorsal-ventral axis.

# Materials and methods

## Key resources table

| Reagent type (species) or resource | Designation | Source or reference | Identifiers | Additional information |
| --- | --- | --- | --- | --- |
| Strain, strain background (*E. coli*) | One Shot TOP10 | Life Technologies | C4040 | Chemically competent |
| Strain, strain background (*Danio rerio*) | TE zebrafish | *Pomreinke et al., 2017* *Donovan et al., 2017* | | Wild type |
| Antibody | anti-phospho-Smad1/Smad5/Smad9 (Rabbit monoclonal) | Cell Signaling Technology | 13820, RRID:AB_2493181 | IF (1:100) |
| Antibody | anti-rabbit Alexa Fluor 488-conjugated secondary (Goat polyclonal) | Life Technologies | A11008, RRID:AB_143165 | IF (1:5000) |
| Antibody | anti-phospho-Smad2/Smad3 (Rabbit monoclonal) | Cell Signaling Technology | 8828, RRID:AB_2631089 | IF (1:5000) |
| Antibody | anti-rabbit horseradish peroxidase (Goat polyclonal) | Jackson ImmunoResearch | 111-035-003, RRID:AB_2313567 | IF (1:500) |
| Antibody | anti-pErk (Mouse monoclonal) | Sigma | M8159, RRID:AB_477245 | IF (1:5000) |
| Antibody | anti-mouse horseradish peroxidase (Donkey polyclonal) | Jackson ImmunoResearch | 715-035-150, RRID:AB_2340770 | IF (1:500) |
| Antibody | anti-digoxigenin horseradish peroxidase Fab fragments (Sheep polyclonal) | Roche | 11207733910, RRID:AB_514500 | FISH (1:150) |
| Recombinant DNA reagent | pCS2-Opto-Alk3 | Generated in this study | | |
| Recombinant DNA reagent | pCS2-Opto-Alk8 | Generated in this study | | |
| Recombinant DNA reagent | pCS2-Opto-BMPR2a | Generated in this study | | |
| Recombinant DNA reagent | pCS2-Opto-BMPR2b | Generated in this study | | |
| Chemical compound, drug | TRIzol reagent | Invitrogen | 5596026 | |
| Chemical compound, drug | Co-Precipitant Pink | Bioline | BIO-37075 | |
| Chemical compound, drug | Cycloheximide | Sigma | C4859 | |
| Chemical compound, drug | Pronase | Roche | 11459643001 | |
| Chemical compound, drug | DMSO | Roth | A994.2 | |

*Continued on next page*

*Continued*

| Reagent type (species) or resource | Designation | Source or reference | Identifiers | Additional information |
|---|---|---|---|---|
| Chemical compound, drug | FBS | Biochrom | S0415 | |
| Chemical compound, drug | DAPI | Life Technologies | D1306 | 1:5000 |
| Chemical compound, drug | Blocking reagent | Roche | 11096176001 | |
| Chemical compound, drug | Low melting temperature agarose | Lonza | 50080 | |
| Chemical compound, drug | Nodal inhibitor SB-505124 | Sigma | S4696-5MG | 50 µM |
| Chemical compound, drug | FGF inhibitor SU-5402 | Sigma | SML0443-5MG | 10 µM |
| Commercial assay or kit | TSA plus cyanine three system | Perkin Elmer | NEL744001KT | FISH/IF (1:75) |
| Commercial assay or kit | RNeasy kit | QIAGEN | 74104 | |
| Commercial assay or kit | Wizard SV Gel and PCR Clean-up System | Promega | A9282 | |
| Commercial assay or kit | pCR-bluntII TOPO kit | Thermo Fisher Scientific | 450245 | |
| Commercial assay or kit | SP6 mMessage mMachine transcription kit | Thermo Fisher Scientific | AM1340 | |
| Commercial assay or kit | DIG RNA labeling mix | Sigma-Aldrich | 11277073910 | |
| Software, algorithm | Fiji | *Schindelin et al., 2012* | https://fiji.sc/ RRID:SCR_002285 | |
| Software, algorithm | Prism | GraphPad Software | https://www.graphpad.com/scientific-software/prism RRID:SCR_002798 | |
| Software, algorithm | COMSOL Multiphysics 3.5a | COMSOL, Inc | https://www.comsol.com/ RRID:SCR_014767 | |
| Software, algorithm | Matlab | Mathworks | http://mathworks.com RRID:SCR_001622 | |
| Software, algorithm | edgeR 3.2.3 | *Robinson et al., 2010* | RRID:SCR_012802 | |
| Software, algorithm | DESeq 1.12.0 | *Anders and Huber, 2010* | RRID:SCR_000154 | |
| Software, algorithm | Cuff diff 2.1.1 | *Trapnell et al., 2010* | https://github.com/cole-trapnell-lab/cufflinks | |
| Software, algorithm | PWM code for controlling LED array | Generated in this study | | |
| Software, algorithm | nSolver 4.0 software | NanoString | RRID:SCR_003420 | |
| Software, algorithm | Excel | Microsoft | RRID:SCR_016137 | |
| Software, algorithm | Maple | Waterloo Maple Inc | RRID:SCR_014449 | |
| Other | RNA-sequencing data | Generated in this study | GEO: GSE135100 | |
| Other | TIP122 complementary power NPN Darlington | STMicroelectronics | | |
| Other | Regulated power supply | Disrelec Group AG | RND 320-KD3000D | |
| Other | 6-well plates | Greiner Bio-One | 657160 | |

*Continued on next page*

*Continued*

| Reagent type (species) or resource | Designation | Source or reference | Identifiers | Additional information |
|---|---|---|---|---|
| Other | Blue LEDs | Nichia | NSPB510AS | |
| Other | Blue LEDs | Everlight | 1363-2SUBC/C470/S400-A4 | |
| Other | Temperature-controlled incubator, Heratherm IMC 18 | ThermoScientific | 50125882 | |
| Other | Raspberry Pi model B | Raspberry Pi Foundation | | |
| Other | LM37 luxmeter | DOSTMANN electronic GmbH | | |
| Other | White worklight | REV Ritter GmbH | 90910 | |
| Other | Red color filters | Rosco | E106 Primary Red | |

## Zebrafish husbandry

Zebrafish husbandry was executed in accordance with the guidelines of the State of Baden-Württemberg (Germany) and approved by the Regierungspräsidium Tübingen (35/9185.46–5, 35/9185.81–5). Wild type TE adult zebrafish were maintained under standard conditions. Embryos were incubated at 28°C in embryo medium (250 mg/l Instant Ocean salt, 1 mg/l methylene blue in reverse osmosis water adjusted to pH 7 with $NaHCO_3$ [*Müller et al., 2012*]) unless otherwise noted.

## mRNA *in vitro* synthesis

*pCS2+*-based plasmids encoding Bmp2b, Chordin (*Pomreinke et al., 2017*), and Opto-BMP (this work, *Figure 3—figure supplement 1*, see below for cloning details) were linearized with NotI-HF (NEB, R3189). Capped mRNA was generated using a mMessage mMachine SP6 kit (ThermoFisher, AM1340). mRNA was purified using an RNeasy Mini kit (Qiagen, 74104) and quantified using a NanoDrop spectrophotometer (ThermoScientific).

## RNA-sequencing

Wild type TE zebrafish embryos were dechorionated with Pronase (Roche, 11459643001) and injected at the one-cell stage with 10 pg mRNA encoding zebrafish Bmp2b, 100 pg mRNA encoding zebrafish Chordin, or left uninjected (*Pomreinke et al., 2017*). When uninjected siblings reached shield stage (~6.75 hpf), embryos were snap-frozen in liquid nitrogen. 10 embryos were collected per sample, three samples per condition.

To prepare total RNA, the TRIzol reagent (Invitrogen, 15596026) manufacturer's protocol was followed until aqueous phase recovery, then 6.25 µl Co-Precipitant Pink (Bioline, BIO-37075) was added to 250 µl aqueous phase, followed by 375 µl 100% EtOH. After vortexing briefly, samples were transferred to RNeasy Mini kit (Qiagen, 74104) spin columns and centrifuged at 13600 rpm at 4°C for 1 min. Flow-through was discarded and columns were washed twice with RPE buffer (Qiagen). RNA was eluted in 50 µl $H_2O$. Total RNA concentration was measured using a NanoDrop spectrophotometer (ThermoScientific). 3–5 µg total RNA per sample were provided to LCG Genomics GmbH (Berlin, Germany) for sequencing and differential expression analysis. Sequences were aligned against the reference genome *Danio rerio* GRCz10 with STAR 2.4.1b, and differential expression analysis was carried out with edgeR 3.2.3, DESeq 1.12.0, and Cuff diff 2.1.1. The p-value threshold for differentially expressed genes was set to 0.05.

Note that endogenous *bmp2b* and *chordin* were not distinguishable from injected mRNAs in *bmp2b*- or *chordin*-injected embryos, respectively, and were therefore excluded from consideration as BMP target genes.

## Opto-BMP constructs

Opto-BMP constructs are based on Opto-Acvr constructs (*Sako et al., 2016*). These *pCS2+*-based Opto-Acvr constructs encode proteins that are tethered to the plasma membrane by an N-terminal myristoylation motif. Next to the membrane is a Nodal receptor kinase domain, followed by the light-oxygen-voltage (VfLOV) domain Aureochrome1 from *Vaucheria frigida* (*Takahashi et al., 2007*),

and finally a C-terminal HA tag. Using splicing by overlap extension (SOE) PCR (*Horton et al., 2013*), Nodal receptor kinase domains in Opto-Acvr were swapped with putative kinase domains from the type I zebrafish BMP receptors Alk3 (NM_131621, bp 691–1566) (*Nikaido et al., 1999*) and Alk8 (NM_131345, bp 622–1497) (*Mintzer et al., 2001*; *Yelick et al., 1998*), and the type II zebrafish receptors BMPR2a (NM_001039817, bp 571–3009) and BMPR2b (NM_001039807, bp 598–1536) (*Monteiro et al., 2008*). In all cases except for Opto-BMPR2a, all residues after the transmembrane domain until the end of the kinase domain were included. Opto-BMPR2a contains all residues after the transmembrane domain until the end of the protein; the kinase domain-only construct was inactive.

An equimolar combination of mRNA encoding Opto-Alk3 (5.2 pg), Opto-Alk8 (5.2 pg), and Opto-BMPR2a (8.9 pg) was found to optimally induce BMP signaling in the light but not in the dark (*Figure 3B*, *Figure 3—figure supplement 1A,K,L*), and was used in all Opto-BMP experiments described here.

## LED array

To facilitate optogenetic experiments requiring control of light intensity and exposure duration, an embedded system-based controller was developed (*Figure 3—figure supplement 1B–D*). To maximize the versatility of the setup for different applications, a single-board computer was deployed (Raspberry Pi 3 model B, running under a Linux kernel, version 4.9). The controller was programmed to generate signals that modulate the duration and intensity of light. The generated signal was further amplified to drive the load of the LED array. A two-stage Darlington amplifier was used (TIP122 complementary power NPN Darlington - STMicroelectronics) to raise the ceiling of the current of amplification. The Darlington pair was used in a common emitter configuration in order to achieve a large power gain. The loads were operated on a constant voltage source provided by a regulated power supply (Disrelec Group AG, RND 320-KD3000D). During initial trials, brief, weak signal spikes could be detected, and an RC filter was subsequently used across the load to dampen any sporadic light flashes. The LED array constituted the circuit load; these LEDs were glued into the plastic cover of 6-well plates (Greiner Bio-One, 657160) (*Figure 3—figure supplement 1B*). Blue Nichia (NSPB510AS) or Everlight (1363-2SUBC/C470/S400-A4) LEDs were used in the array. Both LEDs emitted maximal spectral intensity at 470 nm, with the Nichia LEDs having a broader radiation angle, tighter spectral distribution, and less variable performance. During experiments, the LED array was placed inside a temperature-controlled incubator (Thermo Scientific Heratherum IMC 18, 50125882) set to 28°C. Dark fabric was taped to the interior of the incubator door to prevent outside light from entering.

The circuit schematic (*Figure 3—figure supplement 1D*) shows how the generated square wave was used to drive the LED array. One of the Raspberry Pi's GPIO pins was used as a pulse-width modulation (PWM) output to produce signal. The raspberry-gpio-python module (https://sourceforge.net/projects/raspberry-gpio-python) was used to interface the GPIO. A pulse program was written in Python, which allows for variable parameter settings: GPIO pin number, modulation frequency (10 kHz is the NPN Darlington amplifier linear limit), pulse duration, and duty cycle.

Light intensities were measured using an LM37 luxmeter (DOSTMANN electronic GmbH).

LED array settings used in optogenetic experiments:

| Fig. | Experiment | LED | Voltage (V) | Frequency (Hz) | Intensity (lux) | Duration (min) |
|---|---|---|---|---|---|---|
| 3, 3.1I,J | Shield stage | Everlight | 24 | 200 | 2300 | 30 |
| | High stage | Everlight | 25–28 | 2 | 2300 | 30 |
| 4, 5, 5.1 G-W | Shield stage | Everlight | 24 | 200 | 2300 | 30 |
| | High stage | Everlight | 25–28 | 2 | 2300 | 30 |
| 6, 6.1 | 70 lux | Nichia | 15 | 200 | 70 | 10 or 20 |
| | 3900 lux | Nichia | 21 | 200 | 3900 | 10 or 20 |

*Continued on next page*

*Continued*

| Fig. | Experiment | LED | Voltage (V) | Frequency (Hz) | Intensity (lux) | Duration (min) |
|------|-----------|-----|-------------|----------------|-----------------|----------------|
| 1.1D | *bambia, klf2b, sizzled, smad6a, smad7, ved* | Everlight | 25–28 | 2 | 2300 | 30 |
| | *apoc1l, bmp4, cdx4, crabp2b, eve1, foxi1, gata2a, id2a, tfap2c, znfl2b* | Nichia | 21 | 200 | 3900 | 30 |
| 3.1K,L | Shield stage | Everlight | 24 | 200 | 2300 | 30 |
| | All except shield | Everlight | 25–28 | 2 | 2300 | 30 or 600 |
| 5.1A-F | All | Everlight | 24 | 200 | 2300 | 30 |

For all experiments above, the duty cycle was 100%, and the GPIO pin was 32.

A white worklight (REV Ritter GmbH, 90910) was used in experiments described in *Figure 3—figure supplement 1A*. For all exposure conditions described in this work, no phototoxicity was evident.

PWM code for controlling the LED array (sqr_pls_v01.py):

```python
#!/usr/bin/python
import sys
import time
import getopt
import RPi.GPIO as GPIO

def usage():
    hlp_str = """Basic square pulse programme

input:
-p output BOARD pin number <int>
-f PWM frequency in Hz <int>
-d duty cycle (in percentage terms) <int>
-t length of the pulse in seconds <float>

example usage:
./sqr_pls_v01.py -p 32 -f 200 -d 100 -t 50.0
"""
    print(hlp_str)

def init_out_chan(pin_num, mod_frq):

    """###################
# initiate output #
#————————————#################################################
# input:
# - output PWM pin number (12, 32 or 33) <int>
# - PWM frequency in Hz <int>
# output:
# - pin object
# BOARD numbering mode
# only BOARD channels 12, 32 and 33 are PWModulable
################################################################
    """

    GPIO.setmode(GPIO.BOARD)
```

```
            GPIO.setup(pin_num, GPIO.OUT)
            pin = GPIO.PWM(pin_num, mod_frq)

            return pin

    def sqr_pls(pin_num, mod_frq, dc, span):

        """####################
    # generate output #
    #——————————————####################################################
    # input:
    # - output PWM pin number (12, 32 or 33) <int>
    # - PWM frequency in Hz <int>
    # - duty cycle (in percentage terms) <int>
    # - length of the pulse in seconds <float>
    # output:
    # - 0: completion; 1: interruption <int>
    ##################################################################
        """

        p = init_out_chan(pin_num, mod_frq)
        t_strt = time.time()
        p.start(dc)
        p.ChangeDutyCycle(dc)
        try:
            while (time.time() - t_strt) < span:
                time.sleep(1)
                print("seconds  remaining:  "  +  str(round(span  -  (time.time()-
    t_strt))))
        except KeyboardInterrupt:
            p.stop()
            GPIO.cleanup()
            return 1
        p.stop()
        GPIO.cleanup()

        return 0

    def main():

        try:
            opts, args = getopt.getopt(sys.argv[1:],"p:f:d:t:")
        except getopt.GetoptError as e:
            print(str(e))
            usage()
            sys.exit(2)

        for o, a in opts:
            if o == '-p':
                pin_num=int(a)
                if pin_num not in [12, 32, 33]:
                    print("-USAGE ERROR\n-PIN NUMBER UNACCEPTABLE\n")
                    usage()
                    sys.exit(2)
            elif o == '-f':
                mod_frq=int(a)
```

```
            if (mod_frq > 10000) or (mod_frq < 0):
                print("-USAGE ERROR\n-MODULATION FREQUENCY VALUE OUTSIDE 0-10000
RANGE\n")
                usage()
                sys.exit(2)
        elif o == '-d':
            dc=int(a)
            if (dc > 100) or (dc < 0):
                print("-USAGE ERROR\n-DUTY CYCLE VALUE OUTSIDE 0-100 RANGE\n")
                usage()
                sys.exit(2)
        elif o == '-t':
            t=float(a)
            if (t < 0):
                print("-USAGE ERROR\n-PASSING NEGATIVE TIME")
                usage()
                sys.exit(2)
    #sqr_pls(pin_num, frq, dc, t)
    try:
        print("-commencing square pulse at pin %d modulated at %d Hz at %d%% power
for %.3f seconds" % (pin_num, mod_frq, dc, t))
        print("-starting at %s" % time.ctime())
    except Exception as e:
        print(str(e))
        print("-MISSING ARGUMENT(S) - REVISE USAGE")
        usage()
        sys.exit(2)

    exec_val = sqr_pls(pin_num, mod_frq, dc, t)
    if exec_val:
        print("-Terminating\n-SEQUENCE INTERRUPTED at %s" % time.ctime())
    else:
        print("-Terminating\n-sequence completed at %s" % time.ctime())
    return exec_val

if __name__ == "__main__":
    main()
```

To guard against inadvertent photoactivation, plates containing embryos were wrapped in aluminum foil starting from ~70 min post-injection until light exposure. Where applicable (e.g. *Figure 1—figure supplement 1* and *Figure 3—figure supplement 1*), red color filters (Rosco, E106 Primary Red) were used to cover light sources such as dissecting microscope stages to prevent transmission of VfLOV-dimerizing wavelengths.

## Cycloheximide experiment

For the cycloheximide (Sigma, C4859) experiment in *Figure 1—figure supplement 1D*, embryos from wild type TE incrosses were dechorionated using Pronase (Roche, 11459643001) and injected at the one-cell stage with 5.2 pg *opto-Alk3* + 5.2 *pg opto-Alk8* + 8.9 pg *opto-BMPR2a* mRNA (*Figure 3—figure supplement 1A*). Control siblings were left uninjected, and embryos were sorted into agarose-coated 6-well plates and incubated at 28℃. 70–90 min post-fertilization at the 4–16 cell stage, unfertilized and damaged embryos were removed, and plates were individually wrapped in aluminum foil to prevent light exposure and incubated at 28℃. At 6.25 h post-fertilization (hpf), embryos were transferred into new agarose-coated 6-well dishes containing either 50 µg/ml cycloheximide (*Bennett et al., 2007*; *Poulain and Lepage, 2002*) or an equivalent volume of DMSO (Roth, A994.2) diluted in embryo medium that had been incubated at 28℃ prior to transfer. Red color filters (Rosco, E106 Primary Red) were used to cover the dissecting microscope light source

during the transfer to prevent transmission of VfLOV-dimerizing wavelengths and minimize BMP activation, and plates were wrapped in aluminum foil after transfer. At 6.75 hpf (~shield stage, 30 min after cycloheximide exposure), plates were transferred to a small 28°C incubator containing the LED array (*Figure 3—figure supplement 1B*) and exposed to blue light for 30 min (6.75–7.25 hpf). 20 min after light exposure, when most BMP target genes are maximally induced (*Figure 4*), embryos were fixed and colorimetric *in situ* hybridization was carried out as described in the *Fluorescence and colorimetric* in situ *hybridization* section below.

## pSmad1/5/9, pSmad2/3, and pErk immunofluorescence staining

For pSmad1/5/9, pSmad2/3, and pErk immunofluorescence staining, embryos were fixed in 4% formaldehyde in PBS at 4°C overnight, then transferred to MeOH and stored at −20°C for at least 2 h. See below and *Figure 1—figure supplement 1E* for imaging and quantification details.

### pSmad1/5/9
Embryos were washed at least three times with PBST (phosphate buffered saline + 0.1% Tween-20), then blocked for at least 1 h at room temperature in blocking buffer (10% FBS (Biochrom, S0415), 1% DMSO, 0.1% Tween-20 in PBS). Embryos were incubated in 1:100 rabbit anti-phosphoSmad1/5/9 antibody (Cell Signaling Technology, 13820) in blocking buffer at 4°C overnight. One wash with blocking buffer followed by 3–5 washes with PBST were carried out at room temperature, then embryos were blocked again with blocking buffer for at least 1 h. Embryos were incubated in 1:5000 goat anti-rabbit Alexa Fluor 488-conjugated secondary antibody (Life Technologies, A11008) in blocking buffer at 4°C overnight. Embryos were then incubated in 1:5000 DAPI (Life Technologies, D1306; stock concentration: 5 mg/ml) in blocking buffer at room temperature for at least 1 h, then washed at least five times with PBST. Stained embryos were wrapped in aluminum foil and stored at 4°C overnight before SPIM imaging.

### pSmad2/3
Embryos were incubated in ice-cold acetone (Roth, 5025.5) for 7 min, then washed at least three times with PBST, blocked for at least 1 h in 10% FBS in PBST and incubated in 1:5000 rabbit anti-pSmad2/3 (Cell Signaling Technology, 8828) in 10% FBS in PBST at 4°C overnight. Embryos were then washed at least five times in PBST, blocked again for at least 1 h in 10% FBS in PBST, and incubated in 1:500 goat anti-rabbit HRP secondary antibody (Jackson ImmunoResearch, 111-035-003) in 10% FBS in PBST at 4°C overnight. Next, embryos were washed at least five times in PBST, then once in TSA 1x amplification buffer (TSA Plus Cyanine 3 System, Perkin Elmer, NEL744001KT). For staining, embryos were incubated in 75 µl 1:75 Cy3-TSA in 1x amplification buffer in the dark at room temperature for 45 min. After washing at least six times with PBST, embryos were incubated in 1:5000 DAPI (Life Technologies, D1306; stock concentration: 5 mg/ml) in PBST at room temperature for at least 1 h, then washed at least four times with PBST. Finally, embryos were wrapped in aluminum foil and stored at 4°C overnight before SPIM imaging.

### pErk
Embryos were washed at least three times with PBST, then transferred to ice-cold acetone for 20 min and washed at least three times with PBST. After blocking in 10% FBS in PBST for at least 1 h, embryos were incubated in 1:5000 mouse anti-pErk antibody (Sigma, M8159) in 10% FBS in PBST at 4°C overnight. Embryos were then washed at least five times in PBST, blocked again for at least 1 h in 10% FBS in PBST, and incubated in 1:500 donkey anti-mouse HRP secondary antibody (Jackson ImmunoResearch, 715-035-150) in 10% FBS in PBST at 4°C overnight. Embryos were washed at least five times with PBST, then once in TSA 1x amplification buffer. Next, embryos were incubated in 75 µl 1:75 Cy3-TSA in 1x amplification buffer in the dark at room temperature for 45 min. After washing at least six times with PBST, embryos were incubated in 1:5000 DAPI (Life Technologies, D1306; stock concentration: 5 mg/ml) in PBST at room temperature for at least 1 h, then washed at least four times with PBST. Stained embryos were wrapped in aluminum foil and stored at 4°C overnight before SPIM imaging.

## Fluorescence and colorimetric *in situ* hybridization

BMP target gene probes were generated by amplifying full or partial coding sequences (CDS) from wild type TE zebrafish cDNA and cloning into *pCS2+* or *pCR-bluntII TOPO* (ThermoFisher, 450245) vectors. Plasmids were linearized with the indicated restriction enzymes, column purified (Promega, A9282), and DIG-labeled probes were generated using the indicated polymerase (Roche, 11175025910).

High-confidence BMP target gene *in situ* hybridization probes:

| Gene | Vector | Sequence | Enzyme | Polymerase |
|---|---|---|---|---|
| *apoc1l* | *pCS2+* | entire CDS | ClaI | T7 |
| *bambia* | *pCR-bluntII TOPO* | partial CDS; bp 47–425 | BamHI | T7 |
| *bmp4* | *pCR-bluntII TOPO* | partial CDS; bp 103–558 | EcoRV | SP6 |
| *cdx4* | *pCR-bluntII TOPO* | partial CDS; bp 132–810 | EcoRV | SP6 |
| *crabp2b* | *pCR-bluntII TOPO* | partial CDS; bp 14–436 | EcoRV | SP6 |
| *eve1* | *pCR-bluntII TOPO* | partial CDS; bp 42–665 | BamHI | T7 |
| *foxi1* | *pCS2+* | entire CDS | ClaI | T7 |
| *gata2a* | *pCR-bluntII TOPO* | partial CDS; bp 40–1141 | SpeI | T7 |
| *id2a* | *pCR-bluntII TOPO* | partial CDS; bp 6–401 | BamHI | T7 |
| *klf2b* | *pCS2+* | entire CDS | ClaI | T7 |
| *smad6a* | *pCR-bluntII TOPO* | partial CDS; bp 8–880 | BamHI | T7 |
| *smad7* | *pCR-bluntII TOPO* | partial CDS; bp 23–1024 | BamHI | T7 |
| *sizzled* | *pCS2+* | entire CDS | ClaI | T7 |
| *tfap2c* | *pCS2+* | entire CDS | ClaI | T7 |
| *ved* | *pCR-bluntII TOPO* | partial CDS; bp 7–825 | EcoRV | SP6 |
| *znfl2b* | *pCR-bluntII TOPO* | partial CDS; bp 25–435 | BamHI | T7 |

Note that the *znfl2b in situ* probe contained 47 SNPs compared to the reference genome (*Danio rerio* GRCz11).

The same DIG-labeled probes were used for both fluorescence (*Figures 1F–Y* and *7F–J*, *Figure 7—figure supplement 1K–O*, and *Figure 7—figure supplement 3*) and colorimetric (*Figure 1—figure supplement 1C–D* and *Figure 5—figure supplement 1A–F*) *in situ* hybridization at a concentration of 1 ng/µl.

Whole-mount colorimetric *in situ* hybridization was carried out as described previously (*Thisse and Thisse, 2008*). Embryos were fixed in 4% formaldehyde in PBS, incubated at 4°C overnight, then transferred to MeOH and stored at −20°C for at least 2 h. Stained embryos were imaged in 2:1 benzyl benzoate:benzyl alcohol with an Axio Zoom.V16 microscope (ZEISS).

For fluorescence *in situ* hybridization (FISH), the same protocol was used until the blocking step, at which point embryos were blocked in FISH blocking buffer (2% blocking reagent (Roche, 11096176001) in 1x maleic acid buffer (100 mM maleic acid, 150 mM NaCl, 180 mM NaOH, 0.1% Tween)) for at least 2 h at room temperature with gentle rocking, then incubated at 4°C overnight in 1:150 anti-DIG-POD (Roche, 11207733910). The following day embryos were washed at least five times with PBST. To develop signal, embryos were incubated in 75 µl 1:75 Cy3-TSA in 1x amplification buffer (TSA Plus Cyanine 3 System, Perkin Elmer, NEL744001KT) for 30 min at room temperature in the dark. Embryos were then washed at least five times with PBST, incubated in 1:5000 DAPI (Life Technologies, D1306; stock concentration: 5 mg/ml) with agitation at room temperature for at least 1 h (or overnight at 4°C), then washed at least five times with PBST. One day after Cy3 incubation, embryos were imaged on a ZEISS Lightsheet Z.1 (see below and *Figure 1—figure supplement 1E* for imaging and quantification details). All FISH embryos shown in *Figure 1F–Y* were fertilized and fixed on the same day.

## SPIM imaging of immunofluorescence staining and fluorescence *in situ* hybridization

Fixed embryos were mounted in 1% low melting temperature agarose (Lonza, 50080) using a glass capillary and imaged with a ZEISS Lightsheet Z.1 selective plane illumination microscope (SPIM). The imaging chamber was filled with water, and filters and light sheets were auto-aligned prior to imaging. For fluorescence *in situ* hybridization (FISH) and pSmad1/5/9 immunofluorescence (IF) experiments, embryos were positioned using the DAPI signal with the animal pole pointing toward the imaging objective to produce animal views; for ventral views, embryos in the correct orientation were rotated 90°. For animal views, 50–90 z-slices with 7 µm between each slice were acquired per embryo, covering the entire blastoderm over a distance of 350–630 µm depending on embryo size. For ventral views, ~70 z-slices with 7 µm between each slice were acquired per embryo, spanning roughly half of the embryo.

For pSmad2/3 and pErk IF, embryos were mounted in the orthogonal orientation compared to pSmad1/5/9 and FISH experiments, and three lateral images were acquired per embryo: one at the brightest region, a second rotated 120°, and a third rotated 240°.

All images were acquired with dual light sheet illumination using a W Plan-Apochromat 20x objective at 0.5x zoom and the imaging conditions described below.

SPIM imaging conditions:

| Experiment | Signal | Fluorophore | Laser wave length (nm) | Laser intensity | Filter | Exposure (ms) |
|---|---|---|---|---|---|---|
| pErk IF | pErk | Cy3 | 561 | 1.5% | BP 575–615 | 100 |
| | Nuclei | DAPI | 405 | 1.5% | BP 420–470 | 100 |
| pSmad2/3 IF | pSmad2/3 | Cy3 | 561 | 1% | BP 575–615 | 100 |
| | Nuclei | DAPI | 405 | 1.1% | BP 420–470 | 100 |
| pSmad1/5/9 IF | pSmad 1/5/9 | Alexa488 | 488 | 2% | BP 505–545 | 200 |
| | Nuclei | DAPI | 405 | 1.3% | BP 420–470 | 200 |
| All FISH except SB-treated | FISH | Cy3 | 561 | 1.5% | BP 575–615 | 100 |
| | Nuclei | DAPI | 405 | 1.5% | BP 420–470 | 100 |
| SB-treated FISH | FISH | Cy3 | 561 | 1% | BP 575–615 | 100 |
| | Nuclei | DAPI | 405 | 1.1% | BP 420–470 | 100 |

Maximum intensity projections were generated using the software ZEN (2014 SP1, black edition) and used for the analyses described below.

## Mathematical modeling of target gene induction and decay kinetics

To estimate induction and decay of transcripts from the NanoString data (*Figure 4*), time-dependent pSmad1/5/9 and transcript changes were modeled mathematically. The change in the amount of endogenous ($P_e$) and optogenetically induced ($P_o$) pSmad1/5/9 levels can be described by the following general differential equations:

$$\frac{dP_e}{dt} = \dot{G}(t)$$

$$\frac{dP_o}{dt} = \dot{H}(t)$$

The observed pSmad1/5/9 levels in uninjected embryos correspond to $G(t)$, whereas the observed pSmad1/5/9 levels ($P_s$) in light-exposed Opto-BMP embryos correspond to the sum of $G(t)$ and $H(t)$. Therefore, the change in the amount of $P_s$ over time can be described by:

$$\frac{dP_s}{dt} = \dot{G}(t) + \dot{H}(t) = \dot{I}(t)$$

Thus, the levels of optogenetically induced pSmad1/5/9 can be calculated by subtracting the

pSmad1/5/9 levels in uninjected embryos from the pSmad1/5/9 levels in light-exposed Opto-BMP embryos:

$$I(t) - G(t) = H(t)$$

Similarly, changes in the endogenous transcript levels ($T_e$) and optogenetically induced transcript levels ($T_o$) over time can be described by the following general differential equations:

$$\frac{dT_e}{dt} = \dot{K}(t)$$

$$\frac{dT_o}{dt} = \dot{L}(t)$$

The observed transcript levels in uninjected embryos correspond to $K(t)$, whereas the observed transcript levels ($T_s$) in light-exposed Opto-BMP embryos correspond to the sum of $K(t)$ and $L(t)$. The change in the amount of $T_s$ over time can therefore be described by:

$$\frac{dT_s}{dt} = \dot{K}(t) + \dot{L}(t) = \dot{M}(t)$$

Thus, the levels of optogenetically induced transcripts can be calculated by subtracting the transcript levels in uninjected embryos from the transcript levels in light-exposed Opto-BMP embryos:

$$M(t) - K(t) = L(t)$$

## Modeling method 1

The NanoString transcription data was first analyzed using the simplest model of induction and decay (*Figure 5*):

$$\frac{dT_o}{dt} = \sigma P_o - \lambda T_o$$

where $P_o$ represents the optogenetically induced pSmad1/5/9 input, $T_o$ the pSmad1/5/9-dependent target gene, $\sigma$ the induction rate constant, and $\lambda$ the decay rate constant of the induced gene. $P_o$ was obtained by fitting the measured pSmad1/5/9 immunofluorescence data $H(t)$ (*Figure 3C*, *Figure 5L*, and *Figure 3—figure supplement 1J*) with a polynomial of degree five using the function *polyfit* in MATLAB 7.10.0 (R2010a). The induction-decay model was simulated in COMSOL Multiphysics 3.5a in a 10 μm domain (representing approximately one cell) with no-flux boundary conditions and an initial concentration $T_o(0)$.

For each experiment, the combination of parameters $T_o(0)$, $\sigma$, and $\lambda$ was found that minimizes the sum of squared differences (SSD)

$$SSD = \sum_n (L(t_n) - T_o(t_n))^2$$

between the simulations of the induction-decay model $T_o(t_n)$ and the data $L(t_n)$ for all measured time points $n$.

The minimization was performed numerically using a constrained optimization algorithm (Nelder-Mead, MATLAB 7.10.0) with zero for the initial guesses of $T_o(0)$, $\sigma$, and $\lambda$, and a maximum of 500 iterations. $\sigma$ and $\lambda$ were constrained between biologically plausible values of 0.00001/s and 0.1/s, and $T_o(0)$ was bounded between −100 a.u. and 100 a.u. $R^2$ values were calculated from the minimizing SSD ($SSD_{min}$) to assess the goodness of the fits by

$$R^2 = 1 - \frac{SSD_{min}}{\sum_n \left(L(t_n) - \frac{1}{n}\sum_n L(t_n)\right)^2}$$

Fitted values for high-confidence BMP target genes, experimental repeat 1:

| Target gene | $\sigma$ (1/s) | $\lambda$ (1/s) | $T_o(0)$ (a.u.) | $R^2$ |
|---|---|---|---|---|
| *bambia* | 0.000414 | 0.000879 | −95.84 | 0.9125 |
| *bmp4* | 0.000076 | 0.000452 | −42.71 | 0.7060 |
| *cdx4* | 0.000220 | 0.000434 | −81.48 | 0.6871 |
| *crabp2b* | 0.000041 | 0.000010 | 10.27 | 0.0951 |
| *eve1* | 0.000233 | 0.000514 | −20.49 | 0.7132 |
| *foxi1* | 0.000371 | 0.000835 | −91.10 | 0.7821 |
| *gata2a* | 0.000105 | 0.000336 | −46.01 | 0.6056 |
| *id2a* | 0.000111 | 0.000539 | −20.54 | 0.7354 |
| *klf2b* | 0.000394 | 0.001262 | 5.369 | 0.6880 |
| *smad6a* | 0.000016 | 0.000670 | 1.606 | 0.1780 |
| *smad7* | 0.000116 | 0.000765 | −65.54 | 0.8043 |
| *sizzled* | 0.000222 | 0.000605 | −82.09 | 0.6470 |
| *tfap2c* | 0.000041 | 0.000153 | −15.90 | 0.1922 |
| *ved* | 0.000590 | 0.000590 | −100.0 | 0.8072 |

Fitted values for high-confidence BMP target genes, experimental repeat 2:

| Target gene | $\sigma$ (1/s) | $\lambda$ (1/s) | $T_o(0)$ (a.u.) | $R^2$ |
|---|---|---|---|---|
| *bambia* | 0.000344 | 0.000564 | 90.73 | 0.6248 |
| *bmp4* | 0.000066 | 0.000474 | −17.10 | 0.6114 |
| *cdx4* | 0.000169 | 0.000170 | −26.84 | 0.2825 |
| *crabp2b* | 0.000056 | 0.000010 | −43.72 | 0.2517 |
| *eve1* | 0.000158 | 0.000394 | 7.77 | 0.6689 |
| *foxi1* | 0.000413 | 0.001217 | −14.67 | 0.7806 |
| *gata2a* | 0.000111 | 0.000399 | −71.03 | 0.5647 |
| *id2a* | 0.000141 | 0.000522 | −35.84 | 0.6967 |
| *klf2b* | 0.000708 | 0.003101 | −62.70 | 0.6394 |
| *smad6a* | 0.000029 | 0.000340 | −6.326 | 0.3140 |
| *smad7* | 0.000112 | 0.000613 | −54.51 | 0.7397 |
| *sizzled* | 0.000217 | 0.000692 | −99.99 | 0.6918 |
| *tfap2c* | 0.000056 | 0.000160 | −27.15 | 0.4203 |
| *ved* | 0.000588 | 0.000554 | −100.0 | 0.7354 |

Fitted values for high-confidence BMP target genes, experimental repeat 3:

| Target gene | $\sigma$ (1/s) | $\lambda$ (1/s) | $T_o(0)$ (a.u.) | $R^2$ |
|---|---|---|---|---|
| *bambia* | 0.000640 | 0.001045 | −99.99 | 0.9362 |
| *bmp4* | 0.000094 | 0.000455 | −73.00 | 0.7986 |
| *cdx4* | 0.000181 | 0.000468 | −99.81 | 0.5390 |
| *crabp2b* | 0.000087 | 0.000010 | −87.63 | 0.4399 |
| *eve1* | 0.000334 | 0.000568 | −100.0 | 0.7789 |
| *foxi1* | 0.000505 | 0.001174 | −6.052 | 0.8502 |
| *gata2a* | 0.000148 | 0.000491 | −65.45 | 0.8464 |
| *id2a* | 0.000126 | 0.000579 | 9.476 | 0.8000 |

*Continued on next page*

*Continued*

| Target gene | $\sigma$ (1/s) | $\lambda$ (1/s) | $T_o(0)$ (a.u.) | $R^2$ |
|---|---|---|---|---|
| klf2b | 0.000505 | 0.001407 | −100.0 | 0.6530 |
| smad6a | 0.000051 | 0.000563 | −25.92 | 0.8935 |
| smad7 | 0.000095 | 0.000553 | −19.75 | 0.5364 |
| sizzled | 0.000290 | 0.000721 | −100.0 | 0.7169 |
| tfap2c | 0.000045 | 0.000378 | −38.42 | 0.4709 |
| ved | 0.000602 | 0.000489 | −100.0 | 0.6802 |

Average fitted values for high-confidence BMP target genes:

| Target gene | $\sigma$ (1/s) | | $\lambda$ (1/s) | | $T_o(0)$ (a.u.) | |
|---|---|---|---|---|---|---|
| | Mean | Stdev | Mean | Stdev | Mean | Stdev |
| bambia | 0.00047 | 0.00015 | 0.00083 | 0.00024 | −35.04 | 108.9 |
| bmp4 | 0.00008 | 0.00001 | 0.00046 | 0.00001 | −44.27 | 27.98 |
| cdx4 | 0.00019 | 0.00003 | 0.00036 | 0.00016 | −69.38 | 37.96 |
| crabp2b | 0.00006 | 0.00002 | 0.00001 | 0.00000 | −40.36 | 49.04 |
| eve1 | 0.00024 | 0.00009 | 0.00049 | 0.00009 | −37.57 | 55.88 |
| foxi1 | 0.00043 | 0.00007 | 0.00108 | 0.00021 | −37.27 | 46.81 |
| gata2a | 0.00012 | 0.00002 | 0.00041 | 0.00008 | −60.83 | 13.13 |
| id2a | 0.00013 | 0.00002 | 0.00055 | 0.00003 | −15.63 | 23.05 |
| klf2b | 0.00054 | 0.00016 | 0.00192 | 0.00102 | −52.44 | 53.43 |
| smad6a | 0.00003 | 0.00002 | 0.00052 | 0.00017 | −10.21 | 14.17 |
| smad7 | 0.00011 | 0.00001 | 0.00064 | 0.00011 | −46.60 | 23.90 |
| sizzled | 0.00024 | 0.00004 | 0.00067 | 0.00006 | −94.03 | 10.34 |
| tfap2c | 0.00005 | 0.00001 | 0.00023 | 0.00013 | −27.15 | 11.26 |
| ved | 0.00059 | 0.00001 | 0.00054 | 0.00005 | −100.0 | 0.000 |

## Modeling method 2

In a second approach (*Figure 5—figure supplement 1G–W*), the NanoString transcription data was fitted with the analytical solutions to the differential equation system

$$\frac{dP_e}{dt} = k_1 - k_2 P_e$$

$$\frac{dP_o}{dt} = k_3(\theta(t) - \theta(t - t_L)) - k_2 P_o$$

$$\frac{dT_e}{dt} = k_4 P_e - k_5 T_e$$

$$\frac{dT_o}{dt} = \sigma P_o - \lambda T_o$$

which describes the changes in endogenous as well as optogenetically induced pSmad1/5/9 and transcript levels based on the simplest model of induction and decay after an optogenetic pulse of length $t_L$ (i.e., 30 min = 1800 s for all experiments). $k_1$ represents the activation rate of endogenous pSmad1/5/9, $k_2$ the decay rate constant of pSmad1/5/9, and $k_3$ the activation rate of optogenetically induced pSmad1/5/9. Optogenetic switch-like activation was modeled with the Heaviside step function $\theta$. $k_4$ and $k_5$ represent the activation rate and decay rate constants of endogenously induced

BMP-dependent transcripts, and $\sigma$ and $\lambda$ are the induction rate and decay rate constants of the induced gene.

The analytical solutions to this equation system are:

$$P_e(t) = e^{-k_2 t}\delta_{P_e} + \frac{k_1}{k_2}$$

$$P_o(t) = \frac{1}{k_2}\big(k_3(\theta(t_L) - \theta(t_L - t))e^{-k_2(t-t_L)} + k_3\theta(t_L - t) + (-k_3\theta(t) - \theta(t_L)k_3 + \delta_{P_o}k_2 + k_3)e^{-k_2 t} + k_3(\theta(t) - 1)\big)$$

$$T_e = \frac{1}{k_2 k_5(k_2 - k_5)}\big(k_2 k_5(\delta_{P_e}k_4 + \delta_{T_e}k_2 - \delta_{T_e}k_5)e^{-k_5 t} + (-k_2 k_5\delta_{P_e}e^{-k_2 t} + k_1(k_2 - k_5))k_4\big)$$

$$T_o = \frac{1}{(k_2 - \lambda)k_2\lambda}\Big(-\sigma k_3\lambda(\theta(t_L) - \theta(t_L - t))e^{-k_2(t - t_L)} + \sigma k_2 k_3(\theta(t_L) - \theta(t_L - t))e^{-\lambda(t - t_L)} + \sigma k_3(k_2 - \lambda)\theta(t_L - t) - k_2(\theta(t_L))k_3\sigma + \sigma k_3\theta(t) + (-\delta_{P_o}\lambda - k_3)\sigma - \lambda\delta_{T_o}(k_2 - \lambda))e^{-\lambda t} + \sigma\big(\lambda(\theta(t_L)k_3 + k_3\theta(t) - \delta_{P_o}k_2 - k_3)e^{-k_2 t} + k_3(\theta(t) - 1)(k_2 - \lambda)\big)\Big)$$

with

$$P_e(0) = \delta_{P_e} + \frac{k_1}{k_2}$$

$$P_o(0) = \delta_{P_o}$$

$$T_e(0) = \delta_{T_e} + \frac{k_1 k_4}{k_2 k_5}$$

$$T_o(0) = \delta_{T_o}$$

The pSmad1/5/9 data was fitted with the computer algebra system Maple (Waterloo Maple Inc) using the function *LSSolve* to minimize the difference between the pSmad1/5/9 data in uninjected embryos and $P_e(t)$, as well as the difference between the pSmad1/5/9 data in light-exposed Opto-BMP embryos and $P_e(t) + P_o(t)$ with the initial guesses $\delta_{P_e} = 0$ a.u., $\delta_{P_o} = 0$ a.u., $k_1 = 0/s$, $k_2 = 0.00167/s$, $k_3 = 0/s$, $k_4 = 00167/s$ and a maximum of 20000 iterations and an optimality tolerance of $0.3981071706 \times 10^{-14}$. The best fitting parameters $\delta_{P_e} = -76.19$ a.u, $\delta_{P_o} = 264.1$ a.u., $k_1 = 0.1429$ a.u./s, $k_2 = 0.000900/s$, and $k_3 = 0.954$ a.u./s were then used for the simulation of the gene induction dynamics in the NanoString data.

The NanoString data was fitted in Maple using the function *LSSolve* to simultaneously minimize the difference between the NanoString data in uninjected embryos and $T_e(t)$, as well as the difference between the NanoString data in light-exposed Opto-BMP embryos and $T_e(t) + T_o(t)$ with the initial guesses $\delta_{T_e} = 0$ a.u., $\delta_{T_o} = 0$ a.u., $k_4 = 0/s$, $k_5 = 0.00167/s$, $\sigma = 0/s$, $\lambda = 0.00167/s$ and a maximum of 10000 iterations and an optimality tolerance of $0.3981071706 \times 10^{-14}$.

Fitted values for high-confidence BMP target genes:

| Target gene | $\sigma$ (1/s) | $\lambda$ (1/s) | $\delta_{T_e}$ (a.u.) | $\delta_{T_o}$ (a.u.) | $R^2$ |
|---|---|---|---|---|---|
| *bambia* | 0.000327 | 0.000671 | 520.4 | 16.01 | 0.7509 |
| *bmp4* | 0.000070 | 0.000362 | 171.2 | −41.52 | 0.7912 |
| *cdx4* | 0.000238 | 0.000550 | −1331 | −29.22 | 0.8682 |
| *crabp2b* | 0.000048 | −0.000191 | −310.9 | −40.27 | 0.8111 |
| *eve1* | 0.000177 | 0.000313 | 950.7 | −52.74 | 0.8599 |

*Continued on next page*

*Continued*

| Target gene | $\sigma$ (1/s) | $\lambda$ (1/s) | $\delta_{T_e}$ (a.u.) | $\delta_{T_o}$ (a.u.) | $R^2$ |
|---|---|---|---|---|---|
| *foxi1* | 0.000401 | 0.000094 | 47.34 | −72.26 | 0.5592 |
| *gata2a* | 0.000144 | 0.000427 | 62.39 | −77.72 | 0.4272 |
| *id2a* | 0.000143 | 0.000449 | 231.3 | −21.14 | 0.7586 |
| *klf2b* | 0.000419 | 0.001114 | 9.959 | −97.86 | 0.5082 |
| *smad6a* | 0.000030 | 0.000318 | 49.62 | −15.67 | 0.3307 |
| *smad7* | 0.000125 | 0.000626 | 122.9 | −54.50 | 0.7043 |
| *sizzled* | 0.000274 | 0.000758 | 238.9 | −120.7 | 0.5116 |
| *tfap2c* | 0.000062 | 0.000316 | 44.54 | −31.75 | 0.1597 |
| *ved* | 0.000732 | 0.000531 | 1200 | −403.0 | 0.8063 |

## Inhibition of Nodal and FGF signaling with small molecule inhibitors

The Nodal inhibitor SB-505124 (Sigma, S4696-5MG) (*Soh et al., 2020*; *Almuedo-Castillo et al., 2018*; *Rogers et al., 2017*; *van Boxtel et al., 2015*; *Vogt et al., 2011*; *Fan et al., 2007*; *Hagos and Dougan, 2007*; *Hagos et al., 2007*; *DaCosta Byfield et al., 2004*) and the FGF inhibitor SU-5402 (Sigma SML0443-5MG) (*van Boxtel et al., 2015*; *Poulain et al., 2006*; *Londin et al., 2005*; *Fürthauer et al., 2004*; *Kudoh et al., 2004*; *Mathieu et al., 2004*; *Mohammadi et al., 1997*) were diluted to 10 mM in DMSO (Roth, A994.2), aliquoted, and stored at −20°C. Aliquots were thawed the same day that experiments were carried out and were not re-used. 10 mM stocks of SB-505124 and SU-5402 were diluted to 50 and 10 µM, respectively, in embryo medium the day of each experiment. 5 ml diluted inhibitors were then dispensed into each well of agarose-coated (Sigma, A9539) 6-well plates (Greiner Bio-One, 657160), and plates were incubated at 28°C at least 30 min before embryos were added.

## Quantification of pSmad1/5/9 immunofluorescence staining and fluorescence *in situ* hybridization

To measure spatial intensity profiles along the dorsal-ventral axis (*Figure 1—figure supplement 1E*) from pSmad1/5/9 immunofluorescence experiments (IF) (*Figures 1D–E* and *7A–E*, *Figure 7—figure supplement 1A–J'*, and *Figure 7—figure supplement 2A–K*) and BMP target gene fluorescence *in situ* hybridization (FISH) (*Figure 1P–Y*, *Figure 7F–J*, and *Figure 7—figure supplement 3*), maximum intensity projections of animal views were manually rotated in Fiji (*Schindelin et al., 2012*) with ventral to the left (brightest signal) and dorsal to the right (dimmest signal; for the very early pSmad1/5/9 images prior to clear onset of BMP signaling, embryos were oriented with the brightest side on the left and the dimmest on the right where obvious, but correspondence with ventral-dorsal is not clear in those early cases). A polygonal region of interest (ROI) was then manually drawn around the embryo and used to create a mask in order to remove image background (for FISH experiments, the Cy3 signal was used to draw the mask; for IF experiments the DAPI signal was used). The average pixel intensity in each column of pixels from ventral to dorsal was then acquired (pixel area: 0.46 µm × 0.46 µm). For genes that are restricted to the margin (*cdx4* and *eve1*), a second manually positioned circular ROI was used to exclude the non-margin region of the embryo (*Figure 1—figure supplement 1E*).

For FISH experiments, non-probe-exposed control embryos for background subtraction were imaged and intensity profiles acquired as described above. The orientation of these background subtraction embryos was random. Images for background subtraction controls were acquired in the same imaging session as experimental FISH images.

After intensity profiles were acquired, absolute distance was converted into percent embryo length to account for embryo-to-embryo variability in size, and intensity measurements were averaged into bins of 0.5% embryo length using an automated routine ($0 \leq$ bin 1 < 0.5%, $0.5 \leq$ bin 2 < 1%, etc.).

For FISH experiments, the average intensity at each position in all 10 non-probe-exposed background embryos was calculated. This spatial background average was subtracted from each

experimental FISH raw intensity profile, and data from the first and last 5% embryo length was excluded because the averages at the most ventral and dorsal regions are composed of relatively few pixels and are therefore less reliable.

The profiles of individual embryos were normalized following the procedure in *Gregor et al., 2007* using the model

$$I_n(x) = A_n \bar{c}(x) + b_n$$

which relates the mean intensity profile $\bar{c}(x)$ of all data points for a given target gene to the intensity profile $I_n(x)$ for an embryo $n$ through the embryo-specific proportionality constant $A_n$ and the non-specific background $b_n$. $A_n$ and $b_n$ were determined by minimizing the objective function

$$\sum_i (I_n(x_i) - (A_n \bar{c}(x_i) + b_n))^2$$

for the data points at all positions $x_i$ with the Nelder-Mead algorithm using the function *fminsearch* in MATLAB 7.10.0, the initial guesses 1 and 0 for $A_n$ and $b_n$, a maximum of 10000 function evaluations, and a maximum of 5000 iterations. For display, each average profile was then divided by its maximum intensity (*Figure 1P-Y*, *Figure 7F-J*, and *Figure 7—figure supplement 3*).

The Gaussian function $Ae^{-\frac{(x-\mu)^2}{\varsigma}}$ was fitted to each profile using a constrained Nelder-Mead algorithm in MATLAB 7.10.0 with a maximum of 10000 function evaluations, a maximum of 5000 iterations, the initial guesses 300, 20, and 10000, the lower bounds 300, -50, and 100, and the upper bounds 100000, 50, and 100000 for $A$, $\mu$, and $\varsigma$, respectively. Gene expression range was defined as $r = \mu + 2\sqrt{\varsigma/2}$. The resulting ranges from 9-10 embryos were averaged to define each gene's mean range.

For pSmad1/5/9 IF spatial quantification experiments, the average image background intensity was determined for each image using a small ROI in the corner outside of the embryo, and subtracted from each IF raw intensity profile. Since the averages at the most ventral and dorsal regions are composed of relatively few pixels and are therefore less reliable, data from the first and last 5% embryo length was not considered. The mean of the dorsal-most 5% at 2.75 hpf was then subtracted from all profiles. These profiles were then normalized as described above for the FISH data, assuming embryo-specific constant nonspecific background and proportionality constants that relate immunofluorescent staining intensity to protein concentration.

Number of embryos assessed in spatial quantification experiments:

| Experiment | | Fig. | Number of embryos |
|---|---|---|---|
| FISH | All except *apoc1l* in *bmp*-overexpressing embryos | 1P-Y, 7 F-J, 7.3 | 10 |
| | *apoc1l* in *bmp*-overexpressing embryos | 7J, 7.3A | 9 |
| pSmad 1/5/9 IF | Time course in untreated and SU-5402/SB-505124-treated embryos | 1E,7A, 7.2A-K | 8–9 |
| | Untreated and *bmp*-overexpressing embryos | 7E, 7.1A-A' | 10 |
| | Untreated and SU-5420-treated embryos | 7B, 7.1D-D' | 10 |
| | Untreated and SB-505124-treated embryos | 7C, 7.1 F-F' | 9–10 |
| | Untreated and SU-5402/SB-505124-treated embryos | 7D, 7.1 H-H' | 10 |

To quantify total pSmad1/5/9 IF intensity (*Figures 3C–E* and *6B*, and *Figure 3—figure supplement 1I,J*), an ROI was manually drawn around the embryo in Fiji based on DAPI signal and used to create a mask in order to remove image background as described above. The average intensity within the ROI was then calculated.

For experiments shown in *Figure 3C–D* and *Figure 3—figure supplement 1I*, image background intensity was measured using a small ROI in the corner of each image outside of the embryo. The average image background was then subtracted from the embryo intensity measurements to generate background-subtracted intensities.

For shield-stage experiments shown in *Figure 3C,E*, *Figure 5L*, *Figure 6B*, and *Figure 3—figure supplement 1J*, the average intensity within a small ROI on the dorsal side was measured in

uninjected embryos; for each time point, these values were averaged and subtracted from the embryo intensity measurements to generate background-subtracted intensities.

Number of embryos assessed in total pSmad1/5/9 IF quantification time course experiments:

| Experiment | Fig. | Number of embryos |
| --- | --- | --- |
| High-stage BMP signaling pulse | 3C, 3.1I | 5 |
| Shield-stage BMP signaling pulse | 3C, 5L, 3.1J | 5 |
| Low- and high-amplitude BMP signaling pulse | 6B | 5 uninjected |
| | | 7 Opto-BMP |

## NanoString RNA quantification

For the NanoString time course experiment in untreated (*Figure 2*) and FGF/Nodal-inhibitor-treated embryos (*Figure 7—figure supplement 2L-ZA*), embryos from wild type TE incrosses were collected ~15 min after mating commenced. Embryos were incubated at 28°C, dechorionated using Pronase (Roche, 11459643001) at ~1.5 hpf, and sorted into 10 agarose-coated 6-well plates, one plate per time point. Each plate had one well containing embryo medium and one well containing FGF/Nodal inhibitor. To keep temperature and therefore development steady, plates were only removed from the 28°C incubator immediately prior to embryo collection. Every 30 min from 2.75 to 7.25 hpf, treated and untreated embryos were snap-frozen in liquid nitrogen.

For NanoString experiments quantifying responses to BMP signaling pulses using Opto-BMP (*Figures 4*, *5* and *6*, *Figure 5—figure supplement 1J–W*, and *Figure 6—figure supplement 1*), embryos from TE incrosses were dechorionated using Pronase and injected at the one-cell stage with 5.2 pg *opto-Alk3* + 5.2 pg *opto-Alk8* + 8.9 pg *opto-BMPR2a* mRNA (*Figure 3—figure supplement 1A*). Control siblings were left uninjected, and embryos were sorted into agarose-coated 6-well plates and incubated at 28°C. 70–90 min post-fertilization at the 4–16 cell stage, unfertilized and damaged embryos were removed, and plates were individually wrapped in aluminum foil and incubated at 28°C. At the appropriate time, individual plates were transferred to a small 28°C incubator containing the LED array, exposed to light for the appropriate duration, and embryos were either snap-frozen in liquid nitrogen immediately (e.g., for the 10 min during exposure time point), or rewrapped in aluminum foil and returned to 28°C incubation in the dark (e.g., for the 80 min post-exposure time point).

RNA was prepared as described for the RNA-sequencing experiment. 30 µl aliquots at 20 ng/µl were provided to Proteros GmbH (Planegg-Martinsried, Germany) for analysis using a custom-designed NanoString nCounter Elements TagSet with probes targeting high-confidence BMP target genes identified by RNA-sequencing, and housekeeping genes for normalization. Samples were measured using an nCounter SPRINT according to the standard protocol with a 24–30 h hybridization length.

nSolver 4.0 software (https://www.nanostring.com/products/analysis-software/nsolver) was used to subtract background and normalize the RNA count data using the geometric means of the positive spike-in controls and the housekeeping genes *eef1a1l1* and *act2b*, respectively. Lanes that failed quality control were repeated.

Number of embryos assessed in NanoString experiments:

| Experiment | | Fig. | Number of embryos |
| --- | --- | --- | --- |
| Time course from 2.75 to 7.25 hpf | Untreated embryos | 2, 7.2L-ZA | 25 |
| | SU-5402/SB-505124-treated embryos | 7.2L-ZA | 20–25 |
| High-stage BMP signaling pulse | | 4 | 21–25 |
| Shield-stage BMP signaling pulse | | 4, 5, 5.1I-W | 19–25 |
| Low- and high-amplitude BMP signaling pulse | | 6 C-F, 6.1 | 25 |

Each of the experiments described in the table above was repeated three times.

For experiments in which transcriptional responses to BMP signaling pulses are assessed (*Figures 4*, *5* and *6C–F*, and *Figure 6—figure supplement 1*), it is necessary to determine changes in transcript levels compared to uninduced embryos. Because each of the three sets of Opto-BMP embryos had uninjected control siblings collected at the same time, average induction was calculated by first subtracting the uninjected transcript count from its corresponding injected sibling count, then by averaging the three subtracted counts (also see the section *Mathematical modeling of target gene induction and decay kinetics* above for a formal description of this procedure).

## Calculation of spatial coefficients of variation

The spatial coefficient of variation (*Figure 7N*) for each condition (untreated, *bmp*-overexpressing, +SB-505124, + SU-5402, and +SB-505124 and SU-5402) was calculated as follows: First, at each position *x*, the average normalized intensity

$$\mu(x) = \frac{1}{n}\sum_{i=1}^{n} I_i(x)$$

and standard deviation

$$\sigma(x) = \sqrt{\frac{1}{n-1}\sum_{i=1}^{n}(I_i(x) - \mu(x))^2}$$

for all *n* genes quantified by FISH were determined (*Figure 7F–J*). Next, the standard deviation was divided by the average normalized intensity at that position

$$c_v(x) = \frac{\sigma(x)}{\mu(x)}$$

This was repeated for every position along the dorsal-ventral axis for all five conditions to calculate the spatial coefficients of variation for the 10 measured genes.

## Statistical analyses

In the following experiments, significance was defined as a **p-value$\leq$0.05** using an unpaired two-tailed Student's *t*-test assuming equal variance in Excel.

To determine how light exposure at different developmental stages affects BMP signaling in Opto-BMP embryos, total pSmad1/5/9 immunofluorescence intensity was quantified in uninjected and Opto-BMP-injected embryos exposed to light at high (3.5–4 hpf) or shield (6.75–7.25 hpf) stage (*Figure 3C–D* and *Figure 3—figure supplement 1I,J*).

Early and late light exposure, Opto-BMP versus uninjected p-values (*Figure 3C–D* and *Figure 3—figure supplement 1I,J*):

| Time post-exposure (min) | High stage (early) | Shield stage (late) |
|---|---|---|
| −30 | 0.065 | 0.003 |
| −20 | 0.029 | $1.422 \times 10^{-5}$ |
| −10 | 0.001 | $7.732 \times 10^{-7}$ |
| 0 | $1.189 \times 10^{-5}$ | $1.610 \times 10^{-6}$ |
| 10 | $2.052 \times 10^{-6}$ | $1.181 \times 10^{-5}$ |
| 20 | 0.002 | $6.800 \times 10^{-6}$ |
| 35 | 0.077 | 0.407 |
| 55 | 0.021 | 0.016 |
| 80 | 0.455 | 0.025 |
| 110 | 0.948 | 0.135 |

To determine how different light intensities affect BMP signaling in Opto-BMP embryos, total pSmad1/5/9 immunofluorescence intensity was quantified in uninjected and Opto-BMP-injected embryos exposed to low (70 lux) or high (3900 lux) intensity light for 10 or 20 min (**Figure 6B**).

Low- and high-intensity light, Opto-BMP versus uninjected p-values (**Figure 6B**):

| Time post-exposure (min) | 70 lux, 10 min | 3900 lux, 10 min | 70 lux, 20 min | 3900 lux, 20 min |
|---|---|---|---|---|
| 0 | 0.419 | **0.020** | **0.013** | 0.975 |
| 5 | 0.782 | 0.782 | ND | ND |
| 10 | 0.328 | **0.003** | 0.493 | **0.001** |
| 15 | **0.001** | **0.0004** | ND | ND |
| 20 | 0.097 | **0.0004** | **0.009** | **0.001** |
| 30 | 0.583 | **0.012** | **0.0002** | **0.00003** |
| 40 | 0.059 | **0.018** | 0.367 | $8.656 \times 10^{-7}$ |
| 50 | ND | ND | 0.367 | 0.729 |

To determine whether BMP target gene expression domain boundaries differ significantly in untreated embryos, range was defined in individual embryos as described in the section *Quantification of pSmad1/5/9 immunofluorescence staining and fluorescence* in situ *hybridization*. Ranges were then averaged.

BMP target gene range comparison p-values (**Figure 1P–Y**):

| | bambia | cdx4 | eve1 | foxi1 | gata2a | klf2b | sizzled | tfap2c | ved |
|---|---|---|---|---|---|---|---|---|---|
| apoc1l | 0.0122 | 0.136 | $3.27 \times 10^{-4}$ | $9.47 \times 10^{-5}$ | 0.682 | 0.371 | $1.06 \times 10^{-6}$ | 0.003 | $6.79 \times 10^{-5}$ |
| bambia | | $4.44 \times 10^{-6}$ | $1.06 \times 10^{-5}$ | $1.2 \times 10^{-9}$ | 0.129 | $9.95 \times 10^{-5}$ | $1.12 \times 10^{-18}$ | $2.70 \times 10^{-5}$ | $1.58 \times 10^{-8}$ |
| cdx4 | | | $3.38 \times 10^{-8}$ | $2.70 \times 10^{-11}$ | 0.511 | 0.138 | $3.67 \times 10^{-17}$ | $5.59 \times 10^{-9}$ | $2.15 \times 10^{-10}$ |
| eve1 | | | | 0.182 | 0.010 | $2.64 \times 10^{-7}$ | $5.40 \times 10^{-8}$ | 0.001 | 0.081 |
| foxi1 | | | | | 0.004 | $5.70 \times 10^{-9}$ | $1.66 \times 10^{-9}$ | $2.09 \times 10^{-7}$ | 0.466 |
| gata2a | | | | | | 0.838 | $1.07 \times 10^{-4}$ | 0.049 | 0.003 |
| klf2b | | | | | | | $1.26 \times 10^{-12}$ | $2.01 \times 10^{-6}$ | $8.46 \times 10^{-9}$ |
| sizzled | | | | | | | | $7.92 \times 10^{-18}$ | $5.19 \times 10^{-7}$ |
| tfap2c | | | | | | | | | $1.33 \times 10^{-6}$ |

The shape of the temporal BMP target gene expression profiles assessed by NanoString in untreated and SU-5402/SB-505124-treated embryos can be well approximated by the modified cumulative distribution function of the normal distribution

$$\frac{1}{2}A\left(1+\mathrm{erf}\left(\frac{x-\nu}{\tau\sqrt{2}}\right)\right)+b$$

which was used for regression analysis using a constrained Nelder-Mead algorithm in MATLAB 7.10.0 with a maximum of 10000 function evaluations, a maximum of 5000 iterations, the initial guesses 1000, 5 h, 1 h, and 100, the lower bounds 100, 3 h, 0.05 h, and 0, and the upper bounds 10000, 7 h, 3 h, and 1000 for $A$, $\nu$, $\tau$, and $b$, respectively. The activation time of each BMP target gene was defined as the average time point at which the curves reached about two mean average deviations (i.e., $1.5 \cdot \tau$) from the inflection point $\nu$ (**Figure 2** and **Figure 7—figure supplement 2L-Y**). *id2a* (**Chong et al., 2005**) and *smad6a* (**White et al., 2017**) were excluded from this analysis because they are maternally contributed.

To determine whether FGF/Nodal loss affects the timing of gene activation, activation times in untreated versus SU-5402/SB-505124-treated samples were compared (**Figure 7—figure supplement 2L–Y**).

SU-5402/SB-505124-treated versus untreated activation time p-values (*Figure 7—figure supplement 2L–Y*):

| bambia | bmp4 | cdx4 | crabp 2b | eve1 | foxi1 | gata 2a | klf2b | smad7 | sizzled | tfap2c | ved |
|---|---|---|---|---|---|---|---|---|---|---|---|
| 0.446 | 0.248 | 0.551 | 0.346 | 0.450 | 0.184 | **0.043** | 0.760 | 0.571 | 0.201 | 0.082 | 0.333 |

To identify differences in BMP target gene expression in the absence of FGF/Nodal signaling, transcript counts from SU-5402/SB-505124-treated embryos were compared to counts from untreated embryos (*Figure 7—figure supplement 2L–Y*).

SU-5402/SB-505124-treated versus untreated p-values (*Figure 7—figure supplement 2L–Y*):

| hpf | bambia | bmp4 | cdx4 | crabp 2b | eve1 | foxi1 | gata 2a | id2a | klf2b | smad 6a | smad7 | sizzled | tfap 2c | ved |
|---|---|---|---|---|---|---|---|---|---|---|---|---|---|---|
| 2.75 | 0.796 | 0.677 | 0.770 | 0.389 | 0.835 | 0.654 | 0.675 | 0.961 | 0.652 | 0.578 | 0.826 | 0.824 | 0.897 | 0.968 |
| 3.25 | 0.757 | 0.590 | 0.855 | 0.905 | 0.573 | 0.790 | 0.386 | 0.946 | 0.341 | 0.918 | 0.704 | 0.497 | 0.514 | 0.682 |
| 3.75 | 0.695 | 0.749 | 0.941 | 0.951 | 0.593 | 0.791 | 0.804 | 0.700 | 0.729 | 0.159 | 0.816 | 0.854 | 0.245 | 0.818 |
| 4.25 | 0.565 | 0.954 | 0.650 | 0.434 | 0.561 | 0.661 | 0.590 | 0.720 | 0.855 | 0.785 | 0.358 | 0.258 | 0.521 | 0.751 |
| 4.75 | 0.988 | 0.943 | 0.996 | 0.655 | 0.645 | 0.751 | 0.919 | 0.965 | 0.820 | 0.460 | 0.643 | 0.224 | 0.630 | 0.947 |
| 5.25 | 0.910 | 0.477 | 0.554 | 0.996 | 0.927 | 0.874 | 0.759 | 0.733 | 0.877 | 0.511 | 0.561 | 0.095 | 0.489 | 0.800 |
| 5.75 | 0.877 | 0.323 | 0.622 | 0.405 | 0.237 | 0.324 | 0.083 | 0.589 | 0.108 | 0.739 | 0.615 | 0.122 | 0.319 | 0.926 |
| 6.25 | 0.443 | 0.509 | 0.731 | 0.399 | 0.450 | 0.149 | 0.091 | 0.767 | 0.085 | 0.938 | 0.966 | 0.077 | 0.105 | 0.483 |
| 6.75 | 0.493 | 0.596 | 0.713 | 0.325 | 0.723 | **0.041** | **0.038** | 0.163 | **0.022** | 0.415 | 0.120 | **0.006** | **0.023** | 0.103 |
| 7.25 | 0.346 | 0.078 | 0.657 | 0.262 | 0.874 | **0.021** | **0.014** | 0.256 | **0.011** | 0.067 | 0.055 | **0.008** | **0.019** | **0.020** |

For experiments in which transcriptional responses to BMP signaling pulses at high or shield stage were measured using NanoString (*Figures 4* and *5M–Z*), mRNA counts in Opto-BMP-injected embryos were compared to uninjected embryos.

High-stage BMP signaling pulse, Opto-BMP versus uninjected p-values (*Figure 4*):

| Time post-exposure (min) | bambia | bmp4 | cdx4 | crabp2b | eve1 | foxi1 | gata 2a | id2a | klf2b | smad6a | smad7 | sizzled | tfap2c | ved |
|---|---|---|---|---|---|---|---|---|---|---|---|---|---|---|
| −30 | 0.274 | 0.381 | 0.279 | 0.362 | 0.428 | 0.610 | 0.401 | 0.315 | 0.573 | 0.173 | 0.983 | 0.295 | 0.283 | 0.312 |
| −20 | 0.270 | 0.419 | 0.225 | 0.144 | 0.386 | 0.051 | 0.354 | 0.275 | 0.364 | 0.301 | 0.897 | 0.456 | 0.171 | 0.124 |
| −10 | 0.232 | 0.273 | 0.799 | 0.501 | 0.563 | **0.019** | 0.874 | 0.570 | 0.359 | 0.398 | 0.711 | 0.249 | 0.900 | 0.527 |
| 0 | **0.019** | 0.181 | 0.053 | 0.483 | **0.004** | 0.071 | 0.152 | 0.459 | 0.053 | 0.732 | 0.167 | **0.016** | **0.001** | 0.169 |
| 10 | **0.005** | 0.539 | 0.760 | 0.136 | **0.031** | **0.002** | 0.168 | **0.040** | **0.043** | 0.793 | 0.124 | **0.0001** | **0.012** | **0.017** |
| 20 | **0.002** | **0.034** | 0.190 | 0.902 | **0.002** | 0.083 | **0.021** | **0.001** | 0.067 | 0.156 | **0.007** | **0.001** | **0.019** | 0.060 |
| 35 | **0.002** | **0.002** | 0.458 | 0.726 | **0.006** | **0.0002** | **0.002** | **0.017** | 0.098 | 0.897 | **0.0003** | **0.0004** | **0.0001** | **0.001** |
| 55 | 0.113 | 0.139 | 0.912 | 0.566 | **0.033** | 0.242 | **0.043** | **0.004** | 0.182 | 0.214 | 0.097 | 0.083 | **0.043** | 0.138 |
| 80 | 0.175 | 0.069 | 0.497 | 0.061 | 0.166 | 0.310 | **0.003** | **0.008** | 0.807 | 0.279 | 0.287 | 0.804 | 0.079 | 0.623 |
| 110 | 0.056 | 0.449 | 0.793 | 0.356 | 0.209 | 0.440 | 0.463 | 0.402 | 0.226 | 0.760 | 0.084 | **0.006** | 0.357 | 0.214 |

Shield-stage BMP signaling pulse, Opto-BMP versus uninjected p-values (*Figures 4* and *5M–Z*, and *Figure 5—figure supplement 1J–W*):

| Time post-exposure (min) | bambia | bmp4 | cdx4 | crabp2b | eve1 | foxi1 | gata2a | id2a | klf2b | smad6a | smad7 | sizzled | tfap2c | ved |
|---|---|---|---|---|---|---|---|---|---|---|---|---|---|---|
| −30 | 0.544 | 0.401 | 0.983 | 0.522 | 0.869 | 0.423 | 0.382 | 0.909 | 0.278 | 0.828 | 0.168 | 0.154 | 0.667 | **0.003** |
| −20 | 0.111 | 0.069 | 0.727 | 0.440 | 0.686 | 0.474 | 0.116 | 0.242 | 0.084 | 0.731 | 0.509 | 0.983 | 0.631 | 0.483 |
| −10 | 0.166 | 0.667 | 0.698 | 0.098 | 0.634 | **0.013** | 0.522 | 0.489 | 0.197 | 0.881 | 0.834 | 0.820 | 0.492 | 0.680 |
| 0 | **0.032** | 0.071 | 0.599 | 0.627 | **0.041** | **0.005** | 0.280 | **0.004** | **0.046** | 0.136 | **0.002** | 0.056 | 0.781 | **0.004** |
| 10 | **0.013** | **0.001** | 0.084 | 0.658 | 0.082 | **0.000** | **0.006** | **0.013** | **0.0001** | 0.242 | **0.003** | **0.005** | 0.151 | **0.002** |
| 20 | **0.003** | **0.002** | **0.020** | 0.201 | **0.001** | **0.000** | **0.0001** | **0.012** | **0.007** | **0.045** | **0.006** | **0.004** | **0.006** | **0.002** |
| 35 | **0.006** | **0.012** | **0.031** | 0.254 | 0.088 | **0.015** | **0.005** | **0.002** | 0.222 | 0.067 | **0.013** | 0.075 | **0.034** | **0.020** |
| 55 | 0.415 | **0.001** | 0.139 | 0.408 | 0.158 | **0.011** | 0.460 | **0.024** | **0.038** | **0.044** | 0.766 | 0.563 | 0.051 | 0.162 |
| 80 | 0.237 | 0.067 | 0.726 | 0.231 | 0.101 | 0.067 | 0.695 | 0.089 | 0.067 | 0.336 | 0.374 | **0.031** | 0.710 | **0.011** |
| 110 | 0.673 | **0.009** | 0.828 | **0.050** | 0.079 | 0.568 | 0.783 | 0.410 | 0.094 | 0.222 | 0.214 | **0.005** | 0.537 | **0.049** |

For experiments in which transcriptional responses to low- and high-amplitude BMP signaling pulses of different durations were measured using NanoString (*Figure 6C–F* and *Figure 6—figure supplement 1*), mRNA counts from uninjected embryos were first subtracted from Opto-BMP-injected siblings. Then the subtracted counts from light-exposed embryos were compared to subtracted counts from unexposed control embryos.

Low- and high-amplitude BMP pulses, exposed versus unexposed p-values (*Figure 6C–F*, *Figure 6—figure supplement 1*):

| Exp. | Time into exposure (min) | bambia | bmp4 | cdx4 | crabp2b | eve1 | foxi1 | gata2a | id2a | klf2b | smad6a | smad7 | szl | tfap2c | ved |
|---|---|---|---|---|---|---|---|---|---|---|---|---|---|---|---|
| 70 lux, 10 min | 30 | 0.083 | 0.597 | 0.390 | 0.967 | 0.487 | **0.021** | 0.856 | 0.703 | 0.271 | 0.894 | 0.071 | 0.405 | 0.816 | 0.603 |
| | 40 | 0.945 | 0.917 | 0.247 | 0.928 | 0.467 | **0.020** | 0.700 | 0.436 | 0.586 | 0.263 | **0.045** | 0.309 | 0.230 | 0.291 |
| | 50 | 0.078 | 0.234 | 0.659 | 0.358 | 0.104 | **0.046** | 0.067 | 0.050 | 0.341 | 0.081 | 0.084 | 0.205 | 0.070 | 0.079 |
| 3900 lux, 10 min | 30 | 0.122 | 0.967 | 0.758 | 0.998 | 0.317 | **0.020** | 0.456 | 0.085 | 0.155 | 0.343 | 0.355 | 0.475 | 0.583 | 0.425 |
| | 40 | **0.013** | 0.367 | **0.008** | 0.296 | 0.085 | 0.056 | 0.171 | 0.154 | **0.027** | 0.572 | **0.037** | 0.261 | **0.019** | 0.062 |
| | 50 | 0.029 | 0.013 | 0.169 | 0.805 | 0.517 | **0.030** | **0.011** | **0.015** | 0.163 | 0.332 | **0.005** | 0.051 | 0.190 | 0.206 |
| 70 lux, 20 min | 30 | **0.001** | 0.635 | 0.176 | 0.660 | **0.037** | **0.001** | 0.062 | **0.019** | **0.002** | 0.304 | 0.056 | 0.087 | 0.321 | **0.031** |
| | 40 | 0.120 | 0.348 | 0.217 | 0.126 | 0.479 | **0.011** | 0.172 | 0.104 | **0.031** | 0.270 | 0.181 | 0.136 | 0.102 | 0.250 |
| | 50 | 0.121 | 0.103 | 0.273 | 0.173 | 0.075 | 0.075 | 0.068 | **0.033** | **0.042** | 0.216 | 0.064 | 0.085 | 0.031 | **0.047** |
| 3900 lux, 20 min | 30 | 0.178 | 0.448 | 0.201 | 0.233 | 0.061 | 0.160 | 0.061 | **0.035** | 0.154 | 0.491 | 0.166 | 0.232 | 0.122 | 0.189 |
| | 40 | **0.005** | 0.123 | 0.934 | 0.761 | 0.083 | **0.003** | 0.075 | **0.028** | 0.077 | **0.020** | **0.001** | **0.028** | 0.324 | **0.033** |
| | 50 | 0.324 | 0.271 | **0.006** | 0.615 | 0.077 | 0.540 | 0.144 | 0.382 | 0.062 | 0.929 | 0.238 | 0.160 | 0.225 | 0.237 |

## Data and code availability

The raw images, data, and source code for custom scripts used in this work are available from the corresponding author upon request. Image quantification data and differential gene expression analyses are available in the accompanying source data files and *Supplementary file 1*, respectively. The RNA-sequencing data has been deposited at the GEO repository (accession number: GSE135100).

## Acknowledgements

We are grateful to Keisuke Sako, Harald Janovjak, and Carl-Philipp Heisenberg for kindly sharing Opto-Acvr constructs before publication and for advice on optogenetic experiments. We thank Luis Antoniotti for help constructing the LED array, Jörg Abendroth for providing the photograph of the LED array in *Figure 3—figure supplement 1B*, Jennifer Bergmann for the *sizzled in situ* hybridization probe, and Daniel Čapek, Amit Landge, David Mörsdorf, Autumn Pomreinke, and Hannes Preiß

for helpful discussions. This project has received funding from the European Research Council (ERC) under the European Union's Horizon 2020 research and innovation program (grant agreement No 637840 (*QUANTPATTERN*) and grant agreement No 863952 (*ACE-OF-SPACE*)). This work was also funded by the Max Planck Society and an HFSP Career Development Award (CDA-00031/2013 C).

# Additional information

## Competing interests
Mohammad ElGamacy: is an employee of Heliopolis Biotechnology Ltd. The other authors declare that no competing interests exist.

## Funding

| Funder | Grant reference number | Author |
| --- | --- | --- |
| Max Planck Society | | Patrick Müller |
| Human Frontier Science Program | CDA-00031/2013-C | Patrick Müller |
| European Research Council | 637840 (QUANTPATTERN) | Patrick Müller |
| European Research Council | 863952 (ACE-OF-SPACE) | Patrick Müller |

The funders had no role in study design, data collection and interpretation, or the decision to submit the work for publication.

## Author contributions
Katherine W Rogers, Conceptualization, Resources, Data curation, Formal analysis, Validation, Investigation, Visualization, Methodology, Writing - original draft, Project administration, Writing - review and editing; Mohammad ElGamacy, Resources, Software, Visualization, Methodology, Writing - review and editing; Benjamin M Jordan, Resources, Software, Formal analysis, Methodology, Writing - review and editing; Patrick Müller, Conceptualization, Resources, Data curation, Software, Formal analysis, Supervision, Funding acquisition, Validation, Visualization, Methodology, Project administration, Writing - review and editing

## Author ORCIDs

Katherine W Rogers (iD) https://orcid.org/0000-0001-5700-2662
Patrick Müller (iD) https://orcid.org/0000-0002-0702-6209

## Decision letter and Author response
Decision letter https://doi.org/10.7554/eLife.58641.sa1
Author response https://doi.org/10.7554/eLife.58641.sa2

# Additional files

## Supplementary files
• Supplementary file 1. RNA-sequencing read counts and differential expression analyses. Wild type TE zebrafish embryos were injected at the one-cell stage with 10 pg mRNA encoding zebrafish Bmp2b, 100 pg mRNA encoding zebrafish Chordin, or left uninjected. When uninjected siblings reached shield stage (~6.75 hpf), embryos were snap-frozen in liquid nitrogen. 10 embryos were collected per sample, three samples per condition. Total RNA was sequenced and aligned against the reference genome *Danio rerio* GRCz10 with STAR 2.4.1b. Differential expression analyses comparing BMP-injected versus uninjected ('+BMP vs. uninj.' tabs) and Chordin-injected versus uninjected ('+Chd vs. uninj.' tabs) were carried out with edgeR 3.2.3, DESeq 1.12.0, and Cuff diff 2.1.1. The p-value threshold for differentially expressed genes was set to 0.05. Read counts and DE analyses for all genes ('all' tabs) and for only the significantly differentially expressed genes ('DE' tabs) are available here.

• Transparent reporting form

## Data availability

The RNA-sequencing data has been deposited at the GEO repository (accession number: GSE135100) and can be accessed at https://www.ncbi.nlm.nih.gov/geo/query/acc.cgi?acc=GSE135100. Image quantification data is available in the accompanying source data files.

The following dataset was generated:

| Author(s) | Year | Dataset title | Dataset URL | Database and Identifier |
|---|---|---|---|---|
| Rogers KW, Müller P | 2020 | Identification of BMP-regulated genes in early gastrulation stage zebrafish embryos | http://www.ncbi.nlm.nih.gov/geo/query/acc.cgi?acc=GSE135100 | NCBI Gene Expression Omnibus, GSE135100 |

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
