## [Decision Letter]

Thank you for submitting your article "Combinatorial signaling underlies spatial diversity in BMP target gene expression" for consideration by *eLife*. Your article has been reviewed by three peer reviewers, one of whom is a member of our Board of Reviewing Editors, and the evaluation has been overseen by Naama Barkai as the Senior Editor. The reviewers have opted to remain anonymous.

The reviewers have discussed the reviews with one another and the Reviewing Editor has drafted this decision to help you prepare a revised submission.

1) Modify title and Abstract. The novel aspect is in the readout along the DV axis, only little data is presented for the readout along the animal-vegetative axis, and the results are confirming a model for spatial readout that has been made in numerous other model systems over the last decades

2) Please carefully respond to the points raised by reviewer 1 and 2.

Revisions expected in follow-up work:

Additional experiments are not necessarily requested, but could be later uploaded if the authors feel this adds to their work.

Reviewer #1:

The manuscript submitted by Patrick Müller and colleagues deals with the topic of morphogen signalling, in particular with the readout of different morphogen concentrations. Using optogenetic manipulation of BMP signalling in the early zebrafish embryo, the authors come to the conclusion that BMP-dependent gene regulation is not well explained or supported by a differential response to BMP doses, but instead results from combinatorial signalling. The paper is elegant and original in its experimental approaches (optogenetics, mathematical modelling, excellent assay system in vivo).

The authors look mostly at temporal regulation of genes with different expression patterns along the D/V axis, assuming that more broadly expressed genes should be activated by lower signalling levels, and since signalling levels increase, these genes should also be activated earlier. Since they do not see such a temporal behaviour, they conclude that the absence of a compelling correlation is inconsistent with the gradient threshold model. Using optogenetic approaches, they show that the differences in the competence to respond to BMP signalling at early stages cannot explain the majority of diversity in activation timing.

The major conclusion, namely that combinatorial signalling underlies the spatial diversity in BMP target gene expression, relies mostly on the gene expression patterns along the animal-vegetal axis. However, and as stated by the authors, the morphogen readout of BMP is best described and discussed in the literature with regard to the dorsal-ventral axis. I find it misleading that the major conclusions are derived from an axis along which BMP is not known to act as a morphogen. Different expression pattern in different "cell types" have long been known to arise via combinatorial signalling (Cell papers in 2000; and then many more in the last twenty years).

While the role of FGF and Nodal signaling is interesting, the description of the results obtained takes up only one page of the Results, and does not really contribute much to the overall topic of spatiotemporal expression along the DV axis. Yet, at the end, the results from this short aspect of the paper is placed into the title of the manuscript. Again, as I said, I find this misleading and the message conveyed in the title does not reflect the question addressed in the study.

So, all of the elegantly obtained data does not answer the initially posed question, and possibly leaves the reader a bit perplexed. Maybe it would be better to reformulate title and Abstract such that the experiments and the message are better aligned.

How is the different extend of expression along the gradient achieved? What is their idea?

What about different receptors and/or different ligands (homodimers versus heterodimers) and the expression pattern along the DV axis?

I am not an expert on mathematical modelling, so this part should be assessed by another reviewer who is more closely to this field.

Reviewer #2:

In this manuscript, Rogers et al. studied the read-out of the dorsal-ventral BMP gradient in zebrafish embryos. The authors first use RNA-seq to identify 16 BMP target genes and measured their spatial and temporal expression pattern in zebrafish embryos. They further induce pulses of BMP signaling and evaluate the differences in the gene expression pattern of these genes. By comparing their experimental observations with the predictions from the “gradient threshold” model, the authors conclude that their observations are inconsistent with a model based on threshold of gene activation. The authors then perturb the Nodal and/or FGF signaling pathway and show that these pathways significantly affect the expression pattern of these BMP targets, suggesting that the spatial expression of BMP targets is more dependent on the combination of BMP, Nodal and FGF signaling than on the threshold activation of BMP signal.

The question how gradients are read out is highly relevant and the experimental approach appears state-of-the art (though I am not an expert). However, the focus of this paper on a "gradient threshold" mechanism is surprising, given the large and detailed literature on how dorsal-ventral patterning, an evolutionary highly conserved process already observed in Nematostella, is controlled by a feedback-rich network architecture. The mechanism by which the BMP signaling gradient forms has been mathematically modelled, initially in *Drosophila* (Mizutani et al., 2005), subsequently also in Nematostella and *Xenopus* (Genikhovich et al., 2015), and finally in zebrafish (Zinski et al., 2017), and is rather well understood. One of its hallmarks is the sudden inversion of the BMP gradient opposite the Chordin expression domain (Iber and Gaglia 2007), which is also well visible in the here-quantified pSMAD gradient. This particular spatial dynamic makes it ever so harder to consider spatially averaged mRNA profiles (Figures 2,4,5,6) when trying to understand the regulatory logic.

This paper here focuses on the transcriptional network downstream of the pSMAD gradient (though Bmp and Smad genes themselves are transcriptional targets). Previous mathematical modelling of part of the here-considered components showed that the spatio-temporal dynamics can be captured when taking the regulatory interactions of the known transcription network into account (https://ars.els-cdn.com/content/image/1-s2.0-S2211124715001813-mmc1.pdf). In light of this, it is not unexpected that a simple threshold-based read-out mechanism cannot match the data. It is surprising that despite the rich quantitative data, no such spatio-temporal mathematical model is explored to understand how the BMP gradient is read out during dorsal-ventral patterning – and the available literature is not mentioned.

Beyond this general criticism that concerns the general approach, I have several specific concerns:

1) BMP target genes: The authors first use RNA-sequencing to systematically identify genes activated by BMP during early zebrafish gastrulation. By doing this, they ended up with a list of 16 genes. All their further analysis is highly dependent on such selection. However, it is not clear how those genes are selected, neither how robust the list is. The authors use 3 methods for the differential expression analysis (edgeR, DESeq, and Cuff diff). Why the authors have chosen those 3 methods, how their results were combined, what is the p-value threshold and whether the p-value is corrected by the number of features is not clear. Also, by changing any of those choices, how much the list of the genes would change is unclear. The resulting list of 16 BMP target genes contains 14 known BMP target genes, and the 2 newly identified target genes are not further studied or found to not show a clear BMP response. As such, it is not clear how much this unbiased approach adds over using known BMP target genes.

2) BMP read-out analysis: In order to test whether the "gradient threshold" model is sufficient to explain the expression pattern of the target genes, the authors compare the activation time and the range of expression of these genes. According to the authors, a threshold-based mechanism would implicate that broadly expressed genes are activated by lower signaling levels. Therefore, if such mechanism is the main driver of gene expression, we should observe a correlation between the activation time and the range of gene expression. The authors find a weak correlation between these variables and based on that exclude a threshold-based mechanism as the main drivers of gene expression. However, only 9 of the 16 genes are analyzed and due to such a low number of genes, the correlation is rather sensitive to the addition or removal of a gene. While the conclusion is likely correct given what is known about the feedback architecture of the BMP-dependent regulatory network (see above), the authors should test the statistical power of their conclusion using boot-strapping.

3) Optogenetics Data: It is assumed that endogenous BMP signalling is not active in high state, but the expression of several genes already jumps during high state (Figure 2) – how does this fit with this assumption? In Figure 5, the fits are shown for the dynamics of the difference between control and the induced data. As the amplitude of the difference differs between genes, the authors conclude that the induction rates must differ. However, also the uninduced control kinetics differ substantially between genes, with some expression profiles increasing and others decreasing over time. Those underlying dynamics likely also affect the extent to which the optogenetic pulse can impact, but this seems not to be considered here. Moreover, the change in amplitude reflects both increased expression and spatial expansion (Figure 5—figure supplement 1). The latter is not considered in the models. In addition, Figure 7 shows that the gradients are very different for high BMP levels, yet the comparison is made to the data in Figure 1. For lower and/or shorter pulses (Figure 6), the response is more diverse, but the presentation in Figure 6 hides the actual data. It would be important that at least a supplementary figure is provided that shows the full data as has been done in Figure 5—figure supplement 1so that one can see the temporal profile of each gene, and ideally also the ISH as in Figure 5—figure supplement 1; Figure 5—figure supplement 1itself should be expanded to show the ISH of all genes analysed in Figure 5 because the spatial data in Figure 5—figure supplement 1 nicely show that the pulses change not only the intensity, but also the range of the signal. This is expected in light of existing mathematical models, and makes it important to explicitly consider space in the analysis. In case of Figure 6, it would be important to relate the spatio-temporal kinetics to those observed in Figure 1 in a more detailed way than currently done.

4) Temporal Modelling: The authors use as a model for the transcript levels dTodt= σdPodt− λTo. It is an unusual assumption that the temporal change in the observed transcript levels, T_o_, depends on the temporal change in the p-SMAD levels, P_o_. Standard transcriptional models would assume that it depends on the p-SMAD levels, i.e. dT0dt=σf(Po)−λT0, where the function f is typically a Hill function. After all, their data provides them with P_o_(t) as well as with T_o_(t). As such, in a first simplistic approach, they could simply analyse the correlation (and transfer functions) between the two. But maybe that's what the authors mean and the above formula is just a typo?

5) Spatial Modelling: Current modelling approaches are much more powerful than what has been used in the paper (i.e. Zinski et al., 2017; Genikhovich et al., 2015). The authors should build on such work and analyse their quantitative data in the context of such more sophisticated models to understand how the different target genes are regulated. It does not help to rely on future modellers to pick up the published data as the modelling will identify missing data to arrive at a conclusion. I suspect for instance that spatial measurements of the response genes will be required at more than just a single state to understand the gradient read-out. For now, this data (Figure 1) seems not even to have been used in the analysis. The FGF/Nodal experiments will presumably only be properly interpretable when analysed in the context of the gradient-generating shuttling model from Zinski et al. as FGF appears to impact Chordin.

6) References: The authors do not comment on how the BMP gradient forms and how this depends on Chordin, even though this becomes highly relevant in the context of their FGF/Nodal experiments – and already explains why the ranges change when Chordin levels are altered, something they highlight as an open problem. They should discuss the relevant publications, i.e. Zinski et al., 2017. They should also discuss the literature on the BMP-dependent networks involved in DV patterning. Moreover, they also do not mention the paper Xiong et al., 2013 in the Discussion, which shows that in zebrafish development cells can sort to create sharp domain boundaries. Finally, they cite the neural tube as an example where a network affects the morphogen read-out, even though in the neural tube, the genes that are activated at lowest SHH levels are expressed first (PAX6 -> OLIG2 -> NKX2.2), which would actually be consistent with a threshold-based read-out.

As mentioned in my review, the authors should use boot-strapping to access the sensitivity of the estimated correlation to individual datapoints, and they should show the raw data for Figure 6.

Reviewer #3:

Rogers and co-workers used a combination of optogenetic and pharmacological approaches to test the extent to which the French Flag model of morphogen gradient interpretation can explain BMP signalling interpretation during zebrafish gastrulation. Using RNA-seq they systematically identified BMP targets during early zebrafish embryogenesis and then studied how altering BMP signalling dynamics impact on target gene expression. To do so, they developed an optogenetic approach by tagging the BMP receptor kinase domain with a LOV domain and targeted the fusion construct to the plasma membrane using a myristoylation motif. The efficacy of this optogenetic approach is demonstrated by the ventralization phenotype (normally induced upon over-expression of BMP) upon light exposure and normal development in the dark. The results presented are inconsistent with the French Flag model on three grounds. First, broader domains of target gene expression do not correlate with lower BMP concentration. Second, spatial shifts in the BMP gradient do not correlate with a corresponding shift in target gene expression. Third, they observe a lack of correlation between spatial broadness of target gene expression and levels of BMP required for activation. Together these results suggest that BMP signaling thresholds are alone insufficient to specify target gene expression. By pharmacological inhibiting Nodal and FGF signaling they show that spatial patterning of BMP targets requires the combinatorial inputs from these signaling systems.

Overall this is an interesting study addressing an important question in developmental biology, the experiments are well-controlled, clearly presented, and extensively discussed. I particularly value the development of a novel optogenetic tool to modulate BMP signalling which I think will be very useful for the community. I recommend publication of this study as it is.

[Editors' note: further revisions were suggested prior to acceptance, as described below.]

Thank you for submitting your article "Optogenetic investigation of BMP target gene expression diversity" for consideration by *eLife*. Your revised article has been evaluated by Naama Barkai (Senior Editor) and a Reviewing Editor.

The authors have made all requested technical corrections, but their added analysis confirms the worry that their main conclusion regarding the BMP-read-out is unsupported. Whilst we value the authors experimental approach, we worry that if the paper is published as is it will add to the confusion in the field, questioning basic principles of morphogen-related patterning without adequate experimental support. Accordingly, while no new data is required, we ask the authors to reword some of their conclusions, to further tone down their claims, by addressing the following comments.

Revisions:

1) The authors chose to analyse 9 BMP-dependent target genes. The selection is not unbiased as they say, but the result of technical limitations (as now explained much more clearly). Of these 9 target genes, two (*cdx4* and *eve1*) are special in that they are expressed only in the margin (Figure 1). When included in the analysis, then expression range and activation time are effectively uncorrelated (Figure 2O). However, if *cdx4* is removed from their analysis, the correlation increases to 65% (new analysis that was added), and if they removed also *eve1*, then the correlation can be expected to increase even further, thus supporting the notion that all analysed target genes, but the two margin-restricted *cdx4* and *eve1*, are controlled in a threshold-based manner. That 2 BMP-dependent genes are constrained by further effects cannot be used to claim that the BMP read-out, in general, is not threshold-based – especially as the margin-restriction must indeed happen BMP-independently.

Also, the additional data that is now provided for Figure 6, which is most helpful, shows that the time points that were sampled are too late to reliably compare activation kinetics. Some of the genes also show transient dynamics that further confound the analysis when time points are sampled only late, especially as the variance is high and few time points could be analysed.

2) Citing Briscoe and Small, 2015, the authors use the neural tube as further support that morphogen gradients are not read out threshold-based. However, the progenitor boundaries have so far not been related to the measured SHH gradient (which was published only in 2015) – only to GBS-GFP, which is a floorplate promotor that responds only to the highest SHH concentration. The fact that the responsiveness of this promotor is limited does not mean that the progenitor boundaries are not read out threshold-based.

---

## [Author Response]

Revisions for this paper:1) Modify title and Abstract. The novel aspect is in the readout along the DV axis, only little data is presented for the readout along the animal-vegetative axis, and the results are confirming a model for spatial readout that has been made in numerous other model systems over the last decades

To better highlight the novel aspects of our study, we have changed the title to “Optogenetic investigation of BMP target gene expression diversity”. We have also modified the final sentence of the Abstract to point out that combinatorial signaling is a feature of other patterning systems: “Our results suggest that, similar to other patterning systems, combinatorial signaling is likely to be a major driver of spatial diversity in BMP-dependent gene expression in zebrafish, rather than activation by different BMP signaling thresholds”

Reviewer #1:The manuscript submitted by Patrick Müller and colleagues deals with the topic of morphogen signalling, in particular with the readout of different morphogen concentrations. Using optogenetic manipulation of BMP signalling in the early zebrafish embryo, the authors come to the conclusion that BMP-dependent gene regulation is not well explained or supported by a differential response to BMP doses, but instead results from combinatorial signalling. The paper is elegant and original in its experimental approaches (optogenetics, mathematical modelling, excellent assay system in vivo).The authors look mostly at temporal regulation of genes with different expression patterns along the D/V axis, assuming that more broadly expressed genes should be activated by lower signalling levels, and since signalling levels increase, these genes should also be activated earlier. Since they do not see such a temporal behaviour, they conclude that the absence of a compelling correlation is inconsistent with the gradient threshold model. Using optogenetic approaches, they show that the differences in the competence to respond to BMP signalling at early stages cannot explain the majority of diversity in activation timing.The major conclusion, namely that combinatorial signalling underlies the spatial diversity in BMP target gene expression, relies mostly on the gene expression patterns along the animal-vegetal axis. However, and as stated by the authors, the morphogen readout of BMP is best described and discussed in the literature with regard to the dorsal-ventral axis. I find it misleading that the major conclusions are derived from an axis along which BMP is not known to act as a morphogen. Different expression pattern in different "cell types" have long been known to arise via combinatorial signalling (Cell papers in 2000; and then many more in the last twenty years).While the role of FGF and Nodal signaling is interesting, the description of the results obtained takes up only one page of the Results, and does not really contribute much to the overall topic of spatiotemporal expression along the DV axis. Yet, at the end, the results from this short aspect of the paper is placed into the title of the manuscript. Again, as I said, I find this misleading and the message conveyed in the title does not reflect the question addressed in the study.

We found that combinatorial signaling contributes not only to expression differences along the animal-vegetal axis but also along the dorsal-ventral axis (Figure 7F-I, Figure 7—figure supplement 3). Nodal and FGF inhibit a subset of BMP target genes on the extreme ventral side, causing their expression to peak around 30% dorsal-ventral embryo length in contrast to other BMP targets. Thus, Nodal and FGF sculpt BMP target gene expression along both the animal-vegetal and dorsal-ventral axes.

So, all of the elegantly obtained data does not answer the initially posed question, and possibly leaves the reader a bit perplexed. Maybe it would be better to reformulate title and Abstract such that the experiments and the message are better aligned.

Our goal was “[…] to identify the factors contributing to differences in BMP target gene expression” and rule out factors that do not contribute. While our optogenetic data suggests that differential transcription kinetics in response to BMP signaling or differential BMP signaling activation thresholds do not define differences in spatiotemporal expression patterns, we do find evidence that spatial differences are predominantly due to combinatorial signaling.

As suggested, we have changed the title and Abstract to reflect that the majority of the manuscript is devoted to assessing whether BMP target gene responses to BMP signaling can explain their observed spatiotemporal expression diversity, whereas less space is dedicated to examining combinatorial signaling. We have also added further clarifications in the Introduction to avoid leaving readers perplexed.

How is the different extend of expression along the gradient achieved? What is their idea?

The major contributor to spatial differences appears to be combinatorial signaling. The more minor differences in range could be due to differential responses to other aspects of BMP signaling that we did not analyze (e.g., proportional activation by different BMP signaling levels).

What about different receptors and/or different ligands (homodimers versus heterodimers) and the expression pattern along the DV axis?

Although we are also fascinated by the regulation of BMP protein distribution (Rogers and Müller, 2019; Pomreinke et al., 2017), our current study is agnostic about how the BMP signaling gradient is formed or modified. Here, we consider the relationship between target gene expression and BMP signaling, directly measured by pSmad1/5/9 immunofluorescence. Since reviewer #2 also raised the question of how the BMP gradient forms (see below), we now elaborate more on what is known.

Reviewer #2:In this manuscript, Rogers et al. studied the read-out of the dorsal-ventral BMP gradient in zebrafish embryos. The authors first use RNA-seq to identify 16 BMP target genes and measured their spatial and temporal expression pattern in zebrafish embryos. They further induce pulses of BMP signaling and evaluate the differences in the gene expression pattern of these genes. By comparing their experimental observations with the predictions from the “gradient threshold” model, the authors conclude that their observations are inconsistent with a model based on threshold of gene activation. The authors then perturb the Nodal and/or FGF signaling pathway and show that these pathways significantly affect the expression pattern of these BMP targets, suggesting that the spatial expression of BMP targets is more dependent on the combination of BMP, Nodal and FGF signaling than on the threshold activation of BMP signal.The question how gradients are read out is highly relevant and the experimental approach appears state-of-the art (though I am not an expert). However, the focus of this paper on a "gradient threshold" mechanism is surprising, given the large and detailed literature on how dorsal-ventral patterning, an evolutionary highly conserved process already observed in Nematostella, is controlled by a feedback-rich network architecture.

Much of the literature suggests that BMP functions as a classical morphogen in the context of zebrafish dorsal-ventral patterning (Tuazon and Mullins, 2015; Barth et al., 1999; Nguyen et al., 1998; Neave et al., 1997; Mullins et al., 1996). Although currently there is no definitive consensus in the field, the threshold-based model remains popular. For example, Zinski, Tajer, and Mullins have recently stated that “It remains unclear how the BMP signaling gradient informs the expression of BMP target genes along the DV axis. It is postulated that cells along the BMP gradient sense the amount of signal, which determines their DV tissue fate as a morphogen. […] However, it is not known whether different BMP direct targets along the DV axis are induced by different thresholds of BMP signaling, different durations of BMP signaling, or some combination of the two.” (Zinski et al., 2018).

This motivates the use of our uniquely well-suited optogenetic approach to directly test this and other pervasive models.

The mechanism by which the BMP signaling gradient forms has been mathematically modelled, initially in *Drosophila* (Mizutani et al., 2005), subsequently also in Nematostella and *Xenopus* (Genikhovich et al., 2015), and finally in zebrafish (Zinski et al., 2017), and is rather well understood. One of its hallmarks is the sudden inversion of the BMP gradient opposite the Chordin expression domain (Iber and Gaglia, 2007), which is also well visible in the here-quantified pSMAD gradient. This particular spatial dynamic makes it ever so harder to consider spatially averaged mRNA profiles (Figures 2,4,5,6) when trying to understand the regulatory logic.

As the reviewer notes, the formation and regulation of the BMP signaling gradient is well studied, and we cite Zinski et al., 2017 as well as our own work on the subject (Pomreinke et al., 2017). In fact, our work showed that gradient formation in zebrafish does not depend on Chordin-mediated shuttling, and the mechanism is different from what had been described in *Drosophila*, *Xenopus*, and *Nematostella* (Pomreinke et al., 2017). Nevertheless, as mentioned in response to reviewer #1, our current study is agnostic about the mechanism of BMP signaling gradient formation since we focus on the readout by different target genes that respond to the same signaling gradient. Although we do quantify BMP signaling in our present work, we do not examine the formation or modification of the signaling gradient itself.

We now elaborate more in the Introduction on what is known about BMP gradient formation. We had previously referenced a review (Rogers and Müller, 2019) in which we discussed the Mizutani et al., 2005, Genikhovich et al., 2015, and Zinski et al., 2017 papers suggested by reviewer #2, but we now also include these primary references as well as Iber and Gaglia, 2007 directly in the current manuscript.

This paper here focuses on the transcriptional network downstream of the pSMAD gradient (though Bmp and Smad genes themselves are transcriptional targets). Previous mathematical modelling of part of the here-considered components showed that the spatio-temporal dynamics can be captured when taking the regulatory interactions of the known transcription network into account (https://ars.els-cdn.com/content/image/1-s2.0-S2211124715001813-mmc1.pdf). In light of this, it is not unexpected that a simple threshold-based read-out mechanism cannot match the data. It is surprising that despite the rich quantitative data, no such spatio-temporal mathematical model is explored to understand how the BMP gradient is read out during dorsal-ventral patterning – and the available literature is not mentioned.

As mentioned above, there is currently no consensus on whether BMP acts as a morphogen using a threshold-based strategy during zebrafish dorsal-ventral patterning, and the many factors that are known to regulate BMP gradient formation do not necessarily rule out a threshold-based readout for target genes (Zinski et al., 2018). We agree that further mathematical modeling could help clarify the relationship between BMP signaling and target gene expression. However, meaningful modeling would require significant amounts of additional data, including measurements of the dynamic changes in gene expression amplitudes and spatial profiles over time, which we cannot include in a reasonable timeline.

Beyond this general criticism that concerns the general approach, I have several specific concerns:1) BMP target genes: The authors first use RNA-sequencing to systematically identify genes activated by BMP during early zebrafish gastrulation. By doing this, they ended up with a list of 16 genes. All their further analysis is highly dependent on such selection. However, it is not clear how those genes are selected, neither how robust the list is. The authors use 3 methods for the differential expression analysis (edgeR, DESeq, and Cuff diff). Why the authors have chosen those 3 methods, how their results were combined, what is the p-value threshold and whether the p-value is corrected by the number of features is not clear. Also, by changing any of those choices, how much the list of the genes would change is unclear. The resulting list of 16 BMP target genes contains 14 known BMP target genes, and the 2 newly identified target genes are not further studied or found to not show a clear BMP response. As such, it is not clear how much this unbiased approach adds over using known BMP target genes.

We defined high-confidence BMP target genes as those that were both significantly upregulated in BMP-overexpressing embryos and downregulated in Chordin-overexpressing embryos at shield stage (Figure 1—figure supplement 1). Significance was determined by the well-established and widely used differential expression analysis tools edgeR, DESeq, and Cuff diff (Seyednasrollah et al., 2015), and the resulting p-values can be found in the Supplementary file 1. High-confidence genes consist only of those that were deemed significantly differentially expressed by all three programs. We now also state in the Materials and methods section that the p-value threshold for differentially expressed genes was set to 0.05.

While we identified the vast majority of known direct BMP targets (Zinski et al., 2018) in at least one condition (Supplementary file 1), we note that our strict definition excludes known target genes such as *tp63* (which is not expressed at shield stage and therefore not downregulated in Chordin-overexpressing embryos). However, we wanted to focus on an unbiased set of high-confidence BMP targets as representatives of the patterning system, selected based on clear and logical rules. We believe our definition achieves this aim, and we favor this approach over manually picking a subset of genes. Our approach is further validated by the corroboration of the RNA-seq data by *in situ* hybridization (Figure 1—figure supplement 1C) and by the fact that 14/16 high-confidence target genes have previously been described to be regulated by BMP in zebrafish.

We have published the raw data in the GEO repository, and we also now provide the complete differential expression analysis in the Supplementary file 1.

2) BMP read-out analysis: In order to test whether the "gradient threshold" model is sufficient to explain the expression pattern of the target genes, the authors compare the activation time and the range of expression of these genes. According to the authors, a threshold-based mechanism would implicate that broadly expressed genes are activated by lower signaling levels. Therefore, if such mechanism is the main driver of gene expression, we should observe a correlation between the activation time and the range of gene expression. The authors find a weak correlation between these variables and based on that exclude a threshold-based mechanism as the main drivers of gene expression. However, only 9 of the 16 genes are analyzed and due to such a low number of genes, the correlation is rather sensitive to the addition or removal of a gene. While the conclusion is likely correct given what is known about the feedback architecture of the BMP-dependent regulatory network (see above), the authors should test the statistical power of their conclusion using boot-strapping.

As described in the legends for Figure 1 and Figure 2 and in the Materials and methods section, unfortunately not all 16 high-confidence target genes provided robust FISH signal and some of the NanoString probes failed; thus, some genes could not be spatially or temporally assessed, respectively. We were therefore only able to include 9 genes in this analysis, which we now point out more explicitly in the main text.

We carried out the suggested bootstrapping analysis (Author response table 1) as well as jackknife resampling as its linear approximation (Author response table 2). The Pearson correlation coefficient for the data shown in Figure 2O is -0.23, compared to -0.24 – 0.20 for the bootstrapping analysis and an average of -0.24 – 0.18 for the jackknife estimator. We determined that the weak correlation is relatively insensitive to all data points except for *cdx4*, removal of which results in a more negative correlation (-0.65). However, there is no obvious reason to exclude *cdx4* from the analysis.

**Author response table 1. resptable1:** Bootstrapping analysis for Figure 2O.

100 random bootstrapped Pearson correlation coefficients
-0.55	0.04	-0.22	-0.16	-0.13	-0.33	-0.11	-0.39	-0.35	-0.02
-0.29	-0.38	-0.13	-0.38	-0.09	-0.16	-0.24	-0.10	-0.38	-0.02
-0.26	-0.13	-0.45	0.11	-0.08	-0.38	-0.26	-0.13	-0.17	-0.22
0.02	-0.08	-0.21	-0.33	-0.43	0.03	-0.62	-0.36	-0.24	-0.26
-0.71	-0.20	-0.01	-0.66	-0.05	-0.26	0.11	-0.25	0.15	-0.02
-0.13	-0.31	0.11	-0.35	0.03	-0.21	-0.16	-0.31	-0.66	-0.31
-0.33	-0.36	-0.08	-0.33	-0.66	-0.25	-0.14	-0.39	-0.33	-0.04
-0.62	-0.39	-0.13	-0.11	-0.55	-0.39	-0.03	-0.20	-0.08	-0.30
-0.22	-0.24	-0.04	-0.25	0.15	-0.14	-0.13	-0.68	-0.13	-0.20
-0.16	0.11	-0.22	-0.21	-0.24	-0.68	-0.31	-0.66	-0.28	-0.22

**Author response table 2. resptable2:** Jackknife analysis for Figure 2O.

Pearson correlation coefficients with the indicated gene removed
*bambia*	*cdx4*	*eve1*	*foxi1*	*gata2a*	*klf2b*	*sizzled*	*tfap2c*	*ved*
-0.21	-0.65	-0.17	-0.35	-0.10	-0.04	-0.13	-0.23	-0.24

We have also extended our discussion to point out a subset of genes that shows behaviors consistent with the gradient threshold model: “We note that a subset of three genes (sizzled, ved, and bambia) that are neither restricted to nor excluded from the margin do show a strong negative correlation between range and activation time _(ρp_ = -0.99, although this correlation based on only three data points might be spurious) (Figure 2O) as well as activation dynamics that could be roughly commensurate with signaling input (Figure 6C and Figure 6—figure supplement 1), consistent with the gradient threshold model. However, it remains to be determined to what extent this subset of genes (or others) quantitatively follows the input-output relationships predicted by the gradient threshold model”.

3) Optogenetics Data: It is assumed that endogenous BMP signalling is not active in high state, but the expression of several genes already jumps during high state (Figure 2) – how does this fit with this assumption?

Genes with high levels of transcripts at 2.75 hpf (*id2a* and *smad6a*) are maternally provided (White et al., 2017; Chong et al., 2005), making it difficult to determine their activation times (Figure 2E,H). These genes were therefore excluded from our activation time analyses. However, some genes that are not maternally provided are expressed around high stage (~3.5 hpf, e.g., Figure 2D,L). We based the lack of BMP signaling at high stage on our spatiotemporal analysis of pSmad1/5/9 levels, which indicate undetectable pSmad at high stage (Figure 1D-E and Figure 7—figure supplement 2B-K). It therefore seems possible that other factors are responsible for the early activation of these genes.

In Figure 5, the fits are shown for the dynamics of the difference between control and the induced data. As the amplitude of the difference differs between genes, the authors conclude that the induction rates must differ. However, also the uninduced control kinetics differ substantially between genes, with some expression profiles increasing and others decreasing over time. Those underlying dynamics likely also affect the extent to which the optogenetic pulse can impact, but this seems not to be considered here.

The optogenetic experiments in Figure 5 included uninjected controls collected at the same time points as Opto-BMP-expressing siblings. We were therefore able to subtract the endogenous transcripts from the induced transcripts at all time points. The fact that the endogenous transcript levels have different temporal dynamics is therefore controlled for, both in our data and in the mathematical modeling (Figure 5—figure supplement 1).

Moreover, the change in amplitude reflects both increased expression and spatial expansion (Figure 5—figure supplement 1). The latter is not considered in the models.

We observe that all cells in the embryo can respond to the roughly uniform optogenetically delivered signaling pulse (Figure 3D,E), accounting for the expected spatial expansion in gene expression (the optogenetic constructs act downstream of many of the endogenous extracellular inhibitory factors on the dorsal side). The NanoString data includes all transcripts, regardless of whether they are expressed in the endogenous spatial domain. Because we subtract the uninjected transcripts from the transcripts in Opto-BMP-expressing siblings, spatial changes are accounted for in the data and modeling.

In addition, Figure 7 shows that the gradients are very different for high BMP levels, yet the comparison is made to the data in Figure 1.

Figure 7J quantifies spatial changes in BMP target gene expression when *BMP* mRNA is injected at the one-cell stage, but it does not provide information about target gene activation kinetics. In contrast, in Figure 5M-Z we quantify target gene activation kinetics by optogenetically introducing an acute BMP signaling pulse at shield stage (~6.75 hpf). This allowed us to address the following question: Can the differences in the endogenous spatial expression ranges be explained by differences in activation kinetics? To answer this question, we compared activation kinetics (Figure 5M-Z) to endogenous spatial expression ranges (Figure 1P-Y).

For lower and/or shorter pulses (Figure 6), the response is more diverse, but the presentation in Figure 6 hides the actual data. It would be important that at least a supplementary figure is provided that shows the full data as has been done in Figure 5—figure supplement 1 so that one can see the temporal profile of each gene, and ideally also the ISH as in Figure 5—figure supplement 1; Figure 5—figure supplement 1itself should be expanded to show the ISH of all genes analysed in Figure 5 because the spatial data in Figure 5—figure supplement 1nicely show that the pulses change not only the intensity, but also the range of the signal. This is expected in light of existing mathematical models, and makes it important to explicitly consider space in the analysis. In case of Figure 6, it would be important to relate the spatio-temporal kinetics to those observed in Figure 1 in a more detailed way than currently done.

With the data in Figure 6 we argue that different activation thresholds are unlikely to explain the differences in the endogenous spatial expression ranges (Figure 1 P-Y). According to the gradient threshold model, genes with narrow expression ranges should require high levels of BMP signaling to be activated. However, we observed that both broadly and narrowly expressed genes were activated by low levels of signaling (Figure 6C,E), indicating that differential activation thresholds do not explain the differential spatial ranges observed in Figure 1P-Y.

As suggested, we have re-plotted the data from Figure 6 C-F to show the temporal profile of each gene (Figure 6—figure supplement 1), and the raw data is also available in the Source Data file. Note that, in contrast to Figure 4, Figure 5, and Figure 5—figure supplement 1 in which 10 time points during and after exposure were collected, in Figure 6 we only collected 3 time points (30, 40, and 50 min into exposure). Collecting more time points would have been logistically challenging due to the greater number of conditions and sample numbers, and temporal data was not required to answer the question meant to be addressed by these experiments (in contrast to Figure 4, Figure 5, and Figure 5—figure supplement 1).

The question we sought to address in Figure 6 is whether more broadly expressed genes, such as *klf2b* and *gata2a*, are more likely to be activated by lower signaling levels than more narrowly expressed genes, such as *foxi1* (Figure 1). The new Figure 6—figure supplement 1 shows that at a low-amplitude and short duration of BMP signaling (70 lux, 10 min), *foxi1* is significantly activated, whereas *klf2b* and *gata2a* are not (see Figure 6 and Materials and methods for statistical analyses). This is inconsistent with the gradient threshold model, complementing the results shown in Figure 6.

Although we do not have additional *in situ* hybridization data for the 30 min pulse shown in Figure 5—figure supplement 1, in Author response image 1 we show the requested *in situ* hybridization of four BMP target genes responding to the different amplitudes and durations of BMP signaling applied in Figure 6. This data shows that the activation sensitivity of these genes does not correspond to the extent of the spatial expression domain. For example, the endogenous expression range of *foxi1* and *eve1* is comparable (Figure 1R,S), *foxi1* is much more sensitive to a short and low-intensity Opto-BMP pulse than *eve1* (Author response image 1, Figure 6—figure supplement 1D,F).

**Author response image 1. sa2fig1:** BMP target gene responses to different amplitudes and durations of BMP signalling. Embryos injected with Opto-BMP and uninjected siblings were either left in the dark (A) or exposed to 70 or 3900 lux blue light for 10 (B) or 20 (C) min starting at shield stage (~6.75 hpf), and fixed 20 min post-exposure. BMP target gene expression in response to the resulting low- and high-amplitude BMP signaling pulses (Figure 6B) was assessed using colorimetric *in situ* hybridization.

Since all cells can respond to optogenetically delivered BMP signaling pulses, the expansion of BMP target gene spatial expression ranges is expected and accounted for in our modelling by subtracting the uninjected transcript counts at each time point, as described above.

4) Temporal Modelling: The authors use as a model for the transcript levels dTodt= σdPodt− λTo. It is an unusual assumption that the temporal change in the observed transcript levels, T_o_, depends on the temporal change in the p-SMAD levels, P_o_. Standard transcriptional models would assume that it depends on the p-SMAD levels, i.e. dT0dt=σf(Po)−λT0, where the function f is typically a Hill function. After all, their data provides them with P_o_(t) as well as with T_o_(t). As such, in a first simplistic approach, they could simply analyse the correlation (and transfer functions) between the two. But maybe that's what the authors mean and the above formula is just a typo?

We thank the reviewer for pointing out this typographical error, and we apologize for any resulting confusion. We have now corrected the equations with dot notation to correctly point out the derivatives. As mentioned in the Materials and methods section, we wished to apply the simplest model of induction and decay and therefore did not use a Hill function for *σ*.

5) Spatial Modelling: Current modelling approaches are much more powerful than what has been used in the paper (i.e. Zinski et al., 2017; Genikhovich et al., 2015). The authors should build on such work and analyse their quantitative data in the context of such more sophisticated models to understand how the different target genes are regulated. It does not help to rely on future modellers to pick up the published data as the modelling will identify missing data to arrive at a conclusion. I suspect for instance that spatial measurements of the response genes will be required at more than just a single state to understand the gradient read-out. For now, this data (Figure 1) seems not even to have been used in the analysis.

Our modeling approach with the data in Figure 5 was designed to answer a specific and important question: Can differences in gene activation kinetics explain different spatial expression ranges? This approach suggested that activation kinetics have little explanatory power for the expression ranges of BMP-dependent genes. We agree that further modeling could complete our understanding of feedback interactions in BMP gradient formation and interpretation, but these modeling efforts will require significant amounts of new data to arrive at meaningful conclusions, such as spatiotemporal measurements of gene expression and further information on the role of Nodal and FGF. We therefore anticipate carrying out these analyses in future studies when new data can be obtained to avoid underdetermined models.

The FGF/Nodal experiments will presumably only be properly interpretable when analysed in the context of the gradient-generating shuttling model from Zinski et al. as FGF appears to impact Chordin.

Although our own previous work as well as the paper by Zinski et al., 2017 shows that Chordin does not play a role as a shuttling molecule in the formation of the BMP protein gradient in early zebrafish embryos (Pomreinke et al., 2017; Zinski et al., 2017), Chordin clearly impacts the BMP signaling gradient as an inhibitor and sink of BMP activity (Rogers and Müller, 2019). We mention the known role of Chordin in regulating BMP and the known role of Nodal and FGF in regulating *chordin* expression (Figure 7K). While the specific mechanisms that mediate these interactions are known in part but still merit further investigation (see below), we have precisely quantified the effects of Nodal and FGF loss on the BMP signaling gradient at shield stage, and shown that the effects on target genes are not solely due to changes in BMP signaling (Figure 7L,M).

6) References: The authors do not comment on how the BMP gradient forms and how this depends on Chordin, even though this becomes highly relevant in the context of their FGF/Nodal experiments – and already explains why the ranges change when Chordin levels are altered, something they highlight as an open problem. They should discuss the relevant publications, i.e. Zinski et al., 2017. They should also discuss the literature on the BMP-dependent networks involved in DV patterning.

In the Introduction we provided only a brief explanation of BMP gradient formation, because in this work we are interested in target gene responses, not how the gradient forms. However, as mentioned above, we now expand this discussion of how the BMP gradient forms and the role of Chordin, and discuss in more detail our previous findings (Pomreinke et al., 2017) and those of Zinski et al., 2017. We also describe the multiple ways in which FGF and Nodal are known to affect BMP signaling in zebrafish and cite the relevant literature, including a description of how Nodal and FGF affect *Chordin* expression (Figure 7K, Figure 7—figure supplement 2ZA), (Rogers and Müller, 2019; Sapkota et al., 2007; Varga et al., 2007; Maegawa et al., 2006; Londin et al., 2005; Fürthauer et al., 2004; Kudoh et al., 2004; Koshida et al., 2002; Gritsman et al., 1999; Fürthauer et al., 1997).

We do not claim that it is an open problem why ranges change when Chordin levels are altered. Rather, in the original manuscript refers to two unexplained observations: First, that FGF loss but not Nodal loss causes an increase in BMP signaling, and second, that inhibition of both FGF and Nodal has different effects than inhibition of either alone. These observations are surprising because Nodal is thought to be required for *FGF* expression, meaning that the effect of losing both Nodal and FGF should be equivalent to losing Nodal alone. In addition, we note that Nodal mutants have reduced *Chordin* expression (Gritsman et al., 1999), further highlighting that future work is needed to explain why the BMP signaling gradient is not altered in shield-stage embryos lacking Nodal signaling, even though presumably they produce less Chordin protein. It does not appear to be the case that changes in Chordin alone can explain these results.

Moreover, they also do not mention the paper Xiong et al., 2013 in the Discussion, which shows that in zebrafish development cells can sort to create sharp domain boundaries.

Thank you for this suggestion. We now cite (Xiong et al., 2013) as well as (Akieda et al., 2019) and mention cell sorting-based boundary sharpening mechanisms as a method to refine responses to noisy signaling gradients, and as a possible explanation for how the embryo robustly compensates for excess BMP signaling.

Finally, they cite the neural tube as an example where a network affects the morphogen read-out, even though in the neural tube, the genes that are activated at lowest SHH levels are expressed first (PAX6 -> OLIG2 -> NKX2.2), which would actually be consistent with a threshold-based read-out.As mentioned in my review, the authors should use boot-strapping to access the sensitivity of the estimated correlation to individual datapoints, and they should show the raw data for Figure 6.

The literature we cited demonstrates that combinatorial signaling from BMP, Wnt, and Shh signaling gradients together with cross-talk between target genes patterns the well-characterized neural tube (Briscoe and Small, 2015). Even though the gene expression progression described by the reviewer superficially appears to be consistent with the threshold model, Briscoe and Small stated that “*[…] signaling gradients establish initial conditions that polarize the tissue, but there is no strict correspondence between specific morphogen thresholds and boundary positions”* (Briscoe and Small, 2015) since “*[…] the induction of Nkx2.2 and Olig2 does not appear to be determined simply by a fixed threshold of Gli activity*” (Balaskas et al., 2012), and boundaries are rather set by a Shh-driven AC-DC circuit (Perez-Carrasco et al., 2018; Panovska-Griffiths et al., 2013). This highlights the importance of testing models using multiple approaches.

[Editors' note: further revisions were suggested prior to acceptance, as described below.]

The authors have made all requested technical corrections, but their added analysis confirms the worry that their main conclusion regarding the BMP-read-out is unsupported. Whilst we value the authors experimental approach, we worry that if the paper is published as is it will add to the confusion in the field, questioning basic principles of morphogen-related patterning without adequate experimental support. Accordingly, while no new data is required, we ask the authors to reword some of their conclusions, to further tone down their claims, by addressing the following comments.Revisions:1) The authors chose to analyse 9 BMP-dependent target genes. The selection is not unbiased as they say, but the result of technical limitations (as now explained much more clearly).

The 16 genes classified as “high-confidence” were defined as those that were both significantly upregulated in the *BMP*-overexpression condition and downregulated in the *Chordin*-overexpression condition in our RNA-sequencing experiment. Because FISH and NanoString probes failed in some cases, we were able to analyze the spatial expression of 10/16 genes and the temporal expression of 14/16. Although we do not think that these technical limitations resulted in a selection bias, we have removed the single instance in the Discussion where the 16 high-confidence target genes were referred to as “*an unbiased set*”.

Of these 9 target genes, two (cdx4 and eve1) are special in that they are expressed only in the margin (Figure 1). When included in the analysis, then expression range and activation time are effectively uncorrelated (Figure 2O). However, if cdx4 is removed from their analysis, the correlation increases to 65% (new analysis that was added), and if they removed also eve1, then the correlation can be expected to increase even further, thus supporting the notion that all analysed target genes, but the two margin-restricted cdx4 and eve1, are controlled in a threshold-based manner. That 2 BMP-dependent genes are constrained by further effects cannot be used to claim that the BMP read-out, in general, is not threshold-based – especially as the margin-restriction must indeed happen BMP-independently.

As noted above, the expression of 2/10 spatially assessed BMP target genes was margin-restricted (*cdx4* and *eve1*). In addition, the expression of a further 4/10 genes was excluded from the margin (*foxi1*, *klf2b*, *gata2a*, and *tfap2c*). Importantly, when selecting which genes to spatially assess as representative BMP targets, we chose genes based only on 2 criteria: 1) significant up/downregulation in *BMP/Chordin*-overexpressing embryos, respectively, and 2) functional FISH probes. It is therefore striking that, when selected without regard for spatial expression profiles, the majority of the high-confidence BMP target genes exhibit special expression profiles (6/10). We argue that the most dramatic differences in spatial expression probably arise from these characteristic profiles likely resulting from combinatorial signaling. However, as described in the Discussion section, if all of the special genes are excluded from the analysis in Figure 2O (6/9, the majority), the remaining 3 do show a negative correlation between range and activation time. We note that this subset of genes (or others) could behave consistently with the gradient threshold model, although given that only 3 genes qualified it is difficult to state this conclusively.

To clarify the expectations of the gradient threshold model, we have carried out an additional analysis taking our measured spatiotemporal BMP signaling profiles into account. Author response image 2 shows the spatial expression ranges and calculated activation times for hypothetical BMP target genes with different activation thresholds, based on the pSmad1/5/9 data shown in Figure 1E. We postulated 180 BMP target genes with different linearly spaced expression ranges along the dorsal-ventral axis, and linked each range with the corresponding pSmad1/5/9 level at 7.25 hpf. We then queried the pSmad1/5/9 time course data and determined the time at which each pSmad level first occurred (Author response image 2). We defined these time points as the activation times of the postulated BMP target genes. As expected, genes activated by low levels of signaling (“low-threshold genes”) should be expressed early and at a long range, whereas genes requiring higher levels of signaling (“high-threshold genes”) should be expressed later and at a short range (Author response image 2). Additionally, this analysis shows the expected monotonically decreasing relationship between expression range and activation time. However, it is also evident that the relationship between range and activation time isn’t necessarily linear, depending on where the relevant threshold range lies. We have therefore removed the linear fit from Figure 2O.

**Author response image 2. sa2fig2:** The activation timing and expression ranges for hypothetical genes with different activation thresholds were calculated based on quantified pSmad1/5/9/ activation kinetics (Figure 1E).

In addition, as suggested, we have now also updated the Results section to describe what happens when only the margin-restricted genes are excluded from the analysis in Figure 2O: “The gradient threshold model predicts a monotonic decrease when comparing range and activation time. While this relationship is not observed for the entire dataset (Figure 2O), there is a decreasing monotonic trend when *foxi1*, *eve1*, and *cdx4* are excluded (note that in contrast to the other genes, the expression of *eve1* and *cdx4* was only quantified in the embryonic margin (Figure 1—figure supplement 1E, Materials and methods)). This suggests the possibility that subsets of BMP target genes may behave consistently with the gradient threshold model. We therefore sought to investigate the relationship between BMP signaling and target gene expression further using an optogenetic strategy.”

As stated in the Discussion, we agree that “[…] it remains to be determined to what extent this subset of genes (or others) quantitatively follows the input-output relationships predicted by the gradient threshold model.”. To make this caveat clearer throughout the manuscript, we have further altered the indicated text:

Abstract: “Transcriptional responses to optogenetically delivered high- and low-amplitude BMP signaling pulses indicate that spatiotemporal expression is not fully defined by different BMP signaling activation thresholds. Additionally, we observed negligible correlations between spatiotemporal expression and transcription kinetics for the majority of analyzed genes in response to BMP signaling pulses. […] Our results suggest that, similar to other patterning systems, combinatorial signaling is likely to be a major driver of spatial diversity in BMP-dependent gene expression in zebrafish.”

Introduction: “Further, target gene responses to high- and low-amplitude signaling pulses suggest that not all spatiotemporal target gene expression differences are due to different signaling activation thresholds (Figure 6).”

Results: “As expected, target activation was generally stronger following higher amplitude, longer duration pulses (Figure 6C-F and Figure 6—figure supplement 1). However, after a 10 min low-amplitude exposure, the third most narrowly expressed gene, *foxi1*, was significantly activated, whereas the broader genes were not robustly induced (Figure 6C). A longer 20 min low-amplitude pulse significantly activated both narrowly and broadly expressed genes (Figure 6E). A 10 min low-amplitude pulse significantly activated two of the top 50% earliest expressed genes (*foxi1* and *smad7*), whereas a 20 min low-amplitude pulse significantly activated both early and late-expressed genes (Figure 6D,F). High-amplitude pulses activated genes of all ranges and activation times (Figure 6C-F).

Our experiments exposing embryos to different amplitude BMP signaling pulses suggest that not all spatiotemporal target gene expression differences are due to different signaling activation thresholds, although a subset may be (see Discussion).”

Discussion: “In the context of zebrafish dorsal-ventral patterning, our data suggest minor roles for gene-specific activation thresholds in generating BMP target gene expression diversity. We did not find a clear monotonically decreasing relationship between activation time and gene expression range (Figure 2O), suggesting that more broadly expressed genes are not consistently more likely to be activated by the low levels of BMP present early (Figure 1D-E). We were also unable to detect an unambiguous correlation between range and the levels of signaling required for activation (Figure 6C,E). This suggests that not all BMP target expression boundaries are positioned by gene-specific BMP signaling thresholds (Figure 1A).”

Discussion: “Our results do not rule out the possibility that a different subset of BMP target genes may behave more consistently with these models. We focused on a set of high-confidence BMP targets (Figure 1—figure supplement 1), but other known targets were excluded from our analyses (Supplementary file 1). For example, the BMP target gene tp63 (Bakkers et al., 2002) is not expressed at shield stage, and was therefore excluded since it was not downregulated by chordin overexpression in our RNA-sequencing experiment (Supplementary file 1). We note that a subset of three genes (sizzled, ved, and bambia) that are neither restricted to nor excluded from the margin do show a monotonically decreasing relationship between range and activation time (Figure 2O) as well as activation dynamics that could be roughly commensurate with signaling input (Figure 6C and Figure 6—figure supplement 1), consistent with the gradient threshold model. However, it remains to be determined to what extent this subset of genes (or others) quantitatively follows the input-output relationships predicted by the gradient threshold model.”

Discussion: “Our results suggest that much of the spatial diversity in BMP target gene expression arises from combinatorial signaling.”

Also, the additional data that is now provided for Figure 6, which is most helpful, shows that the time points that were sampled are too late to reliably compare activation kinetics. Some of the genes also show transient dynamics that further confound the analysis when time points are sampled only late, especially as the variance is high and few time points could be analysed.

We agree that in Figure 6 activation kinetics cannot be reliably established. As such, we do not base any conclusions on activation kinetics derived from the data shown in Figure 6. The experiment in Figure 6 was only designed to determine which genes are robustly induced by different levels of signaling. Technical limitations prevented us from collecting the large numbers of samples that would have been necessary to assess induction kinetics in these different scenarios. We therefore chose to focus on time points near the maximum induction level found in other experiments (Figure 4), in which more time points were assessed (and kinetics could be established).

2) Citing Briscoe and Small, 2015, the authors use the neural tube as further support that morphogen gradients are not read out threshold-based. However, the progenitor boundaries have so far not been related to the measured SHH gradient (which was published only in 2015) – only to GBS-GFP, which is a floorplate promotor that responds only to the highest SHH concentration. The fact that the responsiveness of this promotor is limited does not mean that the progenitor boundaries are not read out threshold-based.

We have now cut the following sentence from the Discussion that may have been misleading: “In vertebrates, Wnt, Shh, and BMP pattern the neural tube using a similar combinatorial mechanism together with target gene cross-regulation (Briscoe and Small, 2015).” The remainder of the references to Briscoe and Small 2015 use it either to demonstrate the ubiquity of signaling gradients in development or to emphasize the importance of combinatorial interactions in patterning.

**References**

Balaskas, N., Ribeiro, A., Panovska, J., Dessaud, E., Sasai, N., Page, K.M., Briscoe, J., and Ribes, V. (2012). Gene regulatory logic for reading the Sonic Hedgehog signaling gradient in the vertebrate neural tube. Cell 148, 273-84.

Panovska-Griffiths, J., Page, K.M., and Briscoe, J. (2013). A gene regulatory motif that generates oscillatory or multiway switch outputs. J R Soc Interface 10, 20120826.

Perez-Carrasco, R., Barnes, C.P., Schaerli, Y., Isalan, M., Briscoe, J., and Page, K.M. (2018). Combining a toggle switch and a repressilator within the AC-DC circuit generates distinct dynamical behaviors. Cell Syst 6, 521-530 e3.

Seyednasrollah, F., Laiho, A., and Elo, L.L. (2015). Comparison of software packages for detecting differential expression in RNA-seq studies. Brief Bioinform 16, 59-70.